# Dynamically regulated two-site interaction of viral RNA to capture host translation initiation factor

Shunsuke Imai [1,4] ✉, Hiroshi Suzuki [2,4], Yoshinori Fujiyoshi [2] & Ichio Shimada [1,3] ✉

Many RNA viruses employ internal ribosome entry sites (IRESs) in their genomic RNA to commandeer the host's translational machinery for replication. The IRES from *encephalomyocarditis* virus (EMCV) interacts with eukaryotic translation initiation factor 4 G (eIF4G), recruiting the ribosomal subunit for translation. Here, we analyze the three-dimensional structure of the complex composed of EMCV IRES, the HEAT1 domain fragment of eIF4G, and eIF4A, by cryo-electron microscopy. Two distinct eIF4G-interacting domains on the IRES are identified, and complex formation changes the angle therebetween. Further, we explore the dynamics of these domains by using solution NMR spectroscopy, revealing conformational equilibria in the microsecond to millisecond timescale. In the lowly-populated conformations, the base-pairing register of one domain is shifted with the structural transition of the three-way junction, as in the complex structure. Our study provides insights into the viral RNA's sophisticated strategy for optimal docking to hijack the host protein.

RNA viruses subvert the host translational machinery to engage cellular ribosomes, which is essential for promoting viral genome translation for replication[1–4]. One such mechanism utilizes internal ribosome entry sites (IRESs) in the 5′ untranslated regions of the RNA genome, which commandeer the host translational machinery through multiple RNA–RNA and/or RNA–protein interactions. The regulation of IRES-dependent translation is therefore a critical step for the infection and replication of these viruses, with multiple effects on virulence, tissue tropism, and pathogenicity[2]. Moreover, some eukaryotic cellular mRNAs may also capitalize on the IRES-dependent translation mechanism to regulate processes such as development and responses to cellular stresses[3,4].

IRESs are cis-acting RNA elements that typically adopt three-dimensional structures, either to interact directly with the 40S ribosomal subunit, or to engage translation factors for recruiting the 40S subunit. Among the IRESs identified thus far, that of *encephalomyocarditis* virus (EMCV) in the *Picornaviridae* family exhibits one of the highest translational efficiencies, and requires the eukaryotic initiation factors (eIFs) 4G, 4A, 2, and 3 to recruit the 40S subunit for translation initiation[5]. This requirement of the EMCV IRES is similar to those of some eukaryotic cellular IRESs[6,7], suggesting mechanistic similarities to the eukaryotic counterparts. The EMCV IRES element interacts directly with the HEAT1 domain of eIF4G (eIF4G^HEAT1), and this interaction is further enhanced when eIF4G^HEAT1 is in complex with eIF4A[8,9] (Fig. 1a, b). Within the EMCV IRES, the region G680-C787, designated as J-K-St in this study, is responsible for the specific capture of the eIF4G^HEAT1. We previously determined the structure of this J-K-St region by solution NMR spectroscopy[9]. It is composed of two stem-loops (J and K domains) bifurcating from a base stem (St domain) (Fig. 1c). At the three-way junction, a highly conserved sequence of six adenosines forms a short stem loop, termed A_SL, which interacts with the J, K, and St domains at the junctional part. Using biochemical assays, several interaction sites on the J-K-St or eIF4G^HEAT1 have been reported[10–12]. Although eIF4G^HEAT1 reportedly binds to cellular RNAs[13,14], the interaction between J-K-St and eIF4G^HEAT1 is stronger and more sequence- and/ or structure specific[15], suggesting that J-K-St has evolved to specifically

[1]RIKEN Center for Biosystems Dynamics Research, Tsurumi-ku, Yokohama 230-0045, Japan. [2]Cellular and Structural Physiology Laboratory (CeSPL), Tokyo Medical and Dental University, Bunkyo-ku, Tokyo 113-8510, Japan. [3]Graduate School of Integrated Sciences for Life, Hiroshima University, Higashi-Hiroshima 739-8528, Japan. [4]These authors contributed equally: Shunsuke Imai, Hiroshi Suzuki. ✉e-mail: shunsuke.imai.ku@riken.jp; ichio.shimada@riken.jp

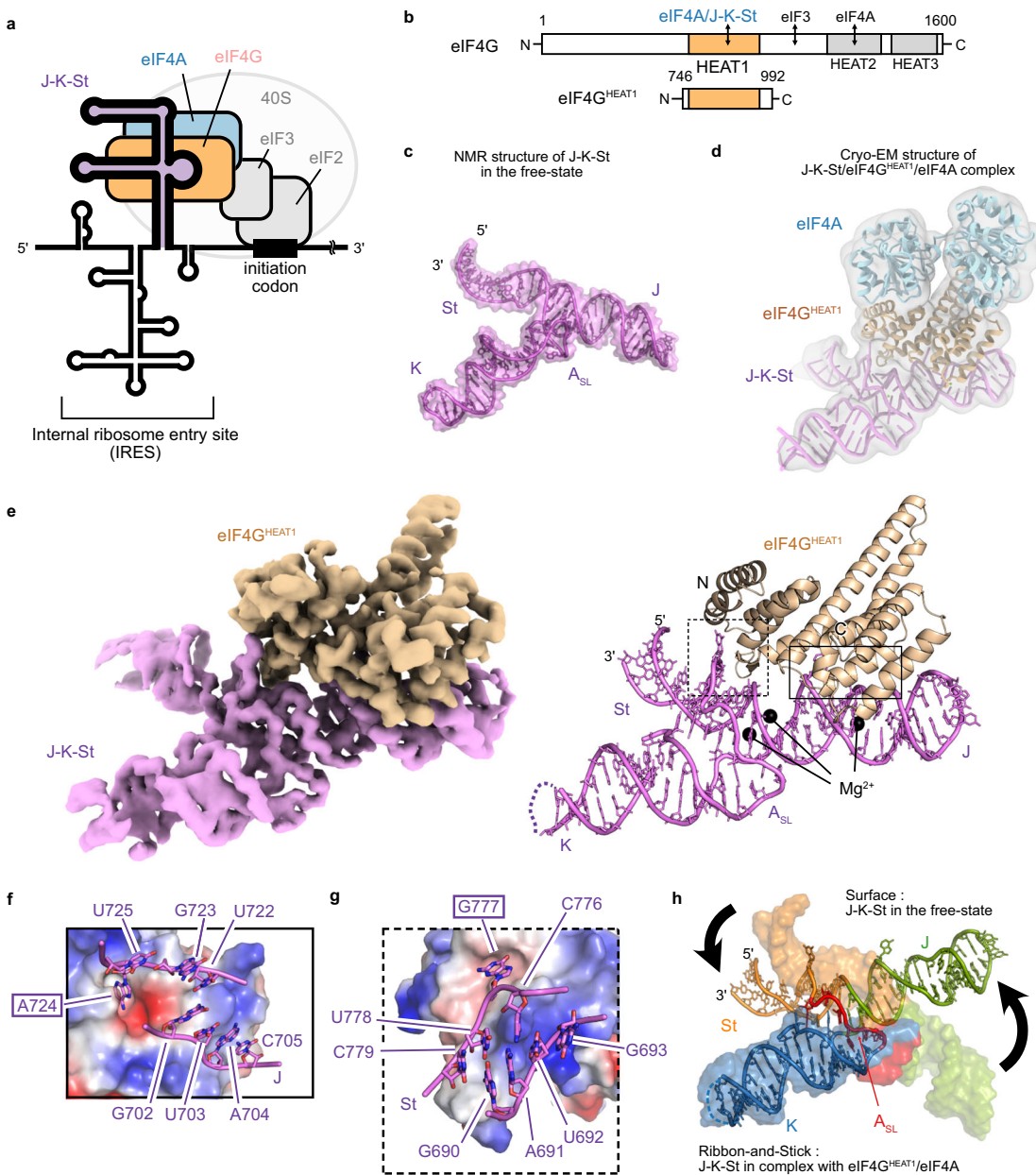

**Fig. 1 | Cryo-EM structure of the J-K-St/eIF4G^HEAT1/eIF4A complex. a** Schematic representation of the translation initiation complex formed by EMCV IRES. The J-K-St region directly interacts with the scaffolding protein eIF4G of the host cell, leading to the recruitments of eIF4A, eIF3, eIF2, and the ribosomal 40S subunit to the initiation codon. This figure is adapted from ref. 9. **b** Domain organization of the full-length eIF4G and the eIF4G^HEAT1 construct. **c** NMR structure of J-K-St in the free-state[9] (PDB ID: 2NBX). **d** Overall structure of the J-K-St/eIF4G^HEAT1/eIF4A ternary complex, overlaid with the lowpass-filtered cryo-EM density map. **e** Focused cryo-EM map (left) and the structure (right) of J-K-St/eIF4G^HEAT1. The J, K, and St domains of J-K-St are labeled along with the A$_{SL}$ domain at the junction. Mg$^{2+}$ ions are shown as black spheres. **f, g** Electrostatic potential representations of the surface of eIF4G^HEAT1 at the interface between the J domain (**f**) and the St domain (**g**), contoured from −5 kT/e (red) to +5 kT/e (blue). The residues of the J-K-St in contact with the eIF4G^HEAT1 are shown as sticks. The labels of the extruded residues, A724 on the J domain (**f**) and G777 on the St domain (**g**), are highlighted with boxes. **h** Comparison of the domain orientations. The structures of J-K-St in the free-state and in complex with eIF4G^HEAT1/eIF4A are shown as surface and ribbon-and-stick representations, respectively. The J, K, St, and A$_{SL}$ domains are colored green, blue, orange, and red, respectively. The structures are aligned with the K domain, and changes in the J and St domain orientations upon complex formations are shown as black arrows.

capture eIF4G^HEAT1. However, the structural mechanism by which the J-K-St region specifically recognizes eIF4G^HEAT1 has remained largely unknown. Typical RNA-binding proteins use interfaces where positively charged residues such as arginine or lysine, and/or aromatic residues such as tyrosine or phenylalanine, are clustered and aligned for the specific recognition of the target RNA molecules[16–18]. However, eIF4G^HEAT1 lacks clusters of such residues[12] (Supplementary Fig. 1), which is corroborated by the prediction of RNA binding residues in eIF4G^HEAT1 from the structure and/or sequence analyses identify no

RNA binding interface[19,20]. Furthermore, in order to hijack and exploit the host translational system, the viral IRES RNA should bind to eIF4G^HEAT1 without perturbing its innate function in translation; namely, recruiting eIF4A. To meet this condition, J-K-St should bind to eIF4G^HEAT1 without competing with eIF4A, which further restricts the possible binding sites on eIF4G^HEAT1.

Here, we analyze the three-dimensional structure of the J-K-St/eIF4G^HEAT1/eIF4A ternary complex by using cryo-electron microscopy (cryo-EM). The structure exhibits two distinct binding sites on the J-K-St

region. Each interacts with two separated negatively charged clefts and the surrounding positively charged patches on eIF4G[HEAT1], respectively, via the extruded nucleobases A724 from the J domain and G777 from the St domain. Importantly, in the structure analyzed in the free state, these nucleobases are not flipped out but stacked within the stems, while both are flipped out in the ternary complex structure. Furthermore, the relative orientation of the J and St domains is quite different from that in the free state. We conduct solution NMR analyses to investigate the dynamic properties of these eIF4G[HEAT1]-binding sites. The results demonstrate that the J and St domains utilize different dynamic modes: The bulges in the J domain are in conformational equilibria where A724 in the bulge is not largely perturbed, whereas the St domain is in conformational equilibria of the register shift in concert with the $A_{SL}$ domain, which leads to the release of G777 from the stacking interaction. In the conformational equilibrium, $A_{SL}$ assumes a conformation that brings the two binding sites to the optimal positions for binding. These dynamics are important for capturing eIF4G[HEAT1], since the suppression of these dynamics by mutations abrogated the interactions without affecting the eIF4G[HEAT1]-binding residues. Together, our study highlights the finding that the EMCV IRES has evolved to use the two eIF4G[HEAT1] binding domains, each tailored for the two small patches on the target protein. These two domains cannot bind to the target protein on their own because of the lack of sufficient intermolecular interactions, but can form a stable complex when they are consolidated and located at the optimal positions by the dynamic rearrangement of the $A_{SL}$ at the three-way junction. These mechanistic and functional insights illuminate the elegant strategy employed by the viral RNA to capture the host protein for efficient replication.

## Results

### Cryo-EM structure of the J-K-St/eIF4G[HEAT1]/eIF4A complex

In this study, we used J-K-St (G680-C787 in the EMCV genome RNA), the HEAT1 domain of human eIF4G (residues 746-992), designated as eIF4G[HEAT1], and the full-length human eIF4A. The isothermal titration calorimetry (ITC) experiment revealed that J-K-St binds to eIF4G[HEAT1] with $K_d$ values of 0.149 ± 0.012 and 2.07 ± 0.28 μM, in the presence of 2 mM $Mg^{2+}$ and the absence of $Mg^{2+}$, respectively (Supplementary

Fig. 2a, b), showing the $Mg^{2+}$ dependence of the interaction. We analyzed the structure of the J-K-St/eIF4G[HEAT1]/eIF4A complex in the presence of $Mg^{2+}$ at a 3.8 Å resolution by cryo-EM (Fig. 1d, and Supplementary Figs. 2c, 3, 4, 5). In the structure, the J and St domains directly contact eIF4G[HEAT1], while eIF4A does not interact with the J-K-St region (Fig. 1d). While the cryo-EM map in the region of J-K-St/ eIF4G[HEAT1] was well resolved, the region of eIF4A exhibited a weaker EM density, especially at the N-terminal lobe, reflecting the flexibility of the subunit (Supplementary Fig. 4a). Further local refinement with a mask on J-K-St/eIF4G[HEAT1] yielded the focused cryo-EM map of the region at a resolution of 3.7 Å (Fig. 1e and Supplementary Fig. 3). In the J domain, G702, U703, A704, C705, U722, G723, A724, and U725 interact with eIF4G[HEAT1] from the minor groove side (Fig. 1f and Supplementary Fig. 5b). All of these bases, except for A724, interact mainly with the positively charged patch on eIF4G[HEAT1] including side chain atoms of Arg 840, Lys841, Lys922, Lys925, and Arg954, whereas the nucleobase of A724 is extruded from the stem to deeply interact with the negatively charged cleft including sidechain atoms of Asp918. In the St domain, G690, A691, U692, G693, C776, G777, U778, and C779 also interact with eIF4G[HEAT1] from the minor groove side (Fig. 1g and Supplementary Fig. 5c). All of these residues, except for G777, interact mainly with the positively charged patch on the eIF4G[HEAT1] including sidechain atoms of Lys826 and Asn838, whereas the nucleobase of G777 is extruded from the stem to interact with another negatively charged cleft including the mainchain carboxy atoms of Thr784, Leu786, and Ala824, which differs from that contacted by the J domain. Based on the cryo-EM and previous NMR structure analyses, the angle between the J and St stem domains in the J-K-St/eIF4G[HEAT1]/eIF4A complex is notably altered compared to the angle formed in the free-state (Fig. 1h), whereas the structures of the eIF4G[HEAT1]/eIF4A themselves still resembled the free-state structures (Supplementary Fig. 5a).

Comparisons of the structures of J-K-St in complex with eIF4G[HEAT1] and eIF4A analyzed by cryo-EM (Fig. 2a) with that in the free-state analyzed by NMR[9] (Fig. 2b) revealed multiple changes in the secondary structures (Fig. 2, gray backgrounds). In the J domain in the free-state structure, the eIF4G[HEAT1] binding bases form bulges and are stacked within the bulge; i.e., A700 is not base-paired, A704 is base-paired with

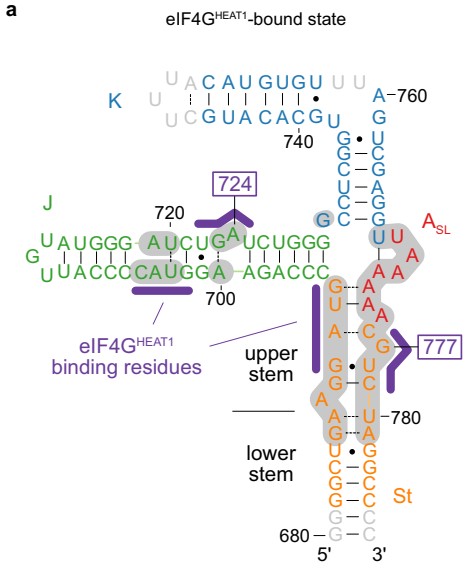

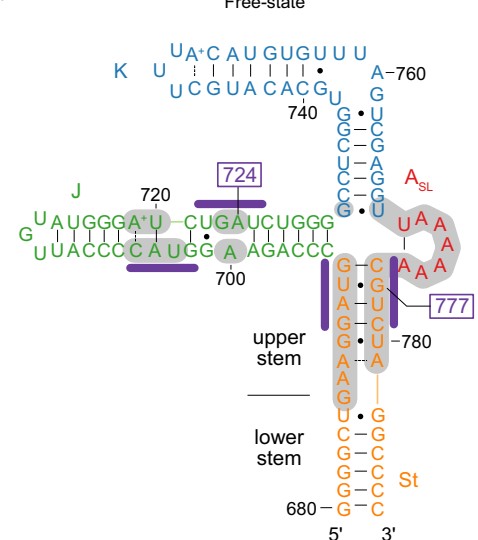

**Fig. 2 | Comparison of the secondary structures of J-K-St. a** The secondary structure of J-K-St in the complex with eIF4G[HEAT1] and eIF4A, determined by cryo-EM in this study. Bases that are not modeled are shown in light gray. **b** The secondary structure of the J-K-St in the free state, determined by NMR[9]. Lines between bases indicate the canonical Watson–Crick base pairs, dots indicate the wobble base pairs, and dashed lines are for non-canonical base pairs. The eIF4G[HEAT1]-binding residues are highlighted with purple lines. The J, K, St, and $A_{SL}$ domains are colored green, blue, orange, and red, respectively. The differences between these two structures are highlighted by gray backgrounds.

U720, C705 with A719, and A724 is not base-paired but stacked within the bulge. In the cryo-EM complex structure in the eIF4G[HEAT1]-bound state, these bases still form bulges with minor changes in the non-canonical base pairing patterns; i.e., A700 is base-paired with G723, U703 with U720, and A724 is flipped out. These base pair changes occur exclusively in the bulges that directly bind to eIF4G[HEAT1].

In stark contrast, structural rearrangements occur not only for the eIF4G-intearcting sites in the St domain, but also those not directly involved in the interaction with eIF4G[HEAT1] (Fig. 2). In our previous free-state structure, the A[SL] domain forms a short stem-loop, and the upper stem of the St domain forms five consecutive Watson-Crick or wobble base pairs stacked with an Ade-Ade non-canonical base pair; i.e., A688-A781, G689-U780, G690-C779, A691-U778, U692-G777, and G693-C776. In the J-K-St structure in complex with eIF4G[HEAT1]/eIF4A, the A[SL] domain does not form the short stem-loop, and the upper stem of the St domain, which includes the eIF4G[HEAT1]-binding site, exhibits large base-pairing rearrangements within the A[SL] domain bases; i.e., G686-A781, A687-U780, G689-C779, G690-U778, A691-C776, U692-A774, and G683-A773. G777 is flipped out to interact with the cleft of eIF4G[HEAT1]. Together, the comparison illustrates that the interaction of the J-K-St region with eIF4G[HEAT1] involves several structural rearrangements: The J domain is locally perturbed within the bulge, whereas the upper stem of the St domain is largely rearranged with the change of the A[SL] domain, although both accompany the flipping-out of the key extruding residues, A724 and G777. Importantly, both of these

domains are required for the binding to eIF4G[HEAT1], because neither isolated domain could bind to eIF4G[HEAT1] (Supplementary Fig. 2d, e)[9].

## Dynamics of the two eIF4G[HEAT1]-binding sites

To gain insights into the mechanism underlying the capture of the host target protein by the viral RNA, we used solution NMR to characterize the dynamic properties of the eIF4G[HEAT1]-binding sites of J-K-St. To this end, we employed [u-$^2$H, Ade-{$^1$H2, $^1$H8, $^{13}$C8}, Gua-{$^1$H8, $^{13}$C8}] isotope labeling of RNA samples, and used it in combination with $^1$H–$^{13}$C aromatic transverse relaxation optimized spectroscopy (TROSY)[21] (Supplementary Fig. 6), which enables observation of NMR signals that reflect on the structure and dynamics of RNA in high sensitivity. The $^1$H8–$^{13}$C8 aromatic TROSY signals from the isolated J domain overlapped with those from the full-length J-K-St, corroborating that the structure of the isolated J domain is retained as in the full-length J-K-St[9] (Supplementary Fig. 7a, b). The $R_{ex}$ values obtained from the $^{13}$C single quantum (SQ) relaxation dispersion experiment, which is the chemical exchange contribution to the transverse relaxation rate $R_2$, reflect the degree of the structural dynamics in the μs-ms timescale that is critical for many RNAs to function[22]. The $R_{ex}$ analyses of the isolated J domain were then conducted in a base-specific manner, in the absence of eIF4G[HEAT1] at 30 °C (Fig. 3a). As a result, G701, A704, G718, and G723 exhibited $R_{ex}$ values larger than 10 s$^{-1}$, whereas G702 and A719 could not be quantitatively analyzed because their larger degrees of dynamics hampered the quantification of the signal intensities, indicating that the bases are in conformational exchange processes in a

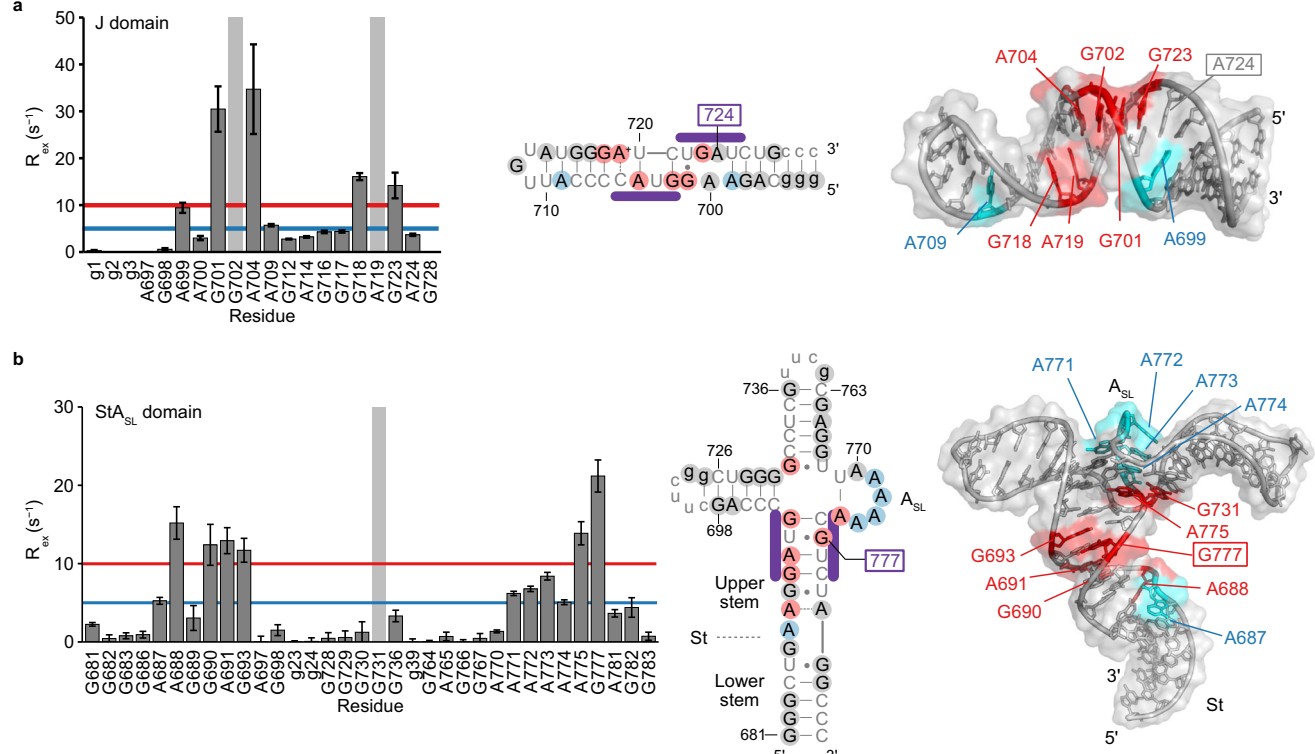

**Fig. 3 | Mapping of the μs-ms dynamics of the eIF4G-interacting domains. a** Left: Residue-specific $R_{ex}$ values of the J domain. The residues that could not be analyzed quantitatively, due to extensive conformational exchanges in the μs-ms time scale, are shown as gray backgrounds. Middle and right: Mapping of the $R_{ex}$ values in the secondary structure (middle) and tertiary structure of the previously reported J domain (right, PDB ID: 2NBY)[9]. The Ade/Gua residues that exhibited $R_{ex}$ values larger than 10 s$^{-1}$ are colored red, whereas those between 5 and 10 s$^{-1}$ are cyan. Source data are provided as a Source Data file. **b** Left: Residue-specific $R_{ex}$ values of the StA[SL] domain. Residues that could not be analyzed quantitatively, due to

extensive conformational exchanges in the μs-ms time scale, are shown with a gray background. Middle and right: Mapping of the $R_{ex}$ values in the secondary structure (middle) and tertiary structure model (right), which is derived from the previously reported structure of the ΔJΔK domain[9] (PDB ID: 2NC1) by substituting the GAAA tetraloops with the UUCG tetraloops. The Ade/Gua residues that exhibited $R_{ex}$ values larger than 10 s$^{-1}$ are colored red, whereas those between 5 and 10 s$^{-1}$ are cyan. Source data are provided as a Source Data file. Error bars indicate experimental errors derived from the signal-to-noise ratio of each correlation, as written in Methods.

μs-ms timescale. These bases are within, or adjacent to, the bulges on the J domain that interact with eIF4G[HEAT1] (Fig. 3a). Importantly, A724, whose nucleobase is extruded to the cleft of eIF4G[HEAT1] in the complex structure, exhibited a relatively small $R_{ex}$ value of $3.7 \pm 0.2$ s$^{-1}$. Since this base is stacked in the bulge with the neighboring bases, G723 and U725, in the free-state structure (Fig. 2b), these results indicated that A724 essentially remains stacked throughout the trajectory of the chemical exchange processes. Together, the $R_{ex}$ analyses of the isolated J domain demonstrated that the flipping-out of the A724 nucleobase does not occur by itself and thus can only be initiated after the J domain docks with eIF4G[HEAT1].

To analyze the conformational dynamics of the eIF4G[HEAT1]-interacting site on the St domain, we used a construct designated as the StA$_{SL}$ domain (Fig. 3b), which includes the isolated St and A$_{SL}$ domains, because the structural rearrangement of the St domain upon binding to eIF4G[HEAT1] involves changes in the A$_{SL}$ domain (Fig. 2). The $^1$H8–$^{13}$C8 aromatic TROSY signals from the isolated StA$_{SL}$ domain sample overlapped with those from the full-length J-K-St, confirming that the structure of the isolated StA$_{SL}$ domain is retained as in the full-length J-K-St[9] (Supplementary Fig. 7a, c). The $R_{ex}$ values of the StA$_{SL}$ domain were obtained at 30 °C and plotted (Fig. 3b). The nucleobases of A688, G690, A691, G693, A775, and G777 exhibited $R_{ex}$ values larger than 10 s$^{-1}$, whereas the nucleobases of A687, A771, A772, A773, and A774 had moderate $R_{ex}$ values between 5 and 10 s$^{-1}$, indicating that the bases are in conformational exchange processes in a μs-ms timescale. G731 could not be quantitatively analyzed because the large degree of dynamics hampered the quantification of the signal intensities. Intriguingly, G777, whose nucleobase is extruded into the cleft on eIF4G[HEAT1] in the complex structure, showed the largest $R_{ex}$ value of $21.2 \pm 2.0$ s$^{-1}$, indicating that the base undergoes chemical exchange processes in the μs-ms timescale.

## Characterization of the excited states of the StA$_{SL}$ domain

To further characterize the dynamics observed for the StA$_{SL}$ domain, we investigated the temperature dependence of the $^1$H8-$^{13}$C8 aromatic TROSY signals, which reflect the change in conformational equilibrium upon changes in temperature. All of the bases that exhibited $R_{ex}$ values larger than 5 s$^{-1}$ (Fig. 3b), except for G777, showed linear chemical shift changes upon temperature reduction from 35 °C to 10 °C, whereas G777 exhibited non-linear chemical shift changes (Fig. 4a). These results indicated that most of the bases (except for G777) exist in an equilibrium between at least two states, whereas G777 does so among at least three, where the populations of each state are perturbed upon temperature changes. We then analyzed the $^{13}$C SQ relaxation dispersion profiles of the StA$_{SL}$ domain, which are the NMR pulse frequency ($\nu_{CPMG}$) dependence of the effective $R_2$ including the chemical exchange contribution, $R_{2,eff}$. These profiles provide insights into exchange processes in terms of the exchange rate, $k_{ex}$, the relative populations of the exchanging states, $p$, and the difference in chemical shift, $\Delta\omega$[23]. The fast two-state exchange model[24] with a single exchange rate ($18,500 \pm 360$ s$^{-1}$) explained the $^{13}$C SQ relaxation dispersion profiles of A687, A688, G690, A691, G693, A771, A772, A773, A774, and A775 simultaneously, demonstrating that the chemical exchanges of these residues occur in a cooperative manner (Fig. 4b, top, and Supplementary Fig. 8). Given that G777 is in the upper stem of the St domain, where A688, G690, A691, and G693 are all located, G777 should also experience this chemical exchange process. By using the exchange parameters obtained from the global fit analyses mentioned above, the $^{13}$C SQ relaxation dispersion profiles of G777 were fit numerically to the three-state exchange model, where G777 transitions between the ground state (GS), and two distinct excited states 1 and 2 (ES1 and ES2) that are not distinguished as chemical shift differences of the other bases (Fig. 4b, bottom). As a result, we obtained a set of kinetic parameters and the $^{13}$C chemical shifts of G777 C8 in each state (Fig. 4c, Supplementary Figs. 9, 10). The populations of GS, ES1, and

ES2 are $80.4 \pm 2.9$, $17.9 \pm 2.9$, and $1.7 \pm 0.1\%$, respectively, whereas the chemical shift values of the C8 atom of G777 in these states are 138.2, 136.6, and 138.5 ppm, respectively, with exchange rates of $18,600 \pm 120$, $410 \pm 370$, and $230 \pm 70$ s$^{-1}$ between GS and ES1, ES1 and ES2, and GS and ES2, respectively. It should be noted here that the chemical shifts of A687, A688, G690, A691, G693, A771, A772, A773, A774, and A775, which were analyzed with the fast two-state exchange model, should be indistinguishable between ES1 and ES2 for each base, apparently reducing the three-state exchange model (Fig. 4c) into the fast two-state exchange model. Thus, the St upper stem and the A$_{SL}$ domain are in the three-state chemical exchange among GS, ES1, and ES2, in which the chemical shifts of G777 are perturbed between ES1 and ES2.

The $^{13}$C8 chemical shifts of purine bases (adenosine and guanosine) reportedly correlate with the degree of stacking with the preceding base on the 5' side[25-27]; in this case, C776. Indeed, a distance-based calculation[28] estimated the chemical shift of the $^{13}$C8 of a guanosine in the ideal A-form helix structure, where the preceding cytosine is fully stacked onto the guanosine, as 137.0 ppm, that of G777 in the NMR structure in the free-state, where C776 is only partially stacked, as 138.1 ppm, and that of G777 from the cryo-EM structure in the complex, where G777 is flipped out without any intramolecular stacking interactions, as 140.6 ppm (Fig. 4c). Since the $^{13}$C8 chemical shifts of G777 in GS, ES1, and ES2 are 138.2, 136.6, and 138.5 ppm, respectively, these values illustrate that G777 is more stacked in ES1, and less stacked in ES2, than it is in GS (Fig. 4c). Hereafter, we refer to these conformations in ES1, ES2, and GS as the excited conformation (EC) 1, EC2, and ground conformation (GC), respectively. The GC, which is the most populated conformation (80.4%) at 30 °C, corresponds to the NMR structure[9] in which G777 is partially stacked, as corroborated by the chemical shift correspondence with G777 in GC and the value calculated from the NMR structure. The lowly populated excited conformations were not detected in previous NMR structural analyses that focused on the most populated ground conformation. In this study, these excited conformations were observed, by conducting the NMR relaxation dispersion analyses to investigate the dynamic properties of J-K-St.

To clarify the mechanism underlying the changes in the stacking interactions at G777, we sought to further characterize the conformational equilibrium observed for the StA$_{SL}$ domain, by stabilizing the excited conformations. First, the U778C variant was designed. U778 is base-paired with A691 in the NMR structure in the free-state, whereas it forms the G·U wobble base pair with G690 in the complex with eIF4G[HEAT1] (Fig. 2). In the $^1$H-$^1$H NOESY spectrum at 10 °C, where cross peaks are observed when two imino protons ($^1$H$_{imino}$) are in spatial proximity (<5 Å), cross peaks between the imino protons of U780, G689, and G690 were only observed for the U778C variant (Fig. 5a). These results indicated that the A687·U780, G689·C779 and G690·C778 base pairs are stabilized by a single U to C substitution at position 778, possibly due to the destabilizing A691·U778 Watson-Crick (WC) base pair, and the stabilizing G690·U778 wobble base pair to the G·C WC base pair (Fig. 5b, Supplementary Fig. 11). This register shift changes the base pairing pattern of the St upper stem to that observed in the eIF4G[HEAT1]-bound state (Figs. 2, 5b). We then compared the $^1$H8-$^{13}$C8 aromatic TROSY signals from G690, which is base paired with C778 in the register-shifted St upper stem. For the wild-type StA$_{SL}$ domain, the chemical shift values of G690 shifted linearly as the temperature was decreased from 35 °C to 10 °C, indicating the population shift of the conformational equilibrium (Fig. 5c). For the U778C variant of the StA$_{SL}$ domain, the signal at 35 °C was observed on the line extrapolated from the temperature-dependent chemical shift changes observed for the wild-type towards the lower temperature, and further shifted on the line as the temperature was decreased to 10 °C, where the $^1$H$_{imino}$-$^1$H$_{imino}$ NOESY spectrum (Fig. 5a) was acquired. This indicates that the excited conformations (EC1 and EC2) in the

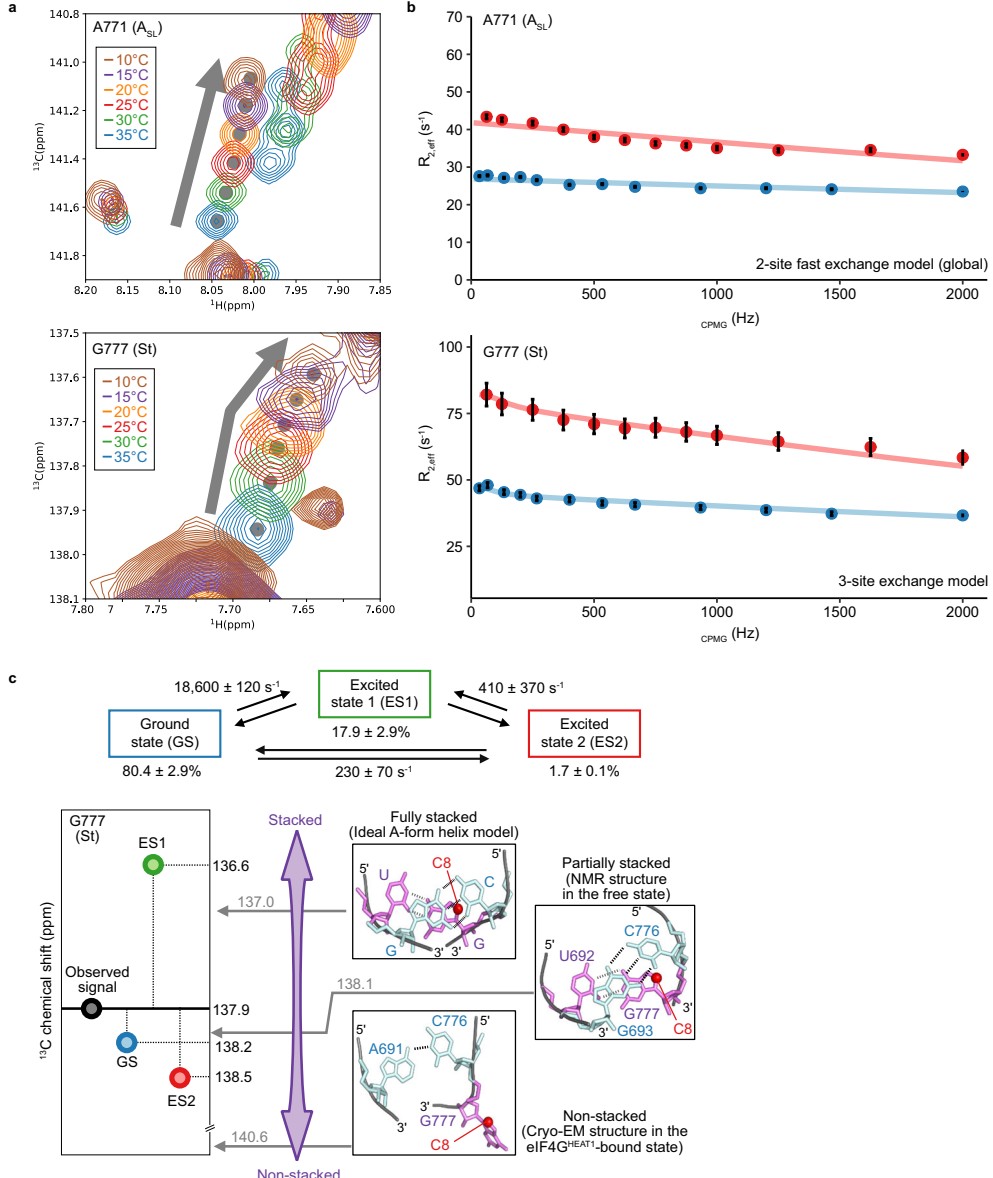

**Fig. 4 | Relaxation dispersion analyses of the StA_SL domain. a** Temperature dependencies of the $^1$H8-$^{13}$C8 aromatic TROSY signals of A771 (A_SL domain) and G777 (St domain). Spectra acquired at 10, 15, 20, 25, 30, and 35 °C at the $^1$H frequency of 1.0 GHz are shown in brown, violet, orange, red, green, and blue, respectively. **b** Relaxation dispersion curve fitting. $R_{2,eff}$ values obtained at 30 °C and at the $^{13}$C frequencies of 250 MHz and 150 MHz ($^1$H frequencies of 1.0 GHz and 600 MHz) are shown by red and blue points, respectively. The relaxation dispersion profile of G777 was fit with the 3-site exchange model, in which it possesses two different chemical shifts in the excited states, using kinetic parameters obtained from the global fitting with the 2-site exchange model (see also Supplementary Fig. 8). The fitted curves are shown as red and blue lines for the $^{13}$C frequencies of 250 MHz and 150 MHz, respectively. Error bars indicate experimental errors

derived from the signal-to-noise ratio of each correlation, as written in Methods. Source data are provided as a Source Data file. **c** Top: The exchange parameters obtained from the relaxation dispersion profile at 30 °C. Bottom: The chemical shifts of $^{13}$C8 of G777. Among the observed chemical shifts, 137.9 ppm is the kinetic average of the ground state (138.2 ppm), excited state 1 (136.6 ppm), and excited state 2 (138.5 ppm). Predicted chemical shift values of $^{13}$C8 of G777 from the ideal A-form helix structure (137.0 ppm), the NMR structure of J-K-St in the free-state (138.1 ppm), and the cryo-EM structure of J-K-St in complex with eIF4G$^{HEAT1}$ (140.6 ppm) are shown in gray, indicating that the $^{13}$C8 chemical shifts mainly reflect the stacking with the preceding residue. Note that the predicted chemical shift values are corrected for the TROSY signals by adding 0.5 ppm.

conformational equilibrium observed for the St upper stem coincide the register-shifted conformation observed for the U778 variant, and the population of excited conformations increases by lowering the temperature. Since G777 does not form a canonical, stable base-pair in this register-shifted configuration, but forms a G-U wobble base pair in the GC (Fig. 5b), the hydrogen bonding interactions with the other strand of the stem that hold the nucleobase of G777 inward are weakened, allowing the nucleobase to flip out in EC2.

We subsequently performed the $^{13}$C SQ relaxation dispersion experiment on the StA_SL U778C variant (Supplementary Fig. 12). The

signal from G777 C8 was broadened beyond detection due to the introduction of the U778C mutation possibly reflecting the shift in the conformational equilibrium, which makes it unanalyzable by the relaxation dispersion experiment. However, it was demonstrated that relaxation dispersion curves of G687, G690, A771, and A772, which exhibited linear profiles in the wild-type (Fig. 4b and Supplementary Fig. 8), displayed non-linear profiles in the U778C variant (Supplementary Fig. 12c). These curves could not be fit by either the fast two-state exchange equation[24] or the general two-state exchange equation[29], suggesting that the modulation in the conformational

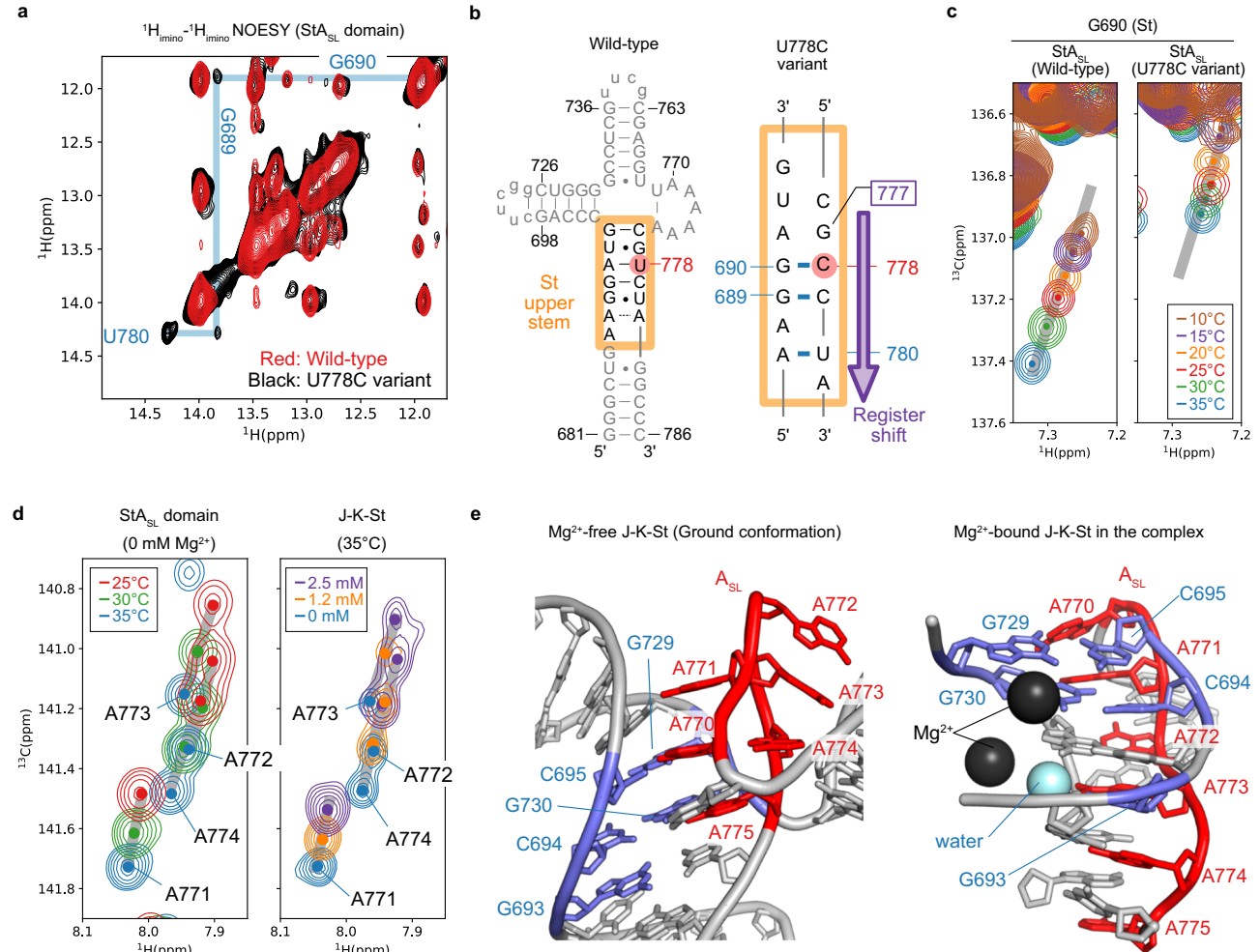

**Fig. 5 | Structural characterization of the excited state of the StA$_{SL}$ domain.**
**a** $^1$H$_{imino}$-$^1$H$_{imino}$ NOESY spectra of the StA$_{SL}$ domain (red) and its U778C variant (black). NOE signals observed only for the U778C variant are shown in blue with their assignments. **b** Secondary structure of the St upper stem, as determined by the NOESY experiments. **c** Temperature dependence of the $^1$H8-$^{13}$C8 TROSY signals of the StA$_{SL}$ domain (left) and its U778C variant (right). Spectra acquired at 10, 15, 20, 25, 30, and 35 °C are shown in brown, violet, orange, red, green, and blue, respectively. In the spectra of the U778C variant, the signal was observed on the extrapolated line (gray) of the parental construct, indicating that the equilibrium was shifted so that the conformations that are more populated at the lower temperature are more stabilized by the U778C mutation. **d** Comparison of the

temperature-dependent chemical shift perturbations of the signals from the A$_{SL}$ domain with the Mg$^{2+}$-dependent chemical shift perturbations. (Left) Temperature-dependent chemical shift perturbations of the signals from A$_{SL}$ in the StA$_{SL}$ domain. Spectra acquired in the absence of Mg$^{2+}$ at 25, 30, and 35 °C are shown in red, green, and blue, respectively. (Right) Mg$^{2+}$-dependent chemical shift perturbations of the A$_{SL}$ domain in J-K-St at 35 °C. Spectra acquired with Mg$^{2+}$ concentrations of 0, 1.2, and 2.5 mM are shown in blue, orange, and purple, respectively. **e** Structural comparison of the A$_{SL}$ domain. (Left) Mg$^{2+}$-unbound A$_{SL}$ structure in the free J-K-St structure (ground state). (Right) Mg$^{2+}$-bound A$_{SL}$ structure in the Mg$^{2+}$-bound state in the J-K-St/eIF4G$^{HEAT1}$/eIF4A complex. The bases in the A$_{SL}$ domain are shown in red, and those involved in the interaction with Mg$^{2+}$ are blue.

equilibrium altered the chemical exchange among GS, ES1 and ES2 evident for these bases. Global fitting of the relaxation dispersion curves of these bases with the three-state exchange model, assuming exchange rates equivalent to the wild-type, revealed that the populations of GS, ES1, and ES2 are 36.9 ± 3.1, 58.8 ± 2.9, and 4.3 ± 0.1%, respectively, for this variant (Supplementary Fig. 12d), which is markedly different from the populations of 80.4 ± 2.9, 17.9 ± 2.9, and 1.7 ± 0.1% observed for the wild-type (Fig. 4c). These results quantitatively indicate that the U778C mutation decreases the population of GS while increasing the populations of ES1 and ES2, and support the notion that the upper stem of the St domain is register-shifted in ES1 and ES2.

To characterize the excited conformation of the A$_{SL}$ domain, we compared the temperature-dependent chemical shift changes with the Mg$^{2+}$-dependent chemical shift changes of the bases from the A$_{SL}$ domain (Fig. 5d). Interestingly, upon decreasing the temperature from 35 °C to 25 °C in the absence of Mg$^{2+}$, the chemical shift values of A771,

A772, A773, and A774 in the StA$_{SL}$ domain exhibited virtually identical changes to those observed for these bases in J-K-St upon changing the Mg$^{2+}$ concentrations from 0 to 2.5 mM, at 35 °C (Fig. 5d). Therefore, the chemical shift changes of the bases in the A$_{SL}$ domain upon decreasing the temperature are identical to those observed upon titrating Mg$^{2+}$. Since decreasing the temperature increases the population of the excited conformation (Fig. 5c), the excited conformation of the A$_{SL}$ domain should be similar to that of the A$_{SL}$ domain in the Mg$^{2+}$-bound state. Considering that two Mg$^{2+}$ ions bind to base triples that include the A$_{SL}$ domain bases in the cryo-EM structure of the complex (Fig. 5e), these results suggest that the excited conformation of the A$_{SL}$ domain without Mg$^{2+}$ is similar to the conformation of the A$_{SL}$ domain bound to Mg$^{2+}$ in the complex structure, preforming the Mg$^{2+}$ binding sites, which are further stabilized upon Mg$^{2+}$ binding.

Together, these results indicate that the ECs of the St upper stem and the A$_{SL}$ domain have structural characteristics similar to those of the eIF4G$^{HEAT1}$-bound state, with the important observation that the

extruded nucleobase G777 is released from the stacking interaction within the stem. Since the St upper stem is the region that interacts directly with eIF4G[HEAT1], and G777 is extruded to interact with the negatively charged cleft thereof, these excited conformations, especially EC2 in which G777 is not stacked on the other bases, are favorably engaged in the interactions at the early stage of the complex formation process. In addition, the excited conformations of the $A_{SL}$ domain at the three-way junction that resemble the eIF4G[HEAT1]-bound state should rearrange the domain orientations to those in the complex structure (Figs. 1h, 5e), bringing the two eIF4G[HEAT1]-binding sites to optimal positions for simultaneous binding.

**Functional relevance of the dynamics in the J and St$A_{SL}$ domains**
Next, we conducted ITC experiments to investigate whether the conformational dynamics observed for the two eIF4G[HEAT1] binding sites on the J and St$A_{SL}$ domains are critical for the interaction with eIF4G[HEAT1] (Supplementary Fig. 13). Given that the U778C mutation increases the populations of the excited conformations, which possess the structural characteristics of J-K-St as in the complex structure, it might be expected that the J-K-St U778C variant would have a similar, or even stronger, affinity for eIF4G[HEAT1] compared to the wild-type. However, the J-K-St U778C variant exhibited a decreased affinity for eIF4G[HEAT1], with a $K_d$ value larger than 10 µM, compared to 149 ± 12 nM for the wild-type (Supplementary Figs. 2a, 13b). This is likely because U778 is on the interface with eIF4G[HEAT1] in the complex structure (Figs. 1g, 2a, and Supplementary Fig. 14), and mutating it might result in decreased free energy upon interaction with eIF4G[HEAT1], reducing the affinity for eIF4G[HEAT1]. We then designed the A700C and U780C variants, to suppress the conformational dynamics in the J and St$A_{SL}$ domains observed in the NMR analyses, respectively, without mutating the bases that directly interact with eIF4G[HEAT1] (Supplementary Fig. 15). For the J domain, changing A700 to cytidine would allow for the canonical Watson-Crick base pair formation with G723, which would suppress the plasticity of the J domain bulge G723-A724 to allow for the conformational change in the complex formation process (Supplementary Fig. 15a, b). For the St$A_{SL}$ domain, changing U780 to a cytidine would stabilize the base pairing with G689, as the canonical Watson-Crick G-C base pair is more stable than the G-U wobble base pair, which would suppress the register shift toward the excited conformations required for the conformational change in the complex formation process (Supplementary Fig. 15d, e). NMR $R_{ex}$ analyses of these isolated domain variants revealed that the dynamics in the µs-ms timescale are suppressed (Supplementary Fig. 15c, f), while the secondary structures of these isolated domains are not altered by the mutation (Supplementary Fig. 15a, d). The ITC experiments revealed that A700C and U780C variants of J-K-St possess reduced affinities for eIF4G[HEAT1] (Supplementary Fig. 13a, b), indicating that the suppression of the conformational dynamics to stabilize the ground conformation results in decreased affinities for eIF4G[HEAT1].

Finally, we sought to investigate the correlation between the dynamics and the biological function of IRES. To this end, we utilized the human cell-free translation system[30], where the EMCV IRES is used to initiate the translation of a reporter protein, β-galactosidase, by interacting with the translation initiation factors in the human cell (Supplementary Fig. 13c). As a result, it was shown that introducing A700C, U780C, or U778C mutations in the full length EMCV IRES decreased the protein expression levels to 86.9 ± 3.2, 89.5 ± 3.4, and 72.3 ± 3.4%, respectively, demonstrating that the conformational dynamics observed in the NMR analyses are related to the biological function of IRES.

## Discussion
In this study, we visualized the three-dimensional structure of the J-K-St/eIF4G[HEAT1]/eIF4A complex, by cryo-EM single particle analysis (Fig. 1

and Supplementary Figs. 3 and 4). This structure not only corroborated the previous biochemical reports on the interaction sites on the J-K-St and eIF4G[HEAT1] (Supplementary Fig. 16)[10–12], but also revealed key features on the residue-specific interactions and conformational changes of J-K-St, upon interaction with eIF4G[HEAT1]. The J-K-St region binds to eIF4G[HEAT1] via two separate sites on the J and St domains. We previously presented a low-resolution SAXS structure of the J-K-St and eIF4G[HEAT1] complex, and proposed that the K and St domains are the eIF4G[HEAT1] binding domains, assuming that the J-K-St does not undergo structural changes upon binding to eIF4G[HEAT1] [9]. Here, in the cryo-EM structure at 3.7 Å resolution, we clearly showed that J-K-St binds to eIF4G[HEAT1] via the J and St domains by adjusting their relative angles to adhere to the surface of eIF4G[HEAT1] (Fig. 1h), but not significantly changing the overall structure of eIF4G[HEAT1] (Supplementary Fig. 5a). This is an important requirement to subdue the host translational machinery, as inducing structural changes that interfere with protein-protein interactions, in this case with eIF4A, would impact the efficient translation of the viral proteins for replication. In order to accomplish the specific interaction of the target protein without interference from the interactions with eIF4A, J-K-St should contact the surface of eIF4G[HEAT1], which has scattered positively and negatively charged patches and only a few exposed aromatic residues (Supplementary Fig. 1). This is a totally different situation from the RNA–protein interactions involving canonical RNA-binding proteins[16–18], which use basic and aromatic residues to form interfaces that are ideal for the electrostatic and π–π stacking interactions with RNA, a negatively charged and highly aromatic molecule (Supplementary Fig. 1).

EMCV IRES enables this interaction by using the two stem domains, J and St, each tailored for the two small patches on the target protein. Importantly, neither of these domains can bind to eIF4G[HEAT1] being isolated (Supplementary Fig. 2d, e), possibly because the energy gain obtained upon binding one domain is not sufficient to form a stable complex by the limited interactions with the small patch. The J domain interacts with a small positively charged surface with the main chain atoms, and a negatively charged cleft with the extruded nucleobase, A724 (Fig. 1f). Our $^{13}C$ SQ relaxation dispersion analyses revealed the conformational dynamics for the J domain bulges, which are formed by the eIF4G[HEAT1] binding residues (Fig. 3a). This could be important to acquire plasticity to fit the charge distribution on the surface of the target. We also found that A724 is minimally perturbed by the conformational equilibrium in the absence of eIF4G[HEAT1], suggesting that the flipping-out of the nucleobase occurs only after the interactions are formed with the other eIF4G[HEAT1]-binding bases, which would lead A724 to the optimal position to interact with the cleft. Meanwhile, the St domain docks onto another positively charged patch on eIF4G[HEAT1], also mainly via the mainchain atoms. In this case, the extruding nucleobase, G777, interacts with a relatively shallow, negatively charged cleft (Fig. 1g). Since the cavity on the St domain formed by extrusion of G777 was filled by a lysine residue, Lys826, of eIF4G[HEAT1] (Supplementary Figs. 5c, 14), the flipping-out of G777 is important not only for its direct interaction with eIF4G[HEAT1], but also for the formation of the binding pocket for this basic residue. The $^{13}C$ SQ relaxation dispersion experiments illustrated that the St upper stem exists in a conformational equilibrium. In the excited conformations EC1 and EC2, the St upper stem is register-shifted, where G777 is more and less base-stacked, respectively. The register shift and weaker stacking of G777 support the notion that EC2, which exists in a low population of 1.7% in the absence of eIF4G[HEAT1] (Fig. 4c), possesses structural features of the bound-state conformation, and is possibly involved in the initial step of the complex formation process.

The two eIF4G[HEAT1]-binding domains, J and St, are consolidated by the three-way junction formed by the $A_{SL}$ domain. Our $^{13}C$ SQ relaxation dispersion experiments also revealed that the $A_{SL}$ domain is in a

conformational equilibrium coupled with the St upper stem (Fig. 4b, c, and Supplementary Figs. 8–10). A comparison of the chemical shifts revealed that the excited conformations of the $A_{SL}$ domain in the absence of $Mg^{2+}$ and eIF4G$^{HEAT1}$ is similar to the $Mg^{2+}$-bound conformation in the absence of eIF4G$^{HEAT1}$ (Fig. 5d, e). Since $Mg^{2+}$ is bound to the $A_{SL}$ domain region in the J-K-St/eIF4G$^{HEAT1}$/eIF4A complex, it is possible that this $A_{SL}$ structure in the complex represents the excited conformation of the $A_{SL}$ domain in the absence of $Mg^{2+}$ and eIF4G$^{HEAT1}$. This suggests that the excited conformations of the $A_{SL}$ domain at the three-way junction bring the J and St domains to appropriate positions for simultaneous docking to the two patches on the eIF4G$^{HEAT1}$ domain, by the intramolecular interactions with the other domains (Figs. 1h, 5e).

Altogether, the J-K-St region uses two interaction sites to capture its target protein, eIF4G, primarily by changing the angle between the J and St stem domains (Figs. 1h, 6). These two sites interact collectively on the target protein, and the binding cannot be accomplished by a single domain (Supplementary Fig. 2a, d, e). The dynamic regulations may be tailored for each binding site on eIF4G$^{HEAT1}$. Importantly, the relative orientation of these two sites should also be precisely organized, so they can interact simultaneously with their respective binding sites. Thus, the structure and dynamics of the $A_{SL}$ domain at the three-way junction, which does not interact directly with eIF4G$^{HEAT1}$, should also play a fundamental and important role in the interaction by adjusting the relative orientation and distance of these two binding sites. We envision that these findings on the function and mechanism of the viral IRES will facilitate the design of novel therapeutics that prevent viral replication by inhibiting IRES-host protein interactions.

## Methods

### Plasmid construction

The plasmids for the in vitro RNA transcription reactions were constructed by inserting PCR-amplified DNA sequences containing the T7 promoter (TAATACGACTCACTATA), followed by the sequence of

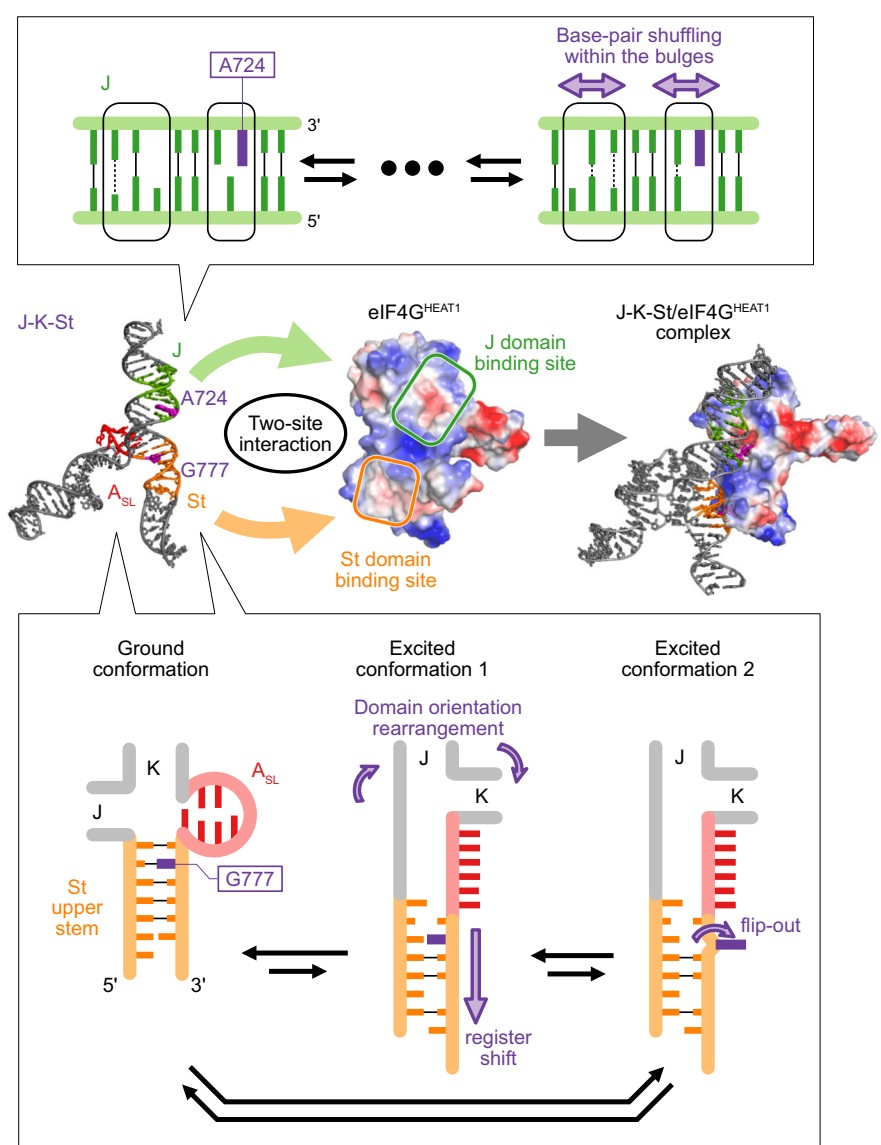

**Fig. 6 | Two-site interaction model.** In J-K-St, J domain is in the conformational equilibrium among multiple conformations where the base pairing patterns in the bulge region are different. The stem regions in the J domain are not perturbed in the equilibrium (top). The St and $A_{SL}$ domains are in the concerted conformational equilibrium among the ground conformation, the excited conformation 1, and the excited conformation 2 (bottom). By the register shift of the St upper stem and the conformational change in the $A_{SL}$ domain, the rearrangement of the domain orientation of the J, K, and St domains occur in the excited conformations, so that J-K-St assumes optimal structure for the simultaneous two-site interaction by the J and St domains, required for the binding of J-K-St to eIF4G$^{HEAT1}$.

interest, into pUC19 (Takara 3219) between the BamHI and EcoRI restriction sites, by using InFusion reactions (Takara-Clontech 639648). The RNA sequences were: J-K-St, 5′-GGGGCUGAAGGAUGCC CAGAAGGUACCCCAUUGUAUGGGAUCUGAUCUGG GGCCUCGGUGC ACAUGCUUUACAUGUGUUUAGUCGAGGUUAAAAAACGUCUAGGCCC C-3′; J domain, 5′-gggCAGAAGGUACCCCAUUGUAUGGGAUCUGAUCU Gccc-3′; StA$_{SL}$ domain, 5′-GGGCUGAAGGAUGCCCAGcuucggCUGGGG CCUCGuucgCGAGGUUAAAAAA CGUCUAGGCCC-3′; and St-extended J-K-St, 5′- gggaaau GGGGCUGAAGGAUGCCCAGAAGGUACCCCAUUGU AUGGGAUCUGAUCUGG GGCCUCGGUGCACAUGCUUUACAUGUGUU UAGUCGAGGUUAAAAAACGUCUAGGCCCCauuuccc −3′, where the nonnative nucleotides are shown in lowercase.

The plasmids for the expression of the enzymes used for the preparation of isotopically labeled rNTPs were amplified from the *Escherichia coli* DH5α genome, and inserted into a pET28a(+) (Novagen 69864) based vector, where silent mutations to increase the expression levels were incorporated into the 5′ untranslated region[31] and a MQLGHNHNHNHNHNHN tag was added at the N-terminus of the translated region, by using inFusion reactions. The cloned enzymes were ribose-phosphate pyrophosphokinase (PRPPS, UniProt P0A717), ribokinase (RK, UniProt P0A9J6), adenine phosphoribosyltransferase (APRT, UniProt P69503), guanylate kinase (GMK, UniProt P60546), xanthine-guanine phosphoribosyl transferase (XGPRT, UniProt P0A9M5), uracil phosphoribosyltransferase (UPRT, UniProt P0A8F0), uridylate kinase (UMPK, UniProt P0A7E9), and CTP synthetase (CTPS, UniProt P0A7E5).

ORF/cDNA clones of human eIF4G (UniProt Q04637) and human eIF4A (UniProt P60842) were purchased from DNAFORM (Yokohama, Japan). The sequence of the HEAT1 domain (residues 746-992) of eIF4G was amplified by PCR and cloned into the pCold I vector (Takara-Clontech 3361) by using the XhoI and SalI restriction sites. The full-length sequence of eIF4A was amplified by PCR and cloned via a SLiCE reaction[32] into a modified pET28 vector (Novagen 69864), with an N-terminal octa-histidine tag and a human rhinovirus 3C protease site.

Sequences of the primers used in the cloning are provided as Supplementary Data 1. All the sequences of the constructed plasmids have been confirmed by DNA sequencing.

### Purification of the enzymes for the preparation of rNTPs
BL21(DE3) cells (ThermoFisher Scientific C600003) transformed with the plasmid with the genes encoding the enzymes were cultured in 200 ml of LB medium containing 50 mg/L kanamycin, for 16 h at 37 °C. The culture was transferred to 2 L of LB medium containing 50 mg/L kanamycin and the cell culture was continued at 37 °C until an $OD_{600}$ of 2 was reached. Protein expression was then induced by adding IPTG to a final concentration of 1 mM, and the cell culture was continued for 3-4 h. The cells were collected by centrifugation at 10,150×$g$ for 15 min and resuspended in 200 ml of buffer, containing 50 mM Tris-HCl (pH 7.5), 150 mM NaCl, and ×1 protease inhibitor cocktail (Nacalai Tesque 03969-21). The cells were disrupted by sonication and centrifuged at 14,000×$g$ for 30 min. The supernatant was applied to 10 mL of Ni-NTA agarose resin (QIAGEN 30230). The resin was washed extensively by 300 mL of buffer, containing 50 mM NaPi (pH 8.0), 300 mM NaCl, 1 mM DTT, and 10 mM imidazole. The enzymes were eluted from the resin by buffer containing 50 mM NaPi (pH 8.0), 300 mM NaCl, 1 mM DTT, and 300 mM imidazole. The fractions containing the enzymes were then pooled and dialyzed against buffer, containing 50 mM NaPi (pH 7.4), 300 mM NaCl, 1 mM DTT, and 10% glycerol. After the dialysis, the samples were concentrated with an Amicon-Ultra15 filter (10 kDa cutoff, Merck UFC901024), and glycerol was added to a final concentration of 40%. The samples were then frozen in liquid nitrogen, and stored at −80 °C until use.

### Preparation of the isotopically labeled rNTPs
The [{1′, 2′, 3′, 4′, 5′, 5″}-$^2$H, $^{13}$C8] ATP was enzymatically synthesized by using D-[1, 2, 3, 4, 5, 5′-$^2$H$_6$] ribose (Omicron Biochemicals RIB-040) and $^{13}$C8 adenine (Cambridge Isotope Laboratories CLM-1654-PK) as substrates, with the dATP regeneration system[33]. In the reaction, in-house purified RK, PRPPS, and APRT were used, in addition to creatine kinase from rabbit muscle (CK, Roche 10127566001), myokinase from rabbit muscle (MK, Sigma-Aldrich M3003), and thermostable inorganic pyrophosphatase (TIPP, New England Biolabs M0296L). The [{1′, 2′, 3′, 4′, 5′, 5″}-$^2$H, $^{13}$C8] GTP was enzymatically synthesized by using D-[1, 2, 3, 4, 5, 5′-$^2$H$_6$] ribose (Omicron Biochemicals RIB-040) and $^{13}$C8 guanine (Cambridge Isotope Laboratories CLM-1019-PK) as substrates, with the dATP regeneration system[33]. In-house purified RK, PRPPS, XGPRT, and GMK were used in addition to CK, MK, and TIPP in the reaction.

Deuterated uracil was prepared by heating a mixture of 150 mg uracil at natural abundance, 15 mg palladium on carbon (Sigma 205680-10 G), and 30 mL D$_2$O under a H$_2$ atmosphere at 160 °C for 24 h[34]. The reaction was filtered, and the solvent was removed by evaporation. The u-$^2$H UTP was enzymatically synthesized by using D-[1, 2, 3, 4, 5, 5′-$^2$H$_6$] ribose (Omicron Biochemicals RIB-040) and the deuterated uracil as substrates, with the dATP regeneration system[35]. In-house purified RK, PRPPS, UPRT, and UMPK were used in addition to CK, MK, and TIPP. The u-$^2$H CTP was obtained by converting the u-$^2$H UTP with CTPS, by using NH$_4$Cl as a substrate[35].

After the reaction, the samples were filtered through an Amicon-Ultra 15 (30 kDa cutoff, Merck UFC903024) to remove the protein components, lyophilized, and re-dissolved in water. The pH value of each sample was adjusted to 9-10 before application to Affi-Gel Boronate gel resin (Bio-Rad 1536103). The resin was washed with a 1 M triethylammonium hydrogen carbonate solution (TEAB, FUJIFILM Wako Pure Chemical 208-08385), and the isotopically labeled rNTPs were eluted by CO$_2$ acidified water (pH 4.6). The fractions containing the rNTPs were pooled, lyophilized, re-dissolved in water, and stored at −20 °C until use.

### Purification of RNA
The plasmids for in vitro RNA transcription were used as templates for PCR reactions to amplify the regions containing the T7 promoter and the sequence of interest. Sequences of the primers used in the cloning are provided as Supplementary Data. For J-K-St and the StA$_{SL}$ domain, the two 5′-terminal nucleotides of the reverse primer were 2′-O-methylated to suppress the heterogeneity at the 3′ ends of the transcripts[36]. The PCR products were extracted with phenol/chloroform/isoamyl alcohol and precipitated with isopropanol. RNA samples were obtained by in vitro transcription using T7 RNA polymerase and the PCR-amplified template DNA. The isotopically labeled rNTPs were used for the [u-$^2$H, Ade-{$^1$H2, $^1$H8, $^{13}$C8}, Gua-{$^1$H8, $^{13}$C8}]-labeled RNA samples. After transcription, the reaction was halted by adding EDTA to a final concentration of 25 mM and heat denaturation, and then cooled to room temperature. The sample was then concentrated by an Amicon-Ultra 15 filter (3 kDa cutoff, Merck UFC900324), and purified with urea-denaturing disk polyacrylamide gels (National Diagnostics EC-835 and EC-840), using an electroosmotic medium pump[37]. The fractions containing the RNA sample were pooled and concentrated with Microcep Advance filters (Pall MCP003C41), and then washed five times with 1 M NaCl before buffer exchanges for further applications.

### Expression and purification of eIF4G$^{HEAT1}$ and eIF4A
Human eIF4G$^{HEAT1}$ was expressed in the *E. coli* BL21 (DE3) strain (Merck 69450) by induction with 1 mM isopropyl β-D-thiogalactopyranoside (IPTG), and cells were grown for 24 h at 15 °C. Harvested cells were disrupted by sonication in Buffer A (20 mM Tris, pH 7.5, and 10% glycerol (v/v)), containing 300 mM KCl, 10 µg/mL DNase I, 0.5 mM PMSF, and a cOmplete protease inhibitor cocktail tablet

(Roche 11697498001). After centrifugation at 70,000 $g$ for 45 min, the supernatant was passed through TALON metal affinity resin (Takara-Clontech 635504). The resin was washed with Buffer A containing 800 mM KCl, and subsequently with Buffer A containing 100 mM KCl and 10 mM imidazole. The protein was eluted with Buffer A containing 100 mM KCl and 300 mM imidazole. The eIF4G$^{HEAT1}$ protein was further purified by size-exclusion chromatography using a Superdex 75 Increase 10/300 column (Cytiva 29148721), with buffer containing 20 mM HEPES, pH 7.5, 150 mM KCl, 5% glycerol (v/v), 0.1 mM EDTA, and 2 mM DTT. Pooled peak fractions were concentrated to ~250 μM with an Amicon-Ultra 4 filter (3 kDa cutoff, Merck UFC800324), flash frozen in liquid nitrogen, and stored at −80 °C.

Full-length human eIF4A was expressed in the *E. coli* BL21 Star (DE3) strain (Thermo Fisher Scientific C601003), induced with 1 mM IPTG and grown for 4 h at 37 °C. Harvested cells were lysed and the protein was purified with TALON resin, as described above. The eluted protein was dialyzed against Q-loading buffer (20 mM Tris, pH 7.5, 100 mM KCl, 5% glycerol (v/v), 0.1 mM EDTA, and 2 mM DTT). The sample was applied to a 1-mL RESOURCE Q column (Cytiva 17117701) equilibrated with Q-loading buffer. The eIF4A protein was eluted with a linear KCl gradient from 100 mM to 400 mM. Pooled peak fractions were concentrated to ~170 μM with an Amicon-Ultra 4 filter (10 kDa cutoff, Merck UFC801024), flash frozen in liquid nitrogen, and stored at −80 °C.

## Cryo-electron microscopy

To unambiguously distinguish the St domain from the other J and K domains in the obtained EM maps and increase the size of the complex, St-extended J-K-St, where the St domain is extended by seven-base pairs, was prepared in addition to J-K-St (Supplementary Fig. 2c). The wild-type and St-extended J-K-St RNA samples were dialyzed against buffer, containing 40 mM potassium phosphate, pH 7.5, 200 mM NaCl, 0.2 mM EDTA, and 5 mM DTT, and concentrated to ~200 μM. The J-K-St/eIF4G$^{HEAT1}$/eIF4A ternary complex was formed by mixing the J-K-St RNA with the thawed eIF4G$^{HEAT1}$ and eIF4A proteins at a molar ratio of 1.07:1:1.13. The sample was mixed with 100x ATP stock to a final concentration of 1 mM, incubated on ice for 10 min, and then applied to a Superdex 200 Increase 10/300 column, equilibrated with buffer containing 20 mM HEPES, pH 7.5, 150 mM KCl, 2 mM MgCl$_2$, 100 μM ATP, and 1 mM DTT. The absorbance at 260 nm of the top peak fraction containing the complex was ~3.2, and 3.5 μl aliquots of the fraction were applied to Quantifoil R1.2/1.3 300 mesh Au grids, blotted for 3 sec at 4 °C, and plunge-frozen in liquid ethane using a Vitrobot Mark IV (Thermo Fisher Scientific). Initial cryo-EM data were manually collected at 300 kV using a JEM-Z300CF electron microscope (JEOL) equipped with a K2 Summit direct electron detector (Gatan) in the electron counting mode. The calibrated pixel size was 0.98 Å on the specimen level, and exposures of 8 sec were dose-fractionated into 40 frames with an electron flux of 8 e-/pix/sec. Larger cryo-EM data of the complex containing the St-extended J-K-St RNA were collected at 300 kV using a JEM-Z320FHC electron microscope (JEOL) equipped with a K2 Summit direct electron detector in the electron counting mode, using SerialEM[38]. The calibrated pixel size was 0.765 Å on the specimen level, and exposures of 8 sec were dose-fractionated into 40 frames with an electron flux of 5 e-/pix/sec.

Image processing was performed with RELION-3.1[39,40]. Specimen movement was corrected using MotionCor2[41], the contrast transfer function (CTF) parameters were estimated using Gctf[42], and images showing substantial ice contamination, abnormal backgrounds, or poor Thon rings were discarded. Particles were initially picked with Gautomatch (https://www.mrc-lmb.cam.ac.uk/kzhang/Gautomatch/) without templates, and then the picked particles were subjected to 2D classification. Five representative class averages were selected as templates for Gautomatch to pick particles from all the micrographs collected on JEM-Z320FHC. The auto-picked particles were extracted

with rescaling from 300 ×300 to 100 ×100 pixel images and subjected to two clean-up rounds of 2D classification. An initial reference map was generated in RELION, and then used as the reference for the 3D classification of the cleaned-up particles into four classes, among which one was selected and used to re-extract the corresponding particles into 200 × 200 pixel images. Subsequent 3D refinement with C1 symmetry, CTF refinement, and Bayesian polishing yielded a map at 3.8 Å resolution. Further 3D refinement by local alignment with a mask on a better-ordered region yielded a focused map at 3.7 Å resolution (Supplementary Fig. 3).

## Model building and refinement

The cryo-EM models of eIF4G$^{HEAT1}$ and eIF4A in the human 48S translational initiation complex (PDB ID: 6ZMW)[43] were used and fit into the cryo-EM map by UCSF Chimera[44] without disrupting the heterodimeric configuration. Further fitting and adjustments were performed manually in COOT, by using the chain refinement module with self-local distance restraints[45]. The NMR model of J-K-St (PDB ID: 2NBX)[9] was manually fit into the cryo-EM map, initially based on the structure of the J domain, by UCSF Chimera. Due to the large conformational changes at the three-way junction and the A$_{SL}$ domain, the K and St domains did not fit well into the density. Therefore, the two domains were separated by removing the residues at the junctional region from the model, and then individually fit into the density by UCSF Chimera. The model was manually adjusted and refined in COOT, and the removed residues were built de novo. The models of the proteins and RNA were combined, and iterative cycles of real-space refinement in PHENIX[46] with secondary structure restraints and manual adjustments were performed to the whole complex map, yielding the cryo-EM model of the ternary complex. The final model of J-K-St/eIF4G$^{HEAT1}$ was obtained by further refinement in PHENIX to the focused map. The refinement statistics are summarized in Supplementary Table 1. The structures are rendered by using PyMol or UCSF ChimeraX[47].

## NMR spectroscopy

For NMR experiments, the RNA samples were dissolved in NMR buffer (10 mM NaPi (pH 6.4), 10 mM KCl). The samples were then lyophilized and redissolved in the same volume of 99.96% D$_2$O (Cambridge Isotope Laboratories DLM-4-10×0.7). All NMR spectra were acquired with Bruker Ascend Evo 1.0 GHz, Bruker Avance 900, Bruker Avance 800, and Bruker Avance 600 spectrometers equipped with a cryogenic probe. Two-dimensional $^1$H–$^{13}$C aromatic TROSY spectra were acquired by using spin-state selective coherence transfer[21], in a non-constant time manner. The assignments of the $^1$H8-$^{13}$C8 HMQC signals from the J domain have been available in the Biological Magnetic Resonance Data Bank (BMRB) under accession code 25997. Differences in the $^{13}$C8 chemical shifts due to the selection of the TROSY components in the $^{13}$C dimension in the $^1$H8-$^{13}$C8 TROSY spectra are considered to obtain the assignments used in this study. For the StA$_{SL}$ domain, the $^1$H-$^1$H NOESY spectra in D$_2$O were analyzed by referring to our previous report for the ΔJΔK domain[9], whose chemical shifts are available in the BMRB under accession code 26000. The chemical shift values of the $^1$H8 signals of the ΔJΔK domain were used to assign $^{13}$C8 aromatic TROSY signals in the StA$_{SL}$ domain, by finding corresponding signals in the $^1$H8-$^{13}$C8 aromatic TROSY spectrum.

The $^{13}$C single quantum relaxation dispersion experiments were recorded at 30 °C, using a TROSY-based pulse sequence[23] with a phase modification in the spin-state selective inversion element during the Carr-Purcell-Meiboom-Gill (CPMG) pulse train[48]. The $R_{ex}$ values were obtained at the $^{13}$C frequency of 200 MHz with the constant time CPMG relaxation periods were set to 30 ms for the StA$_{SL}$ domain and 40 ms for the J domain, respectively. The CPMG frequencies, ν$_{CPMG}$, were set to 33.3 and 1,500 Hz for the StA$_{SL}$ domain, and 25 and 1,500 Hz for the J domain. For the relaxation dispersion analyses of the StA$_{SL}$ domain, the constant time CPMG relaxation periods were set to 16 and

30 ms at the $^{13}$C frequency of 250 and 150 MHz, respectively. The $\nu_{CPMG}$ values were varied between 62.5 Hz and 2000 Hz and 33.3 Hz and 2000 Hz at the $^{13}$C frequency of 250 and 150 MHz, respectively. All data were processed with Topspin 4.0.8 (Bruker)

## Analyses of the NMR relaxation dispersion data

The values of the effective relaxation rates measured in the presence of a $\nu_{CPMG}$ Hz CPMG pulse train, $R_{2,\text{eff}}(\nu_{CPMG})$, were obtained using Eq. 1, where $I(\nu_{CPMG})$ and $I(0)$ represent the peak intensities with and without the relaxation period T, respectively.

$$R_{2,\text{eff}} = -\frac{1}{T}\ln\left(\frac{I\left(\nu_{CPMG}\right)}{I(0)}\right) \tag{1}$$

The uncertainties of $R_{2,\text{eff}}(\nu_{CPMG})$ [$\sigma_{R2,\text{eff}}(\nu_{CPMG})$] were calculated using Eq. (2), where $\Delta(\nu_{CPMG})$ represents the noise level at $\nu_{CPMG}$.

$$\sigma_{R2,\text{eff}}\left(\nu_{CPMG}\right) = \frac{1}{T}\sqrt{\left(\frac{\Delta\left(\nu_{CPMG}\right)}{I\left(\nu_{CPMG}\right)}\right)^2 + \left(\frac{\Delta(0)}{I(0)}\right)^2} \tag{2}$$

For global fitting, the $^{13}$C SQ relaxation dispersion curves obtained with two static magnetic fields were simultaneously fitted to the fast 2-state exchange equation (Eq. 3)[24]

$$R_{2,\text{eff}} = R_{2,0} + \frac{\Phi}{k_{ex}}\left(1 - \frac{4\nu_{CPMG}}{k_{ex}}\tanh\left(\frac{k_{ex}}{4\nu_{CPMG}}\right)\right) \tag{3}$$

where $\Phi$ denotes the dispersion amplitude parameter, $R_{2,0}$ denotes the intrinsic transverse relaxation rate, $k_{ex}$ denotes the exchange rate, $\Delta\omega$ denotes the chemical shift difference, and $p_B$ denotes the minor state population. Fitting procedures were conducted by using the L-BFGS-B algorithm implemented in the optim function in R (version 4.2.1). The uncertainties in the fitted exchange parameters were estimated by 100 jackknife simulations, in which two points were removed randomly from each relaxation dispersion curve.

For G777, the $^{13}$C SQ relaxation dispersion curves obtained with two static magnetic fields were fitted numerically to the 3-state exchange model between conformations A, B, and C, where conformation A corresponds to the major conformation in the above-mentioned 2-state model (Eq. 4),

$$R_{2,\text{eff}} = -\frac{1}{T}\ln\left(\frac{\mathbf{M}(T)}{\mathbf{M}(0)}\right) \tag{4}$$

where $\mathbf{M}(0)$ is proportional to the vector $[p_A, \ p_B, \ p_C]^T$, $\mathbf{M}(T) = \left(\mathbf{A}_{\mp}\mathbf{A}_{\pm}\mathbf{A}_{\pm}\mathbf{A}_{\mp}\right)^n\mathbf{M}(0)$, and $n$ is the number of 180° pulses in the CPMG pulse train. The matrix $\mathbf{A}$ is defined as follows (Eqs. 5 and 6),

$$\mathbf{A}_{\pm} = e^{\mathbf{a}_{\pm}\cdot\tau}\text{CPMG} \tag{5}$$

$$\mathbf{a}_{\pm} = \begin{bmatrix} -k_{AB} - k_{AC} - R_{2,0} & k_{BA} & k_{CA} \\ k_{AB} & -k_{BA} - k_{BC} - R_{2,0} \pm i\triangle\omega_{AB} & k_{CB} \\ k_{AC} & k_{BC} & -k_{CB} - k_{CA} - R_{2,0} \pm i\triangle\omega_{AC} \end{bmatrix} \tag{6}$$

where $k_{XY}$ denotes the exchange rate from conformation X to conformation Y, $R_{2,0}$ denotes the intrinsic transverse relaxation rate, $\tau_{CPMG}$ denotes the delay between CPMG pulses, $\Delta\omega_{XY}$ denotes the chemical shift difference between conformations X and Y, and $p_X$ denotes the population of conformation X. The exchange rate between conformations A and B of the above-mentioned global fitting were used as a constant in the fitting. The uncertainties in the exchange parameters were estimated from a set of 100 results of the fitting calculations

where the initial parameters were varied. The obtained parameters were then used in the line-shape analyses, confirming that the fitted exchange parameters explain the experimentally observed signals.

## Isothermal titration calorimetry

Before all isothermal titration calorimetry (ITC) experiments, the eIF4G$^{HEAT1}$ and RNAs were dissolved in ITC buffer (20 mM HEPES-NaOH, pH 6.5, 150 mM NaCl, 2 mM MgCl$_2$, and 1 mM Tris(2-carboxyethyl)phosphine hydrochloride), unless otherwise indicated. For each ITC experiment, reaction heats (in μcal s$^{-1}$) were measured for 2-μl titrations of 150–250 μM of eIF4G$^{HEAT1}$ into 10–20 μM of RNA at 25 °C, with a Microcal PEAQ-ITC (Malvern). The titration data were analyzed using the one-site binding model from MicroCal PEAQ-ITC Analysis Software (Malvern). The $K_d$ values are reported when the c value, which is the ratio of the RNA concentration to the calculated $K_d$ value, is smaller than 5. Error values represent the standard error of the ITC fit. The experiments for the interaction between the J-K-St wild-type and the eIF4G$^{HEAT1}$ wild-type in the presence of 2 mM Mg$^{2+}$ were repeated three times with similar results. All the other experiments were repeated at least twice with similar results.

## Native polyacrylamide gel electrophoresis (PAGE)

PAGE gels of 12% acrylamide were prepared using 30% acrylamide/bis (29:1) solution in 1x TBE buffer (89 mM Tris-HCl, 89 mM boric acid, 2 mM EDTA, pH 8.3). Electrophoresis was conducted in 1x TBE buffer at a constant current of 10 mA at 4 °C for 1.5 h. Following electrophoresis, the gel was stained with a Stains-all solution (Sigma E9379-1G).

## In vitro translation assay

A human cell-free expression system (TaKaRa 3281) was used for transcription-coupled IRES-dependent translation of the reporter protein, β-galactosidase. The DNA construct containing the EMCV IRES and β-galactosidase (supplied as the positive control vector from TaKaRa 3281) was used as a template to obtain A700C, U778C, and U780C variants by PCR-based mutagenesis. These wild-type and mutated DNA constructs were used as templates for in vitro T7 transcription-translation reactions according to the manufacturer's instructions, at the final concentration of 15 ng/μl, in a 10 μl reaction scale. After incubation for 3 hrs at 32 °C, EDTA was added to the final concentration of 25 mM. The amount of β-galactosidase expressed was quantified using the Luminescent β-galactosidase Detection Kit II (Clontech 631712), according to the manufacturer's instructions. The luminescence was measured with an Envision 2104 multilabel plate reader (PerkinElmer). The experiments were repeated seven times and mean values and standard error are reported.

## Reporting summary

Further information on research design is available in the Nature Portfolio Reporting Summary linked to this article.

# Data availability

The cryo-EM density map and corresponding atomic coordinate of the J-K-St/eIF4G$^{HEAT1}$/eIF4A complex have been deposited in the Electron Microscopy Data Bank and the Protein Data Bank under accession codes EMD-35041 and 8HUJ, respectively. The focused cryo-EM density map and corresponding atomic coordinate of the J-K-St/eIF4G$^{HEAT1}$ complex have been deposited under accession codes EMD-36046 and 8J7R, respectively. Other structure data used in this study are available in the Protein Data Bank under accession codes 2NBX, 2NBY, 2NC1, 6ZMW, 1HU3, 6GC5, 4PMI, 6HTU, and 2VSO. The chemical shift data of the StA$_{SL}$ and J domain have been deposited in the BMRB under accession codes 51905 and 51906, respectively. Chemical shift data used in this study are available from the BMRB under the accession codes of 25997 and 26000. Protein sequences used in this study are available from Uniprot under accession codes P0A717 (PRPPS), P0A9J6

(RK), P69503 (APRT), P60546 (GMK), P0A9M5 (XGPRT), P0A8F0 (UPRT), P0A7E9 (UMPK), P0A7E5 (CTPS), Q04637 (eIF4G), and P60842 (eIF4A). Source data are provided with this paper.

## Code availability

Custom codes used to fit the relaxation dispersion data are available at Zenodo under https://doi.org/10.5281/zenodo.8166504.

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

## Acknowledgements

This work was supported by a grant from the Japan Agency for Medical Research and Development (AMED) Grant Number JP21ae0121028 (to I.S.). This work was also supported by a Grant-in-Aid for Scientific Research (A) under Grant Number 20H00451 (to Y.F.).

## Author contributions

S.I., H.S., Y.F., and I.S. designed the research. S.I. prepared the RNA samples, conducted ITC and NMR experiments. H.S. prepared the protein samples and conducted cryo-EM experiments. S.I., H.S., Y.F., and I.S. analyzed the data and wrote the manuscript.

## Competing interests

The authors declare no competing interests.
