## [Peer Review File · Nature Communications]

REVIEWER COMMENTS

Reviewer #1 (Remarks to the Author):

The manuscript by Imai et al. presents an excellent structural and dynamic study of viral IRES RNA and its interaction with host protein factors. IRESs are important for RNA viruses to activate host translational machinery and produce viral proteins. Among them, IRES from the encephalomyocarditis virus (EMCV) interacts with various eukaryotic initiation factors (eIFs) and exhibits one of the highest translational efficiencies. However, the underlying mechanism remains unknown. In the present study, the authors first solved a 3.7Å-resolution cryo-EM structure of EMCV IRES in complex with eIF4G and eIF4A. The ternary structure unveils how eIF4G recognizes EMCV IRES through two binding sites on IRES's J and St domains. Compared to their previously determined free-state structure, the authors found that EMCV IRES undergoes distinct conformational transitions upon eIF4G binding. The authors then carried out NMR relaxation dispersion experiments to characterize the molecular basis of the observed conformational changes of IRES. They revealed that EMCV IRES undergoes us-ms timescale conformational dynamics in its free state and samples lowly populated excited conformations states. Remarkably, one of the excited states in the St domain recapitulates structural features of the eIF4G-bound state. The authors showed that mutations that suppress the us-ms timescale dynamics in EMCV IRES also eliminate its binding to eIF4G. Together, these exciting results unveiled not only the structural basis but also the dynamic basis for how IRES interacts with host eIFs, providing a comprehensive mechanistic understanding of these recognitions. Overall, this study is very well designed, executed, and presented, and the findings should be of significant interest to readers of Nature Communications. Hence, the reviewer highly recommends publishing this excellent work.

In the following, the reviewer has a few minor points for the authors to consider in improving their manuscript.

1. The cryo-EM maps and structures showed a 1:1 ratio between IRES and eIF4G HEAT1 domain. While the ITC data from the J-K-St construct is consistent with the structural data, the ITC data from the St-extended J-K-St construct exhibits a stoichiometry of $N=0.50$. The authors should elaborate on these observed differences for better clarity.
2. The authors analyzed the CPMG relaxation dispersion data of G777 using a three-state exchange model and obtained chemical shifts for GS, ES1, and ES2. CPMG data generally provides absolute

differences in chemical shifts between exchanging states. The authors should elaborate on their derivation of the exact chemical shifts for these states with a fast exchange rate between GS and ES1.

3. The three-state exchange model of G777 showed that ES1 and ES2 interconvert at a relatively slow exchange rate. Since the U778C variant of the StASL domain favors the excited states, it will be insightful to carry out CPMG relaxation dispersion measurements of this variant to directly examine the exchange process between ES1 and ES2 and provide further support for the three-state exchange model.

Reviewer #2 (Remarks to the Author):

The manuscript of Imai et al. presents a nice combined structural and dynamics investigation about how conformational rearrangements of a viral RNA enable interaction with a host initiation factor. The data presented are clear and support the conclusions described in the text. The integration of structure and dynamics derived from NMR experiments provides more insight than the cryo-EM structure alone would have. That said, the results presented are quite detailed and technical, and do not really benefit from the short format of a Nature Communication. The writing, especially the NMR section, is very technical in nature. For a journal like Nature Communications it should be fit for a broader audience, possibly by including a short, more general description of each technique and what it is measuring in practical terms for each experiment. In its current form, I do not think it is appropriate for publication in Nature Communications, and probably should be submitted to a more technical, longer format journal. That said, I offer some suggestions for them to improve the manuscript.

The authors must frame their work in a wider context of previously published research. This is particularly evident in the repeated description of eIF4G as lacking intrinsic RNA binding (specific locations in text noted below), an activity previously shown for yeast and mammalian eIF4G (for example, Berset et al. RNA 2003, DOI: 10.1261/rna.5380903; Yanagiya et al. Mol Cell Biol 2009, DOI: 10.1128/MCB.01187-08). The latter reference maps RNA binding activity to a region of mammalian eIF4G included in the construct used here. There is also an opportunity to compare their structure to previous structural information from biochemical experiments that identified sites of interaction in this complex on both the RNA and protein (Clark et al. J Virol 2003, DOI: 10.1128/JVI.77.23.12441-12449.2003; Kolupaeva et al. Mol Cell Biol 2003, DOI: 10.1128/MCB.23.2.687-698.2003), papers that are relevant but not mentioned in the introduction or discussion.

One concern I have with the structure is that for eIF4A, the PDB validation report lists 63% of the residues as having poor fit to the EM map – I think the authors should justify their inclusion of these residues or model only the portion of the protein that fits into the density.

Finally, I think there are a few experiments that could help strengthen this paper and test their model, especially a functional assay. For example, to connect the conformational dynamics to biochemistry, how does the U778C mutation described affect binding of eIF4G to the RNA? The changes observed in this RNA by NMR for U778C, A700C, and U780C are not connected well to biochemical or functional

experiments - how do these mutations affect IRES function, possibly using a luciferase reporter, or viral fitness in a replicon or full-length infection model? These insights could also give it broader appeal more suitable for Nature Communications, helping it to be more than just a technical structure paper.

Specific comments:

Abstract:

Lines 6-7: "...which is not an intrinsic RNA binder" See comment above about previous data showing intrinsic RNA binding activity of eIF4G.

Line 8: "Here, we analyzed the three-dimensional structure of the complex formed by the EMCV IRES with eIF4G..." Consider editing to clarify that only a fragment of eIF4G was used in this structure, and that eIF4A was also present.

Line 10: "...to hijack a host protein that lacks an intrinsic RNA binding site." See comment above about previous data showing intrinsic RNA binding activity of eIF4G.

Introduction:

Paragraph 2, line 17: "...and is not an intrinsic RNA binder per se" See comment above about previous data showing intrinsic RNA binding activity of eIF4G.

Paragraph 2: the reasoning in the second half of this paragraph seems shaky to me. It's predicated on the assumption that eIF4G doesn't bind RNA, and visual inspection of the surface charge, a very qualitative measure, is offered as evidence to support that idea.

Cryo-EM structure of the J-K-St/eIF4GHEAT1/eIF4A complex:

Paragraph 2, lines 10-11: "All of these bases, except for A724, interact mainly with the positively-charged patch on eIF4GHEAT1..." Consider listing some of these residues, and making a reference to Supplementary Figure 3b and c so that the reader can find the side chain depiction of these sites.

Paragraph 2, lines 13-14: "All of these residues, except for G777, interact mainly with the positively-charged patch on the eIF4GHEAT1..." As written, it is unclear whether this is the same or a different positively charged patch than described earlier.

Characterization of the excited states of the StASL domain:

Paragraph 3: How did U778C affect binding of eIF4GHEAT1 to RNA? This would be nice to tie the NMR data to the ITC data reported for other constructs.

Paragraph 5, lines 11-13: "...as suppressing the dynamics by introducing mutations..." Why were these particular mutations chosen? Please describe why. Another explanation for the loss of binding in these

mutants is that they disrupt the secondary structure of the RNA. Were any experiments done to show that the structure is not disrupted? Something like what is shown in Supplementary Figure 5b and c for the isolated J and StAS1 constructs could suffice.

Discussion:

Paragraph 2, line 1: "EMCV IRES enables this uninvited interaction..." See comment above about previous data showing intrinsic RNA binding activity of eIF4G.

Methods:

Is there a reason why the buffers for the cryo-EM and NMR samples were so different? The pH 7.5 vs. pH 6.4 and 150 mM salt vs. 10 mM salt alone could modulate RNA structure.

Figures:

Figure 1: consider adding a schematic showing the domains of full-length eIF4G, and highlighting which portions are included in the construct used here.

Figure 1a: this diagram makes it seem like J-K-St is sufficient to promote translation initiation. Consider including a cartoon of the full IRES element with J-K-St highlighted.

Figure 2a: what do the light grey U residues in the K domain mean? Are these not modeled in the structure? Please indicate in the figure legend.

Supplementary Figure 1a: the authors use surface charge as an argument supporting no intrinsic RNA binding ability of eIF4G. Are there examples of other 'known' RNA binding proteins that could be shown alongside 4GHEAT1 to show the difference in appearance claimed in the text?

Supplementary Figure 1: how many replicates of the ITC experiment were performed? How were the reported errors determined? Please indicate in the legend or methods.

Reviewer #3 (Remarks to the Author):

Imai et al describe in this manuscript a structural characterization of a fragment of the EMCV IRES in complex with the eukaryotic initiation factor eIF4. IRESs of this sub-type are highly understudied at a structural level what makes this study interesting as it contributes insights in the early recruitment of the EMCV IRES to the ribosome to hijack the protein synthesis capacity of the host cell. The study combines two structural methodologies, namely cryoEM and NMR and a solid biochemical

characterization. Overall, I find the study well designed and executed with a clear and accurate presentation, it is also technically sound. I only have minor concerns which have to do mainly with the visualization of the cryoEM map obtained and the refinement of the atomic model in the cryoEM maps.

-The quality of the cryoEM map is hard to ascertain as only a far view of the map is shown in figure 1c. The map seems to agree with the reported resolution, but a much-detailed graphical explanation is needed. A multipaneled and multiscale subfigure should be provided, focusing in the most interesting regions of the complex so the reader can confidently assess the quality of the map in those regions with key interactions. Similarly, in supplementary figure 2, much-detailed views of the local resolution figure are needed so the reader can unambiguously judge the cryoEM quality map in all areas.

-The final statistics for the atomic model refined into the cryoEM map are not ideal. A cryoEM map at 3.7Å resolution, granted a “modest” resolution, should be enough to obtain a refined model with a molprobity all-atom clashcore better than the reported 24.89. The authors should inspect and manually fix the model in problematic areas and re-refine the fixed model with the aim to obtain molprobity score ideally in the single digit range.

-A phase randomized FSC curve should be provided in supplementary figure 2 to discard artefacts generated by a tight mask during postprocessing as described in:

<https://pubmed.ncbi.nlm.nih.gov/23872039/>

Point-to-point responses to the reviewer comments

Reviewer #1 (Remarks to the Author):

#1-1. The manuscript by Imai et al. presents an excellent structural and dynamic study of viral IRES RNA and its interaction with host protein factors. IRESs are important for RNA viruses to activate host translational machinery and produce viral proteins. Among them, IRES from the encephalomyocarditis virus (EMCV) interacts with various eukaryotic initiation factors (eIFs) and exhibits one of the highest translational efficiencies. However, the underlying mechanism remains unknown. In the present study, the authors first solved a 3.7Å-resolution cryo-EM structure of EMCV IRES in complex with eIF4G and eIF4A. The ternary structure unveils how eIF4G recognizes EMCV IRES through two binding sites on IRES's J and St domains. Compared to their previously determined free-state structure, the authors found that EMCV IRES undergoes distinct conformational transitions upon eIF4G binding. The authors then carried out NMR relaxation dispersion experiments to characterize the molecular basis of the observed conformational changes of IRES. They revealed that EMCV IRES undergoes us-ms timescale conformational dynamics in its free state and samples lowly populated excited conformations states. Remarkably, one of the excited states in the St domain recapitulates structural features of the eIF4G-bound state. The authors showed that mutations that suppress the us-ms timescale dynamics in EMCV IRES also eliminate its binding to eIF4G. Together, these exciting results unveiled not only the structural basis but also the dynamic basis for how IRES interacts with host eIFs, providing a comprehensive mechanistic understanding of these recognitions. Overall, this study is very well designed, executed, and presented, and the findings should be of significant interest to readers of Nature Communications. Hence, the reviewer highly recommends publishing this excellent work.

In the following, the reviewer has a few minor points for the authors to consider in improving their manuscript.

Thank you for providing us with your insightful and constructive feedback on our manuscript. We greatly appreciate your positive evaluation of our work and your recommendation for publication. We have carefully considered your suggestions and have incorporated the changes to enhance the quality of our manuscript. Please

find our point-to-point responses to your comments below.

#1-2. 1. The cryo-EM maps and structures showed a 1:1 ratio between IRES and eIF4G HEAT1 domain. While the ITC data from the J-K-St construct is consistent with the structural data, the ITC data from the St-extended J-K-St construct exhibits a stoichiometry of $N=0.50$. The authors should elaborate on these observed differences for better clarity.

We appreciate the reviewer's comment regarding the observed stoichiometry in the ITC experiments for the St-extended J-K-St with eIF4G^{HEAT1}. As the reviewer mentioned, we observed a stoichiometry of 0.50 for the St-extended J-K-St in the ITC experiment, while the J-K-St exhibited a stoichiometry of 1.13 (Supplementary Fig. S2, a and c, shown below for reference). We agree that this observation requires clarification.

Our interpretation of the observed differences in stoichiometry is that a fraction of the St-extended J-K-St is misfolded to form a multimer, resulting in an effective concentration of the active monomeric St-extended J-K-St lower than that estimated from the absorbance at 260 nm. To confirm this possibility, we conducted native PAGE analyses on the J-K-St and St-extended J-K-St (Supplementary Fig. 2f). As a result, bands corresponding to dimers and trimers are observed only for the St-extended J-K-St, suggesting that the St-extended J-K-St can form multimers, which is not the case for the J-K-St. It should be noted here that discrepancy of the effective concentration of the sample in the ITC cell (i.e., the RNA) does not affect the K_d values obtained in the one-site fitting model employed here. Also, we would like to emphasize that despite the heterogeneous folding of the St-extended J-K-St, the misfolded molecules were not included in the cryo-EM analyses. The samples were purified by size exclusion chromatography before the cryo-EM analyses, ensuring that only the RNA molecules that formed a complex with eIF4G^{HEAT1}/eIF4A were used.

In response to the reviewer's suggestion, we have added the native PAGE analysis in the Supplementary Fig. 2f, and referred to these data from the main text, as follows. Thanks to the change, the observed difference of the stoichiometry is now presented in a clearer manner.

Results

P.7, lines 123-128

The isothermal titration calorimetry (ITC) experiment revealed that J-K-St binds to eIF4G^{HEAT1} with K_d values of 0.149 ± 0.012 and 2.07 ± 0.28 μM , in the presence of 2 mM Mg^{2+} and the absence of Mg^{2+} , respectively (Supplementary Fig. 2, a and b), showing the Mg^{2+} dependence of the interaction. We analyzed the structure of the J-K-St/eIF4G^{HEAT1}/eIF4A complex in the presence of Mg^{2+} at a 3.8 Å resolution by cryo-EM (Fig. 1d, and Supplementary Figs. 2c, 3, and 4).

Methods

P.37, lines 664-668

Native polyacrylamide gel electrophoresis (PAGE)

PAGE gels of 12% acrylamide were prepared using 30% acrylamide/bis (29:1) solution in 1x TBE buffer (89 mM Tris-HCl, 89 mM boric acid, 2 mM EDTA, pH 8.3). Electrophoresis was conducted in 1x TBE buffer at a constant current of 10 mA at 4°C for 1.5 h. Following electrophoresis, the gel was stained with Stains-all solution (Sigma).

Supplementary Fig. 2 | ITC experiments

(a, b) ITC data of the interaction between J-K-St and eIF4G^{HEAT1}, in the presence of 2 mM Mg²⁺ (a) and in the absence of Mg²⁺ (b).

(c) ITC data of the interaction between St-extended J-K-St and eIF4G^{HEAT1}, in the presence of 2 mM Mg²⁺. Extended regions at the terminus of the St domain are schematically shown in green in the inset. **See also (f) for the reduced stoichiometry.**

(d) ITC data of the interaction between the J domain and eIF4G^{HEAT1}, in the presence of 2 mM Mg²⁺. N.D., not determined. See Supplementary Fig. 6a for the design of the J domain.

(e) ITC data of the interaction between the St_{ASL} domain and eIF4G^{HEAT1} in the presence of 2 mM Mg²⁺. N.D., not determined. See Supplementary Fig. 6a for the design of the St_{ASL} domain.

(f) Native polyacrylamide gel electrophoresis (PAGE) analysis of J-K-St and St-extended J-K-St. Bands corresponding to the dimer and trimer were observed only for St-extended J-K-St. These multimers, formed due to misfolding, decrease the effective concentration of the variant with respect to that calculated from the absorbance at 260 nm, resulting in the reduced stoichiometry ($N = 0.50 \pm 0.01$) observed in (c). It should be noted here that discrepancy of the effective concentration of the sample in the ITC cell (i.e., the RNA) does not affect the K_d values obtained in the one-site fitting model employed here. The misfolded molecules were not included in the cryo-EM analyses, as the samples were purified before the cryo-EM analyses, ensuring that only the RNA molecules that formed a complex with eIF4G^{HEAT1}/eIF4A were used.

N.D., not determined. The ITC experiment (a) was repeated three times with similar results. The ITC experiments (b-e) were conducted at least two times with similar results. The \pm values indicate the standard error of the fitting.

#1-3. 2. The authors analyzed the CPMG relaxation dispersion data of G777 using a three-state exchange model and obtained chemical shifts for GS, ES1, and ES2. CPMG data generally provides absolute differences in chemical shifts between exchanging states. The authors should elaborate on their derivation of the exact chemical shifts for these states with a fast exchange rate between GS and ES1.

We appreciate the reviewer for the important comment on the determination of the chemical shift differences obtained from the CPMG relaxation dispersion analyses. As the reviewer commented, CPMG data provides absolute differences in chemical shift between exchanging states. In the case of the three-state exchange model for G777, numerical simulation shows that the ^{13}C chemical shift differences between the ground state (GS) and the excited state 1 (ES1) (defined as $\Delta\omega_{\text{C,ES1}}$) and GS and ES2 (defined as $\Delta\omega_{\text{C,ES2}}$) have opposite signs, while the signs of each chemical shift differences, i.e., which excited state is at the higher magnetic field with respect to GS, cannot be determined from the CPMG data, because reversing the signs of $\Delta\omega_{\text{C,ES1}}$ and $\Delta\omega_{\text{C,ES2}}$ gave the identical CPMG curves (Supplementary Fig.8). This is apparent from the fact that the exchange between ES1 and ES2 is active in the three-state exchange model, where the absolute chemical shift difference between ES1 and ES2, $|\Delta\omega_{\text{C,ES1}} - \Delta\omega_{\text{C,ES2}}|$, is relevant to the relaxation dispersion profile.

Supplementary Fig. 8 | Effect of the signs in the chemical shift differences on the relaxation dispersion curves from G777

(a) Definitions of parameters used in the three-state exchange.

(b-e) The calculated dispersion curves for the three-state exchange of G777 are shown as lines, with experimentally obtained $R_{2,\text{eff}}$ values shown as points. Data shown in red are at the ^{13}C frequency of 250 MHz, whereas data shown in blue are at 150 MHz. Error bars indicate experimental errors derived from the signal-to-noise ratio of each correlation, as written in Methods. (b) $\Delta\omega_{\text{C,ES1}} = -1.59$ ppm, $\Delta\omega_{\text{C,ES2}} = 0.26$ ppm, (c) $\Delta\omega_{\text{C,ES1}} = 1.59$ ppm, $\Delta\omega_{\text{C,ES2}} = -0.26$ ppm, (d) $\Delta\omega_{\text{C,ES1}} = 1.59$ ppm, $\Delta\omega_{\text{C,ES2}} = 0.26$ ppm, (e) $\Delta\omega_{\text{C,ES1}} = -1.59$ ppm, $\Delta\omega_{\text{C,ES2}} = -0.26$ ppm. The other parameters, $\rho_{\text{ES1}} = 0.183$, $\rho_{\text{ES2}} = 0.017$, $k_{\text{exGS,ES1}} = 18,100 \text{ s}^{-1}$, $k_{\text{exGS,ES2}} = 163 \text{ s}^{-1}$, and $k_{\text{exES1,ES2}} = 630 \text{ s}^{-1}$, are identical for all the four panels. On the right, the expanded region of ν_{CPMG} of 0-300 Hz (shown as green dashed boxes on the left) are shown. Reversing the signs of $\Delta\omega_{\text{C,ES1}}$ and $\Delta\omega_{\text{C,ES2}}$ at the same time retains the curve indistinguishable (b and c), whereas reversing the signs of either of $\Delta\omega_{\text{C,ES2}}$ (b and d) or

$\Delta\omega_{C,ES1}$ (b and e) changes the curves, and the calculated curves differ from the experimental data points (d and e, green open boxes on the right). These results demonstrate that $\Delta\omega_{C,ES1}$ and $\Delta\omega_{C,ES2}$ should have different signs, but it is not possible to distinguish which one has the negative sign.

To determine the signs of $\Delta\omega_{C,ES1}$ and $\Delta\omega_{C,ES2}$ we compared the ^{13}C chemical shifts of the G777 C8 signals observed in heteronuclear single quantum coherence (HSQC) and heteronuclear multiple quantum coherence (HMQC) spectroscopy, following the method proposed by Kay's group (Skrynnikov NR, Dahlquist FW, and Kay LE., *J. Am. Chem. Soc.* (2002) 123:12352. [Redacted]). In this paper, it is shown that in the two-state exchange system, the HSQC peak is positioned closer to the resonance of the invisible minor species than the corresponding HMQC peak when $|\Delta\omega| < \sqrt{3}k_b$, where $\Delta\omega$ is the difference of Zeeman frequencies between the major and minor states in rad s^{-1} , and k_b is the backward exchange rate from the minor to the major state.

[Redacted]

Although this HSQC-HMQC shift method is shown to be applicable only to the two-state exchange system in the paper, line shape simulation of the three-state exchanging system of G777 showed that as long as $|\Delta\omega_{H,ES1}|$, the absolute ^1H chemical shift difference between GS and ES1, is smaller than 0.6 ppm, the HSQC peak of G777 $^{13}\text{C}8$ is observed closer to the invisible ES2 peak (Supplementary Fig. 9, a and b, within the green boxes). When $|\Delta\omega_{H,ES1}|$ is larger than 0.6 ppm, the HSQC peak of G777 C8 may be observed further from the ES2 peak than the HMQC peak (Supplementary Fig. 9, a and b, out of the green boxes). However, if that were the case, the HMQC signal would be severely broadened so that the ^{13}C line width of G777 C8 would be more than twice that of the HSQC signal because of the exchange broadening effect due to the contribution of $\Delta\omega_{H,ES1}$ to the HMQC signal, which would be apparent when two signals experimentally obtained are compared. Supplementary Fig. 9c below shows the ^{13}C signals of G777 C8 in the HSQC and HMQC spectra, experimentally obtained at the ^1H frequency of 600 MHz. The HSQC signal is observed downfield to the HMQC spectra, and the HMQC signal is not broadened more than the HSQC signal. Actually, the HMQC signal is even

sharper than the HSQC signal, possibly because the broadening due to the large one-bond heteronuclear ^1H - ^{13}C dipole-dipole interaction is absent in the indirect dimension of HMQC spectra (Marino JP *et al.*, *J. Am. Chem. Soc.* (1997) 119:7361), which is not taken into account in the above-mentioned calculation. These results indicated that the invisible ES2 signal is located to the downfield of the observed signal, and thus is $\Delta\omega_{\text{C,ES1}}$ and $\Delta\omega_{\text{C,ES2}}$ are negative and positive, respectively.

Supplementary Fig. 9 | Determination of the signs of the chemical shift differences by comparing HSQC and HMQC signals

(a and b) Simulated ^{13}C line shapes of the signals in the indirect dimension of the heteronuclear single-quantum coherence (HSQC, blue) and heteronuclear multiple quantum coherence (HMQC, red) spectra at the ^{13}C frequency of 150 MHz. Calculations were conducted by using the parameters $p_{ES1} = 0.183$, $p_{ES2} = 0.017$, $k_{\text{exGS},ES1} = 18,100 \text{ s}^{-1}$, $k_{\text{exGS},ES2} = 163 \text{ s}^{-1}$, and $k_{\text{exES1},ES2} = 630 \text{ s}^{-1}$, where $\Delta\omega_{C,ES1} = -1.59$ ppm and $\Delta\omega_{C,ES2} = 0.26$ ppm (a), or $\Delta\omega_{C,ES1} = 1.59$ ppm, $\Delta\omega_{C,ES2} = -0.26$ ppm (b). For calculation of the line shape in the HMQC spectra, chemical shift differences of the ES1 and ES2 from GS in the ^1H dimension, $\Delta\omega_{H,ES1}$ and $\Delta\omega_{H,ES2}$, are systematically varied to be -0.6 , -0.3 , 0 , 0.3 , or 0.6 ppm. Green boxes indicate the conditions where HMQC signals are not broadened more than twice than HSQC signals. Within the conditions within the green boxes, HSQC signals would be observed at the lower field when $\Delta\omega_{C,ES1} = -1.59$ ppm and $\Delta\omega_{C,ES2} = 0.26$ ppm (a),

whereas at the upper field when $\Delta\omega_{C,ES1} = 1.59$ ppm, $\Delta\omega_{C,ES2} = -0.26$ ppm (b). The relative direction of HSQC signals with respect to the HMQC signals are indicated by green arrows.

(c) Experimentally obtained signal of G777 at 30°C and at the ^{13}C frequency of 150 MHz. The HSQC signal was observed 0.01 ppm lower field with respect to the HMQC signal, indicating that $\Delta\omega_{C,ES1} = -1.59$ ppm and $\Delta\omega_{C,ES2} = 0.26$ ppm (a) is correct.

In response to the reviewer's comment, we added Supplementary Figs. 8 and 9, and referred to these from the main text, as follows. We thank the reviewer for pointing out the importance of explaining the derivation of the chemical shifts of the exchanging states from the CPMG data. We believe that these changes have significantly improved the clarity and completeness of our manuscript.

Results

P.13, lines 229-234

By using the exchange parameters obtained from the global fit analyses mentioned above, the ^{13}C SQ relaxation dispersion profiles of G777 were fit numerically to the three-state exchange model, where G777 transitions between the ground state (GS), and two distinct excited states 1 and 2 (ES1 and ES2) that are not distinguished as chemical shift differences of the other bases (Fig. 4b, bottom). As a result, we obtained a set of kinetic parameters and the ^{13}C chemical shifts of G777 C8 in each state (Fig. 4c, Supplementary Figs. 8 and 9).

#1-4. 3. The three-state exchange model of G777 showed that ES1 and ES2 interconvert at a relatively slow exchange rate. Since the U778C variant of the StASL domain favors the excited states, it will be insightful to carry out CPMG relaxation dispersion measurements of this variant to directly examine the exchange process between ES1 and ES2 and provide further support for the three-state exchange model.

We are grateful for the reviewer's insightful suggestion to conduct CPMG relaxation dispersion experiments on the U778C variant of StASL domain, which preferentially occupies the excited states. In response to this

suggestion, we prepared [$u\text{-}^2\text{H}$, Ade- $\{^1\text{H8}, ^1\text{H2}, ^{13}\text{C8}\}$, Gua- $\{^1\text{H8}, ^{13}\text{C8}\}$] StA_{SL}(U778C) and performed $^1\text{H8}\text{-}^{13}\text{C8}$ aromatic TROSY and the ^{13}C relaxation dispersion experiments. Notably, the signal from G777 is broadened and not observed in the ^{13}C aromatic TROSY for this variant (Fig. R1). In addition to G777, signals from A_{SL} and St domains, namely those from G686, A687, A688, G689, G690, A691, G693, G767, A771, A772, A773, A774, A775, A781, G782, and G783 were broadened or exhibited chemical shift perturbations at the ^1H frequency of 1 GHz. These observations support that the cooperative conformational equilibrium observed in the St and A_{SL} domain is altered by the U778C point mutation.

Fig. R1 $^1\text{H8}\text{-}^{13}\text{C8}$ aromatic TROSY of StA_{SL} U778C variant

(a) Overlay of the $^1\text{H}\text{-}^{13}\text{C}$ aromatic TROSY spectra of the wild-type StA_{SL} domain (black) and its U778C (red) variant, acquired at 30°C and at the ^1H frequency of 1 GHz. Signals broadened or exhibited chemical shift perturbation are labeled. The other signals are not largely perturbed, indicating that the molecular structure are not deteriorated by the introduction of the U778C mutation. Asterisks indicate unassigned minor peaks.

(b) Mapping of the perturbed signals onto the secondary structure (left) or ternary structure model (right) of the StA_{SL} domain. Red: U778, green: broadened or chemical shift-perturbed bases. These perturbed bases are located widely in the St and A_{SL} domains, indicating that the mutation perturbed the cooperative conformational equilibrium within these domains.

Although relaxation dispersion analyses could not be performed for G777 in this U778C variant due to the broadening, we were able to conduct quantitative analyses for A687, G690, A771, and A772 (Fig. R2). Intriguingly, these bases demonstrated non-linear relaxation dispersion profiles that could not be accounted for by either the fast two-state exchange equation (Luz and Meiboom, *J. Chem. Phys.* (1963) 39:366–370) or the general two-state exchange equation (Carver and Richards, *J. Magn. Reson.* (1972) 6:89). Consequently, we employed the general three-state exchange model derived from the analysis of G777 in the wild-type, assuming that the exchange rates remain unaffected by the mutation, and that the chemical shift differences between ES1 and ES2 are smaller than 0.2 ppm, since the linear profile of the relaxation dispersion curves observed in the wild type indicated that the chemical shift differences between ES1 and ES2 are small. As a result, it was shown that the relaxation profiles of these bases can be concurrently explained by populations of the ground state of $36.9 \pm 3.1\%$ that were determined for the wild type, supporting that the ground state is destabilized by the mutation (Fig. R2b). These results provide further support for the three-state exchange model.

Fig. R2 ^{13}C relaxation dispersion analyses of the StAS_L domain U778C variant

(a) Relaxation dispersion profiles of A687, A690, A771, A772 of the wild-type StAS_L domain (left, reprinted from Supplementary Fig. 7 for reference), or the U778C variant (right). Red and blue points are the data measured at the ^{13}C frequency of 250 and 150 MHz, respectively. The lines represent the calculated curve obtained from the fitting analyses at the corresponding magnetic fields. The profiles observed for the U778C variants cannot be accounted for by the simple two-state exchange model, indicating that the exchange process has changed by the U778C mutation.

(b) Exchange parameters obtained from the fitting. (left) Parameters obtained from the fitting of the relaxation dispersion curves of G777 in the wild-type, reprinted from Fig.4c for reference. (right) Parameters obtained from the global fitting of the relaxation dispersion curves of A687, G690, A771, and A772 in the U778C variant (a), with the exchange rates fixed as those from the wild-type.

In accordance with the reviewer's comment, we have added Supplementary Fig. 11 and incorporated the results of the relaxation dispersion analyses of the U778C variant into the main text, as follows. We would like to appreciate the reviewer for suggesting to perform the relaxation dispersion analyses of the U778C variant, as it made our three-state exchange model more solid and sophisticated.

Results

P.16, lines 286-301

We subsequently performed the ^{13}C SQ relaxation dispersion experiment on the St_{ASL} U778C variant (Supplementary Fig. 11). The signal from G777 C8 was broadened beyond detection due to the introduction of the U778C mutation possibly reflecting the shift in the conformational equilibrium, which makes it unanalyzable by the relaxation dispersion experiment. However, it was demonstrated that relaxation dispersion curves of G687, G690, A771, and A772, which exhibited linear profiles in the wild-type (Fig. 4b and Supplementary Fig. 7), displayed non-linear profiles in the U778C variant (Supplementary Fig. 11c). These curves could not be fit by either the fast two-state exchange equation²⁴ or the general two-state exchange equation²⁹, suggesting that the modulation in the conformational equilibrium altered the chemical exchange among GS, ES1 and ES2 evident for these bases. Global fitting of the relaxation dispersion curves of these bases with the three-state exchange model, assuming exchange rates equivalent to the wild-type, revealed that the populations of GS, ES1, and ES2 are 36.9 ± 3.1 , 58.8 ± 2.9 , and 4.3 ± 0.1 %, respectively, for this variant (Supplementary Fig. 11d), which is markedly different from the populations of 80.4 ± 2.9 , 17.9 ± 2.9 , and 1.7 ± 0.1 % observed for the wild-type (Fig. 4c). These results quantitatively indicate that the U778C mutation decreases the population of GS while increasing the populations of ES1 and ES2, and support the notion that the upper stem of the St domain is register-shifted in ES1 and ES2.

Supplementary Fig. 11 | Relaxation dispersion analyses of the U778C variant of the St_{ASL} domain (a) Overlay of the ¹H8-¹³C8 aromatic TROSY spectra of the wild-type St_{ASL} domain (black) and its U778C variant (red). Signals broadened or exhibited chemical shift perturbation are labeled. The other signals are not largely perturbed, indicating that the structure is not deteriorated by the introduction of the U778C mutation. Asterisks indicate unassigned minor peaks. (b) Mapping of the perturbed signals onto the secondary structure (left) or tertiary structure model (right) of the St_{ASL} domain. Red: U778, green: broadened or chemical shift-perturbed bases. These perturbed bases are located widely in the St and A_{SL} domains, indicating that the U778C mutation perturbed the cooperative conformational equilibrium within these domains. (c) Relaxation dispersion curve fitting. R_{2,eff} values obtained at 30°C and at the ¹³C frequencies of 250 MHz and 150 MHz are shown by red and blue points, respectively. The relaxation dispersion profiles were globally fit with the 3-site exchange model with the exchange rates obtained for G777

in the wild-type (Fig. 4c), while assuming that the chemical shift differences between ES1 and ES2 are smaller than 0.2 ppm. A representative fitted curves are shown as red and blue lines for the ^{13}C frequencies of 250 MHz and 150 MHz, respectively. Error bars indicate experimental errors derived from the signal-to-noise ratio of each correlation, as written in Methods.

(d) Exchange parameters obtained from the fitting. Parameters obtained from the global fitting of the relaxation dispersion curves of A687, G690, A771, and A772 in the U778C variant (c), with the exchange rates fixed as those from the analyses in the wild-type (Fig. 4c).

References

24. Luz, Z. & Meiboom, S. Nuclear Magnetic Resonance Study of the Protolysis of Trimethylammonium Ion in Aqueous Solution—Order of the Reaction with Respect to Solvent. *J. Chem. Phys.* **39**, 366–370 (1963).
29. Carver, J. . & Richards, R. . A general two-site solution for the chemical exchange produced dependence of T2 upon the carr-Purcell pulse separation. *J. Magn. Reson.* **6**, 89–105 (1972).

Reviewer #2 (Remarks to the Author):

#2-1. The manuscript of Imai et al. presents a nice combined structural and dynamics investigation about how conformational rearrangements of a viral RNA enable interaction with a host initiation factor. The data presented are clear and support the conclusions described in the text. The integration of structure and dynamics derived from NMR experiments provides more insight than the cryo-EM structure alone would have. That said, the results presented are quite detailed and technical, and do not really benefit from the short format of a Nature Communication. The writing, especially the NMR section, is very technical in nature. For a journal like Nature Communications it should be fit for a broader audience, possibly by including a short, more general description of each technique and what it is measuring in practical terms for each experiment. In its current form, I do not think it is appropriate for publication in Nature Communications, and probably should be submitted to a more technical, longer format journal. That said, I offer some suggestions for them to improve the manuscript.

Thank you for taking the time to review our manuscript. We appreciate your positive comments about our study. We also appreciate your constructive comments about the technical nature of the writing and agree that we could make the manuscript more accessible to a broader audience. In response to your comments, we have shortened the Introduction part as described in **#2-2** below, while including a more general description of each technique used in the NMR section, as well as what each experiment is measuring in practical terms, as follows. We hope these revisions have made the manuscript accessible to a wider readership.

Results

P. 10, lines 176-179

To this end, we employed [u-²H, Ade-¹H2, ¹H8, ¹³C8}, Gua-¹H8, ¹³C8}] isotope labeling of RNA samples, and used it in combination with ¹H-¹³C aromatic transverse relaxation optimized spectroscopy (TROSY)²¹ (Supplementary Fig. 5), which enables observation of NMR signals that reflect on the structure and dynamics of RNA in high sensitivity.

P. 10, lines 182-189

The R_{ex} values obtained from the ^{13}C single quantum (SQ) relaxation dispersion experiment, which is the chemical exchange contribution to the transverse relaxation rate R_2 , reflect the degree of the structural dynamics in the μs - ms timescale that is critical for many RNAs to function²². The R_{ex} analyses of the isolated J domain was then conducted in a base-specific manner, in the absence of eIF4G^{HEAT1} at 30°C (Fig. 3a). As a result, G701, A704, G718, and G723 exhibited R_{ex} values larger than 10 s^{-1} , whereas G702 and A719 could not be quantitatively analyzed because their larger degrees of dynamics hampered the quantification of the signal intensities, indicating that the bases are in conformational exchange processes in a μs - ms timescale.

P. 12, lines 203-206

The nucleobases of A688, G690, A691, G693, A775, and G777 exhibited R_{ex} values larger than 10 s^{-1} , whereas the nucleobases of A687, A771, A772, A773, and A774 had moderate R_{ex} values between 5 and 10 s^{-1} , indicating that the bases are in conformational exchange processes in a μs - ms timescale.

P. 12, lines 213-215

To further characterize the dynamics observed for the StA_{SL} domain, we investigated the temperature dependence of the ^1H - ^{13}C aromatic TROSY signals, which reflects the change in conformational equilibrium upon changes in temperature.

P. 13, lines 220-224

We then analyzed the ^{13}C SQ relaxation dispersion profiles of the StA_{SL} domain, which are the NMR pulse frequency (ν_{CPMG}) dependence of the effective R_2 including the chemical exchange

contribution, $R_{2,\text{eff}}$. These profiles provide insights into exchange processes in terms of the exchange rate, k_{ex} , the relative populations of the exchanging states, p , and the difference in chemical shift, $\Delta\omega^{23}$.

P. 15, lines 263-267

First, the U778C variant was designed. U778 is base-paired with A691 in the NMR structure in the free-state, whereas it forms the G-U wobble base pair with G690 in the complex with eIF4G^{HEAT1} (Fig. 2). In the ¹H-¹H NOESY spectrum at 10°C, where cross peaks are observed when two imino protons (¹H_{imino}) are in spatial proximity (< 5 Å), cross peaks between the imino protons of U780, G689, and G690 were only observed for the U778C variant (Fig. 5a).

References

21. Meissner, A. & Sørensen, O. W. The role of coherence transfer efficiency in design of TROSY-type multidimensional NMR experiments. *J. Magn. Reson.* **139**, 439–42 (1999).
22. Marušič, M., Schlagnitweit, J. & Petzold, K. RNA Dynamics by NMR Spectroscopy. *Chembiochem* **20**, 2685–2710 (2019).
23. Weininger, U., Respondek, M. & Akke, M. Conformational exchange of aromatic side chains characterized by L-optimized TROSY-selected ¹³C CPMG relaxation dispersion. *J. Biomol. NMR* **54**, 9–14 (2012).

#2-2. The authors must frame their work in a wider context of previously published research. This is particularly evident in the repeated description of eIF4G as lacking intrinsic RNA binding (specific locations in text noted below), an activity previously shown for yeast and mammalian eIF4G (for example, Berset et al. RNA 2003, DOI: 10.1261/rna.5380903; Yanagiya et al. Mol Cell Biol 2009, DOI: 10.1128/MCB.01187-08). The latter reference maps RNA binding activity to a region of mammalian eIF4G included in the construct used here.

We agree with the reviewer's advice to properly cite the previous literature, and apologize for the incorrect description on the RNA binding activity of eIF4G. As the reviewer pointed out, eIF4G does have the RNA binding activity, and we have removed all the sentences that incorrectly describe the absence of the RNA binding activity of eIF4G, as follows.

Abstract

P. 3, lines 39-43

One prominent example is the IRES from encephalomyocarditis virus (EMCV), which interacts with the eukaryotic translation initiation factor 4G (eIF4G), thus recruiting the ribosomal 40S subunit to the initiation codon. The specific capture of eIF4G by the IRES is therefore imperative for viral replication, although the structural mechanism by which the viral RNA commandeers the host protein, ~~which is not an intrinsic RNA binder,~~ has remained unclear.

P. 4, lines 56-57

These results provide mechanistic and functional insights into the sophisticated strategy that the viral RNA employs to hijack a host protein ~~that lacks an intrinsic RNA binding site.~~

Introduction

P. 5, lines 81-86

~~Using biochemical assays, several interaction sites on the J-K-St or eIF4G^{HEAT1} have been reported¹⁰⁻¹². Although eIF4G^{HEAT1} reportedly binds to cellular RNAs^{13,14}, the interaction between J-K-St and eIF4G^{HEAT1} is stronger and more sequence- and/or structure specific¹⁵, suggesting that J-K-St has evolved to specifically capture eIF4G^{HEAT1}. However, the structural mechanism by which the J-K-St region specifically recognizes eIF4G^{HEAT1}, ~~which functions as the protein scaffold in the translation initiation factor complex by mediating multiple protein-protein interactions and is not~~~~

~~an intrinsic RNA binder *per se*, has remained largely unknown. As eIF4G^{HEAT1} is not an RNA-binding protein, it does not harbor an intrinsic RNA-binding site, where J-K-St would compete with the intrinsic RNA molecules.~~

P. 5, lines 89-91

However, eIF4G^{HEAT1} lacks clusters of such residues ~~serving as interfaces with RNA~~¹² (Supplementary Fig. 1), which is corroborated by the prediction of RNA binding residues in eIF4G^{HEAT1} from the structure and/or sequence analyses failed to identify an no RNA binding interface^{19,20}, ~~corroborating that eIF4G^{HEAT1} is not configured as an RNA-binding protein.~~

P. 7, lines 112-113

Together, our study highlights the finding that the EMCV IRES has evolved to use the two eIF4G^{HEAT1} binding domains, each tailored for the two small patches on the target protein, ~~which lacks interfaces for single-site interactions.~~

P. 7, lines 116-118

These mechanistic and functional insights illuminate the elegant strategy employed by the viral RNA to capture the host protein, ~~which does not harbor an intrinsic RNA-binding site~~, for efficient replication.

Discussion

P. 22, lines 382-383

EMCV IRES enables this ~~uninvited~~ interaction by using the two stem domains, J and St, each tailored for the two small patches on the target protein.

Furthermore, we have described about eIF4G's RNA binding activity, referring to the papers that the reviewer introduced. Importantly, although several mRNAs are reportedly bound to eIF4G, there is a critical distinction between the nonspecific interactions of eIF4G with mRNA and the stronger and specific interactions with various IRESs. Indeed, single-molecule fluorescence assays and RIP-seq experiments indicate that eIF4G orthologs bind nonspecifically and dynamically with mRNA, whereas the interactions of eIF4G with IRESs are structure- and/or sequence specific (Friedrich *et al.*, *Nucl Acids Res* (2022) 50:5424-5442, doi: 10.1093/nar/gkac342, and references therein). Therefore, the EMCV IRES should possess structural characteristics that could dictate the distinctive interactions with the HEAT1 domain of eIF4G among the other RNA molecules, which should be important for the hijacking of the translational machinery of the host cell by the viral RNA. We thus added the following description to the Introduction section, as follows.

Introduction

P. 5, lines 82-91

Although eIF4G^{HEAT1} reportedly binds to cellular RNAs^{13,14}, the interaction between J-K-St and eIF4G^{HEAT1} is stronger and more sequence- and/or structure specific¹⁵, suggesting that J-K-St has evolved to specifically capture eIF4G^{HEAT1}. However, the structural mechanism by which the J-K-St region specifically recognizes eIF4G^{HEAT1} has remained largely unknown. Typical RNA-binding proteins use interfaces where positively-charged residues such as arginine or lysine, and/or aromatic residues such as tyrosine or phenylalanine, are clustered and aligned for the specific recognition of the target RNA molecules¹⁶⁻¹⁸. However, eIF4G^{HEAT1} lacks clusters of such residues¹² (Supplementary Fig. 1), which is corroborated by the prediction of RNA binding residues in eIF4G^{HEAT1} from the structure and/or sequence analyses identify no RNA binding interface^{19,20}.

References

13. Yanagiya, A. *et al.* Requirement of RNA Binding of Mammalian Eukaryotic Translation Initiation Factor 4GI (eIF4GI) for Efficient Interaction of eIF4E with the mRNA Cap. *Mol. Cell*.

- Biol.* **29**, 1661–1669 (2009).
14. Berset, C., Zurbriggen, A., Djafarzadeh, S., Altmann, M. & Trachsel, H. RNA-binding activity of translation initiation factor eIF4G1 from *Saccharomyces cerevisiae*. *RNA* **9**, 871–80 (2003).
 15. Friedrich, D., Marintchev, A. & Arthanari, H. The metaphorical swiss army knife: The multitude and diverse roles of HEAT domains in eukaryotic translation initiation. *Nucleic Acids Res.* **50**, 5424–5442 (2022).
 16. Jones, S., Daley, D. T., Luscombe, N. M., Berman, H. M. & Thornton, J. M. Protein-RNA interactions: a structural analysis. *Nucleic Acids Res.* **29**, 943–54 (2001).
 17. Corley, M., Burns, M. C. & Yeo, G. W. How RNA-Binding Proteins Interact with RNA: Molecules and Mechanisms. *Mol. Cell* **78**, 9–29 (2020).
 18. Krüger, D. M., Neubacher, S. & Grossmann, T. N. Protein-RNA interactions: structural characteristics and hotspot amino acids. *RNA* **24**, 1457–1465 (2018).
 19. Li, S., Yamashita, K., Amada, K. M. & Standley, D. M. Quantifying sequence and structural features of protein-RNA interactions. *Nucleic Acids Res.* **42**, 10086–98 (2014).
 20. Yan, J. & Kurgan, L. DRNApred, fast sequence-based method that accurately predicts and discriminates DNA- and RNA-binding residues. *Nucleic Acids Res.* **45**, e84 (2017).

#2-3. There is also an opportunity to compare their structure to previous structural information from biochemical experiments that identified sites of interaction in this complex on both the RNA and protein (Clark et al. *J Virol* 2003, DOI: 10.1128/JVI.77.23.12441-12449.2003; Kolupaeva et al. *Mol Cell Biol* 2003, DOI: 10.1128/MCB.23.2.687-698.2003), papers that are relevant but not mentioned in the introduction or discussion.

We appreciate the reviewer’s suggestion to compare the cryo-EM structure of the J-K-St/eIF4G^{HEAT1}/eIF4A complex to previous structural information from biochemical experiments. The former paper (Clark *et al. J Virol* 2003, DOI: 10.1128/JVI.77.23.12441-12449.2003) used IRES-dependent cell selection system and identified that A704, C705, G723, and A724 in the J domain are important for the efficient IRES-dependent translation initiation (Fig. 1 of *J. Virol.* 77:12441, [Redacted]).

In our cryo-EM structure, all these residues are directly contacting with eIF4G^{HEAT1} (Fig. R3). Of these, base moieties of A704 and A724 are directly contacting with eIF4G^{HEAT1}, whereas those of C705 and G723 are pointing away from the interface, and contacts with the sugar or phosphate moieties. The cryo-EM structure

provided direct information on the interaction in a base-specific manner, distinguishing the contributions of these bases on the direct interaction with eIF4G^{HEAT1}.

Fig. R3 Mapping of the bases identified to be important for the function of EMCV IRES in *J. Virol.* 77:12441

The latter paper (Kolupaeva *et al. Mol Cell Biol* 2003, DOI: 10.1128/MCB.23.2.687-698.2003) used chemical and enzymatic footprinting as well as directed hydroxyl radical probing to determine the proximities between Fe(II)-BABE tags introduced onto eIF4G^{HEAT1} and EMCV IRES, and mapped the interaction onto the crystal structure of eIF4G^{HEAT1} and the secondary structure of J-K-St (Fig. 6, panels C and D of Kolupaeva *et al. Mol Cell Biol* (2003) 23:687, [Redacted]). As a result, they reported that the Fe(II)-BABE tags introduced simultaneously to Cys820, Cys822, Cys844 are close to G775-U778 region in J-K-St, the tag introduced Cys830 is close to regions A699-A700, U713-U715, and A724-C726 regions, and that the tag introduced Cys930 is close to U713-U715 region, respectively. From these results, the authors proposed the orientation of the HEAT1 domain on EMCV IRES.

[Redacted]

When we map these residues-bases proximity information on the cryo-EM structure of J-K-St/eIF4G^{HEAT1}/eIF4A complex, the residues that were modified by the Fe(II)-BABE tags on eIF4G^{HEAT1} are relatively close to the regions in J-K-St that are cleaved by the hydroxyl radical formed by the tags (Fig. R4). Furthermore, the relative orientation of the HEAT1 domain proposed in Kolupaeva *et al. Mol Cell Biol* (2003) 23:687, i.e., the N-terminus towards the J1 helix (termed lower stem of the St domain in the current study) and the C-terminus towards the J domain, corresponded to that observed in the cryo-EM structure. Our cryo-EM structure provided residue specific information, identifying the surface on eIF4G^{HEAT1} directly interacting with J-K-St.

Fig. R4 Mapping of the proximity pair identified in *Mol Cell Biol* (2003) 23:687

The proximity pair between Fe(II)-BABE tags introduced onto the HEAT1 domain and cleaved region in J-K-St by hydroxy radical formed by the tag, identified in *Mol Cell Biol* (2003) 23:687, are colored red on the cryo-EM structure of the J-K-St/eIF4G^{HEAT1}/eIF4A complex. Note that the residue numbers of eIF4G^{HEAT1} in the current study are smaller by one than the ones reported in *Mol Cell Biol* (2003) 23:687, because of the eIF4G1 isoforms referring to (Uniprot ID Q04637-8 in *Mol Cell Biol* (2003) 23:687, Q04637-1 in the current study) are different, although the sequences of the HEAT1 domain are identical.

(left) Cys 819, Cys821, and Cys847 on eIF4G^{HEAT1} and bases 775-778 in J-K-St

(middle) Cys 829 on eIF4G^{HEAT1} and bases 699-700, 724-726, and 713-715 in J-K-St

(right) Cys929 on eIF4G^{HEAT1} and bases 713-715 in J-K-St

There are also mutational studies on HEAT1 domain to identify the surface that interacts with EMCV IRES, which are summarized in the paper that reported the x-ray crystal structure of the HEAT1 domain of human eIF4GII (Marcotrigiano *et al., Mol Cell.* 2001 7:193-203. doi: 10.1016/s1097-2765(01)00167-8, ref 13 in the

original version (now ref 12), Fig. 6A [Redacted]), proposing that a molecular surface area including basic residues Arg814, Lys820, Arg834, and Lys835 is involved in the interaction with EMCV IRES. In the cryo-EM structure of the J-K-St/eIF4G^{HEAT1}/eIF4A complex, these residues are located on the interface with the St domain (Fig. R5). The cryo-EM structure elucidated that there is another surface area that directly interacts with the J domain of J-K-St, that has not been reported previously.

[Redacted]

Fig. R5 Mapping of the interface residues identified in *Mol Cell* (2001) 7:193-203

The residues on eIF4G^{HEAT1} identified to be involved in the interaction with J-K-St by the mutational study (Marcotrigiano *et al.*, *Mol Cell*. 2001 7:193-203) are mapped onto the cryo-EM structure of the J-K-St/eIF4G^{HEAT1}/eIF4A complex in red. Note that the residue number is different by six from the numbers in the previous literature, due to the differences in the homolog of eIF4G targeted in the study. These residues are within, or neighboring to, the interface with the St domain.

According to the reviewer's suggestion, we cited these papers in Introduction, as follows.

Introduction

P. 5, lines 81-86

Using biochemical assays, several interaction sites on the J-K-St or eIF4G^{HEAT1} have been reported¹⁰⁻¹². Although eIF4G^{HEAT1} reportedly binds to cellular RNAs^{13,14}, the interaction between J-K-St and eIF4G^{HEAT1} is stronger and more sequence- and/or structure specific¹⁵, suggesting that J-K-St has evolved to specifically capture eIF4G^{HEAT1}. However, the structural mechanism by which the J-K-St region specifically recognizes eIF4G^{HEAT1} has remained largely unknown.

References

10. Clark, A. T., Robertson, M. E. M., Conn, G. L. & Belsham, G. J. Conserved nucleotides within the J domain of the encephalomyocarditis virus internal ribosome entry site are required for activity and for interaction with eIF4G. *J. Virol.* **77**, 12441–9 (2003).
11. Kolupaeva, V. G., Lomakin, I. B., Pestova, T. V & Hellen, C. U. T. Eukaryotic initiation factors 4G and 4A mediate conformational changes downstream of the initiation codon of the encephalomyocarditis virus internal ribosomal entry site. *Mol. Cell. Biol.* **23**, 687–98 (2003).
12. Marcotrigiano, J. *et al.* A conserved HEAT domain within eIF4G directs assembly of the translation initiation machinery. *Mol. Cell* **7**, 193–203 (2001).
13. Yanagiya, A. *et al.* Requirement of RNA Binding of Mammalian Eukaryotic Translation Initiation Factor 4GI (eIF4GI) for Efficient Interaction of eIF4E with the mRNA Cap. *Mol. Cell. Biol.* **29**, 1661–1669 (2009).
14. Berset, C., Zurbriggen, A., Djafarzadeh, S., Altmann, M. & Trachsel, H. RNA-binding activity of translation initiation factor eIF4G1 from *Saccharomyces cerevisiae*. *RNA* **9**, 871–80 (2003).
15. Friedrich, D., Marintchev, A. & Arthanari, H. The metaphorical swiss army knife: The multitude and diverse roles of HEAT domains in eukaryotic translation initiation. *Nucleic Acids Res.* **50**, 5424–5442 (2022).

Furthermore, we added the above comparison of the cryo-EM complex structure to the previous structural information from biochemical experiments in Discussion and referred to Supplementary Fig. 15, as follows. Thanks to the suggestion, we wish we now successfully frame our work in a wider context of previously published research.

Discussion

P. 21, lines 363-366

In this study, we visualized the three-dimensional structure of the J-K-St/eIF4G^{HEAT1}/eIF4A complex, by cryo-EM single particle analysis (Fig. 1 and Supplementary Fig. 3). This structure not only corroborated the previous biochemical reports on the interaction sites on the J-K-St and eIF4G^{HEAT1} (Supplementary Fig. 15)¹⁰⁻¹², but also revealed key features on the residue-specific interactions and conformational changes of J-K-St, upon interaction with eIF4G^{HEAT1}.

References

10. Clark, A. T., Robertson, M. E. M., Conn, G. L. & Belsham, G. J. Conserved nucleotides within the J domain of the encephalomyocarditis virus internal ribosome entry site are required for activity and for interaction with eIF4G. *J. Virol.* **77**, 12441–9 (2003).
11. Kolupaeva, V. G., Lomakin, I. B., Pestova, T. V & Hellen, C. U. T. Eukaryotic initiation factors 4G and 4A mediate conformational changes downstream of the initiation codon of the encephalomyocarditis virus internal ribosomal entry site. *Mol. Cell. Biol.* **23**, 687–98 (2003).
12. Marcotrigiano, J. *et al.* A conserved HEAT domain within eIF4G directs assembly of the translation initiation machinery. *Mol. Cell* **7**, 193–203 (2001).

Supplementary Fig. 15 | Comparison with the previous biochemical assays on the interaction between eIF4G^{HEAT1} and J-K-St.

(a) Mapping of the J domain bases identified to be involved in the IRES-dependent translation in the previous literature¹⁰. The bases in the J domain, A704, C705, G723, and A724, that are shown to be important in the IRES-dependent translation¹⁰ is mapped on the cryo-EM structure of the J-K-St/eIF4G^{HEAT1}/eIF4A complex. These bases are on the interface of the eIF4G^{HEAT1} in the structure, corroborating that these bases play important roles in the interaction with eIF4G^{HEAT1}.

(b) Mapping of the proximity regions identified by directed hydroxyl radical cleavage assay in the previous literature¹¹. The residues on the eIF4G HEAT1 domain that were labeled with Fe(II)-BABE tags and the regions cleaved by the hydroxyl radical generated from the tags¹¹ are mapped on the cryo-EM structure of the J-K-St/eIF4G^{HEAT1}/eIF4A complex. (left) Cys 819, Cys821, and Cys847 on eIF4G^{HEAT1} and bases 775-778 in J-K-St, (middle) Cys 829 on eIF4G^{HEAT1} and bases 699-700, 724-726, and 713-715 in J-K-St, (right) Cys929 on eIF4G^{HEAT1} and bases 713-715 in J-K-St. The structure corresponds to the relative orientation of the eIF4G^{HEAT1} to J-K-St identified by the directed hydroxyl radical cleavage assay, i.e., the N-terminus towards St domain and C-terminus towards J domain.

(c) Mapping of the mutational assay on the eIF4G HEAT1 domain in the previous literature¹². The residues on eIF4G^{HEAT1} previously identified to be involved in the interaction with J-K-St by the

mutational study¹² are mapped onto the cryo-EM structure of the J-K-St/eIF4G^{HEAT1}/eIF4A complex in red. These residues are within, or neighboring to, the interface with the St domain, corroborating that these residues play important roles in the interaction with J-K-St.

#2-4. One concern I have with the structure is that for eIF4A, the PDB validation report lists 63% of the residues as having poor fit to the EM map – I think the authors should justify their inclusion of these residues or model only the portion of the protein that fits into the density.

The poor fitting of eIF4A to the post-processed map at the given threshold value is due to the flexibility of the subunit in the complex, particularly in the N-terminal half of the helicase lobes, which caused the low-resolution EM map at the region. By applying lowpass filtering to the map and fitting the starting model of eIF4A (from the human 48S translational initiation complex) with self-local distance restraints, we have placed the model of eIF4A in the ternary complex with a reasonable fit to the density at 5-6 Å resolutions. The entire model of the ternary complex helped us to clearly illustrate the binding site on the eIF4G^{HEAT1} domain to the J-K-St RNA (Fig. 1d, shown below for reference) as well as the relatively less conformational changes in the eIF4A/eIF4G^{HEAT1} subunits (Supplementary Fig. 4a, shown below for reference). However, the detailed structure of eIF4A in the complex is not our focus in the manuscript, and thus we further processed the data with a focused refinement with a local mask excluding eIF4A and assigned a model including only J-K-St and eIF4G^{HEAT1} (identical to the structures in the model of the full complex). We have deposited the focused model and map to PDB/EMDB as new entries 8J7R/EMD-36046, and used the model in most of the figures. Accordingly, we have revised the figures of the cryo-EM workflow in Supplementary Fig. 3 (in the revised manuscript) and the cryo-EM statistics in Supplementary Table 1, as shown below. We have also modified the method section, as follows.

Methods

P. 32, lines 562-575

Image processing was performed with RELION-3.1^{39,40}. Specimen movement was corrected using MotionCorr⁴¹, the contrast transfer function (CTF) parameters were estimated using Gctf⁴², and images showing substantial ice contamination, abnormal backgrounds, or poor Thon rings were discarded. Particles were initially picked with Gautomatch (<https://www.mrc-lmb.cam.ac.uk/kzhang/Gautomatch/>) without templates, and then the picked particles were subjected to 2D classification. Five representative class averages were selected as templates for Gautomatch to pick particles from all the micrographs collected on JEM-Z320FHC. The auto-picked particles were extracted with rescaling from 300 x 300 to 100 x 100 pixel images and subjected to two clean-up rounds of 2D classification. An initial reference map was generated in RELION, and then used as the reference for the 3D classification of the cleaned-up particles into four classes, among which one was selected and used to re-extract the corresponding particles into 200 × 200 pixel images. Subsequent 3D refinement with C1 symmetry, CTF refinement, and Bayesian polishing yielded a map at 3.8 Å resolution. Further 3D refinement by local alignment with a mask on a better-ordered region yielded a focused map at 3.7 Å resolution (Supplementary Fig. 3).

P. 33, lines 587-591

The models of the proteins and RNA were combined, and iterative cycles of real-space refinement in PHENIX⁴⁶ with secondary structure restraints and manual adjustments were performed to the whole complex map, yielding the cryo-EM model of the ternary complex. The final model of J-K-St/eIF4G^{HEAT1} was obtained by further refinement in PHENIX to the focused map. The refinement statistics are summarized in Supplementary Table 1.

Fig. 1 | Cryo-EM structure of the J-K-St/eIF4G^{HEAT1}/eIF4A complex

(a) Schematic representation of the translation initiation complex formed by EMCV IRES. The J-K-St region directly interacts with the scaffolding protein eIF4G of the host cell, leading to the recruitments of eIF4A, eIF3, eIF2, and the ribosomal 40S subunit to the initiation codon. (b) Domain organization of the full-length eIF4G and the eIF4G^{HEAT1} construct. (c) NMR structure of J-K-St in the free-state⁹ (PDB ID: 2NBX). (d) Overall structure of the J-K-St/eIF4G^{HEAT1}/eIF4A ternary complex, overlaid with the lowpass-filtered cryo-EM density map. (e) Focused cryo-EM map (left) and the structure (right) of J-K-St/eIF4G^{HEAT1}. The J, K, and St domains of J-K-St are labeled along with the A_{SL} domain at the junction. Mg²⁺ ions are shown as black spheres. (f and g) Electrostatic potential representations of the surface of eIF4G^{HEAT1} at the interface between the J domain (f) and the St domain (g), contoured from -5 kT/e (red) to $+5$ kT/e (blue). The residues of the J-K-St in contact with the eIF4G^{HEAT1} are shown as sticks. The labels of the extruded residues, A724 on the J domain (f) and G777 on the St domain (g), are highlighted with boxes. (h) Comparison of the domain orientations. The structures of J-K-St in the free-state and in complex with eIF4G^{HEAT1}/eIF4A are shown as surface and ribbon-and-stick representations, respectively. The J, K, St, and A_{SL} domains are colored green, blue, and orange, and red, respectively. The structures are aligned with the K domain, and changes in the J and St domain orientations upon complex formations are shown as black arrows.

Supplementary Fig. 4 | Analyses of the J-K-St/eIF4G^{HEAT1}/eIF4A structure

(a) Structural alignment of the eIF4G^{HEAT1}/eIF4A complex in the absence and presence of J-K-St. The human eIF4G^{HEAT1} (brown) / human eIF4A (cyan) dimer from the J-K-St/ eIF4G^{HEAT1}/eIF4A ternary complex (this study) is overlaid with the yeast eIF4G^{HEAT1} (yellow) / yeast eIF4A (violet) dimer (PDB ID: 2VSO). Even though the primary sequences are different, with sequence identities of 64.2% for eIF4A and 28.8% for eIF4G^{HEAT1}, the r.m.s.d. value of the two dimers is as low as 2.4 Å, indicating that the structural rearrangement upon the interaction with J-K-St is small.

(b) eIF4G^{HEAT1} residues interacting with the J domain in the J-K-St/eIF4G^{HEAT1}/eIF4A complex. Cryo-EM map of the J domain from J-K-St is shown as a purple surface, where the nucleobase extruding from the J domain, A724, is labeled, and the model the J-K-S RNA is shown in stick representation. A part of the eIF4G^{HEAT1} backbone structure is shown in cartoon representation (brown), and the side chains at the RNA-protein interface are shown in stick representation.

(c) eIF4G^{HEAT1} residues interacting with the St domain in the J-K-St/eIF4G^{HEAT1}/eIF4A complex. Cryo-EM map of the St domain from J-K-St is shown as a purple surface, where the nucleobase extruding from the St domain, G777, is labeled, and the model the J-K-S RNA is shown in stick representation. A part of the eIF4G^{HEAT1} backbone structure is shown in cartoon representation (brown), and the side chains at the RNA-protein interface are shown in stick representation. Lys826, whose side chain is inserted within the cavity formed by the flipping-out of G777, is highlighted by a box.

Supplementary Fig. 3 | Cryo-electron microscopy analyses

- (a) Size exclusion chromatography profile of the J-K-St/eIF4G^{HEAT1}/eIF4A ternary complex. The positions corresponding to the elutions of the complex and free components are labeled and indicated by arrows.
- (b) SDS-PAGE analysis of the peak fraction corresponding to the ternary complex. Proteins in the gel were stained with Coomassie Brilliant Blue G-250.
- (c) Representative cryo-EM micrograph of the vitrified J-K-St/eIF4G^{HEAT1}/eIF4A complex. Green circles indicate examples of individual particles. Scale bar is 50 nm.
- (d) Selected 2D-class averages of the ternary complex.
- (e) Low-pass-filtered Cryo-EM maps obtained from the pilot data of the ternary complexes containing WT J-K-St (left, grey) or St-extended J-K-St (right, green), collected on JEM-Z300CF. Red arrows indicate the region of an extra density observed only in the St-extended J-K-St/eIF4G^{HEAT1}/eIF4A ternary complex.
- (f) Cryo-EM data-processing workflow for the St-extended J-K-St/eIF4G^{HEAT1}/eIF4A ternary complex collected on JEM-Z320FHC.
- (g and h) The refined cryo-EM map of the whole complex (g) and the refined map with local alignment and a mask (inset) focused on the region of J-K-St/eIF4G^{HEAT1} (h).
- (i and j) Angular distribution plot of all particles that contributed to the refined map of the whole complex (i) and the J-K-St/eIF4G^{HEAT1}-focused map (j).
- (k and l) Gold-standard Fourier shell correlation (FSC) curves for the whole complex map (k) and the J-K-St/eIF4G^{HEAT1}-focused map (l) after correction for masking effects (black line). The phase randomized FSC curves are colored red. The map resolutions were estimated based on the FSC = 0.143 criterion.
- (m and n) Local resolutions for the whole complex map (m) and the focused map (n) were calculated using ResMap⁴⁸.
- (o) Cryo-EM densities and refined models for selected regions at the St/A_{SL} domain (lower left) and the J domain (lower right) of J-K-St RNA, eIF4G^{HEAT1} (upper right), and eIF4A (upper left) in the ternary complex.
- (p and q) Cross-validation FSC curves of the refined models versus the final maps. The correlation is above 0.5 up to resolutions of 3.8 Å and 3.7 Å for the ternary complex (p) and the focused region of J-K-St/eIF4G^{HEAT1} (q), respectively.

Supplementary Table 1 | Cryo-EM data collection and refinement statistics.

	J-K-St/eIF4G ^{HEAT1} /eIF4A	J-K-St/eIF4G ^{HEAT1} (Focused)
EMDB ID	EMD-35041	EMD-36046
PDB ID	8HUI	8J7R
Data collection		
Microscope	JEM-Z320FHC	
Detector	K2 Summit	
Pixel size (Å)	0.765	
Defocus range (µm)	-1.0 to -2.5	
Voltage (kV)	300	
Total electron dose (e ⁻ Å ⁻²)	68.3	
Reconstruction		
Final particle number	255,256	
Pixel size (Å)	1.1475	
Box size (pixels)	200	
Map resolution (Å) (FSC = 0.143)	3.76	3.70
Map sharpening B-factor (Å ⁻²)	-110	-132
Refinement		
Model composition		
Non-hydrogen atoms	7,143	4,034
Protein residues	621	238
Nucleotide residues	99	97
Water	1	1
Ligands	Mg: 3	Mg: 3
Map CC (mask)	0.78	0.76
Map CC (volume)	0.78	0.76
R.m.s. deviations		
Bond length (Å)	0.005	0.010
Bond angles (°)	1.012	1.340
Validation		
MolProbity score	2.2	2.2
All-atom clashscore	13.6	9.2
Ramachandran plot		
Outliers (%)	0.0	0.0
Allowed (%)	3.2	2.5
Favored (%)	96.8	97.5
Rotamer Outliers (%)	2.5	6.85
C-beta deviations (%)	0.0	0.0

#2-5. Finally, I think there are a few experiments that could help strengthen this paper and test their model, especially a functional assay. For example, to connect the conformational dynamics to biochemistry, how does the U778C mutation described affect binding of eIF4G to the RNA?

The U778C mutation severely abolished the affinity of the J-K-St to eIF4G^{HEAT1} with the K_d value larger than 10 μM, as evidenced by our ITC experiment conducted in response to the reviewer's comment (Fig. R6a). Since the U778C mutation stabilizes the excited conformations that favorably bind with eIF4G^{HEAT1}, one might expect that the U778C variant would have a similar, or even stronger, binding, as compared to the wild-type. However, this residue is on the interface with the eIF4G^{HEAT1} in the cryo-EM structure of the complex, forming cation-π stacking interaction with Lys826 of eIF4G^{HEAT1} (Fig. R6b). It is thus likely that, although the U778C mutation increases the population of the excited conformations, the binding energy obtained upon binding decreases due to the mutation, because cytosine features an NH₂ group on its pyrimidine ring structure that disperses the electron density on the pyrimidine ring, leading to a reduced electron density for forming cation-π interactions. The importance of Lys826 on the interaction with J-K-St has been previously reported (Marcotrigiano *et al.*, *Mol Cell*. 2001 7:193-203). In order to verify the contribution of Lys826 on the interaction, we mutated Lys826 of eIF4G^{HEAT1} to Ala, and investigated the interaction of this K826A variant with the J-K-St (Fig. R6a). As a result, the interaction is abrogated by the mutation, supporting the importance of the intermolecular interactions involving Lys826 on the binding of eIF4G^{HEAT1} to J-K-St. Thus, the effect of the U778C mutation on the conformational dynamics in the free state cannot be analyzed by the ITC, but can be observed by NMR spectroscopy that directly detects the dynamic properties of the biomolecules, not indirectly via the interaction with their binding partners.

Fig. R6 ITC experiments of the U778C variant of J-K-St

(a) ITC experiments of the J-K-St wild type and eIF4G^{HEAT1} wild-type (left), J-K-St U778C variant and eIF4G^{HEAT1} wild-type (middle), and J-K-St wild-type and eIF4G^{HEAT1} K826A variant (right), in the presence of 2 mM Mg²⁺. N.D., K_d value not determined.

(b) Close-up view of the U778-Lys826 interaction. The side chain of Lys826 of eIF4G^{HEAT1} is extruded to stack onto the pyrimidine ring of U778.

In response to the reviewer's suggestion, we have added a new subsection at the end of the Results section, and Supplementary Fig. 12 and 13, as follows. Thanks to these modifications, the effect of the U778C mutation on the binding to eIF4G^{HEAT1}, and the reason why the mutation decreases the binding, are now clear to the readership.

Results

P.18, lines 326-337

Functional relevance of the dynamics in the J and StA_{SL} domains

Next, we conducted ITC experiments to investigate whether the conformational dynamics observed for the two eIF4G^{HEAT1} binding sites on the J and StA_{SL} domains are critical for the interaction with eIF4G^{HEAT1} (Supplementary Fig. 12). Given that the U778C mutation increases the populations of the excited conformations, which possess the structural characteristics of J-K-St as in the complex structure, it might be expected that the J-K-St U778C variant would have a similar, or even stronger, affinity for eIF4G^{HEAT1} compared to the wild-type. However, the J-K-St U778C variant exhibited a decreased affinity for eIF4G^{HEAT1}, with a K_d value larger than 10 μ M, compared to 149 ± 12 nM for the wild-type (Supplementary Figs. 2a and 12b). This is likely because U778 is on the interface with eIF4G^{HEAT1} in the complex structure (Figs. 1g and 2a, and Supplementary Fig. 13), and mutating it might result in decreased free energy upon interaction with eIF4G^{HEAT1}, reducing the affinity for eIF4G^{HEAT1}.

Supplementary Fig. 12 | Correlation between the NMR-observed dynamics and function
 (a) Overlay of the $^1\text{H}8$ - $^{13}\text{C}8$ aromatic TROSY spectra of the J-K-St U778C (left), A700C (middle), and U780C (right) variants with the corresponding wild-type J-K-St. All spectra were acquired at 35°C and at the ^1H frequency of 900 MHz. Although the signals observed for J-K-St is not assigned, more than 80% of the signals observed for the variants are overlaid with those from the wild-type, indicating that the overall secondary structures of these variants are not disrupted by the mutation.
 (b) ITC experiments of the J-K-St variants. U778C, A700C, and U780C variants exhibited a lower affinity (K_d value larger than $10\ \mu\text{M}$) for eIF4G^{HEAT1}, compared to the wild-type ($K_d = 149 \pm 12\ \text{nM}$). N.D., not determined. The ITC experiments were conducted at least two times with similar results.

(c) *In vitro* translation assay. β -galactosidase assay using a human cell-derived *in vitro* coupled transcription/translation system (n=7 independent replicates). Total amounts of β -galactosidase translated from the EMCV IRES-mediated translation were quantified as luminescence emitted as a result of β -galactosidase reaction. Data are presented as mean \pm standard error of the mean (s.e.m.). Statistical significance was determined by two-tailed unpaired Student's t test.

Supplementary Fig. 13 | Interaction of U778 in J-K-St and Lys826 in eIF4G^{HEAT1}

(a) Close-up view of the U778-Lys826 interaction. The side chain of Lys826 of eIF4G^{HEAT1} is extruded to stack onto the pyrimidine ring of U778.

(b) The ITC experiment of the J-K-St wild-type and eIF4G^{HEAT1} K826A. N.D., not determined (K_d larger than 10 μM). The ITC experiment was conducted at least two times with similar results.

#2-6. The changes observed in this RNA by NMR for U778C, A700C, and U780C are not connected well to biochemical or functional experiments - how do these mutations affect IRES function, possibly using a luciferase reporter, or viral fitness in a replicon or full-length infection model? These insights could also give it broader appeal more suitable for Nature Communications, helping it to be more than just a technical structure paper.

We appreciate the reviewer's suggestion to connect the changes observed by NMR for A700C, U778C, and U780C mutations to functional experiments. In response, we have conducted the β -galactosidase assay as an alternative to the suggested luciferase reporter assay, by an *in vitro* translation assay using a human cell-free protein expression system (Mikami *et al.*, *Protein Expr. Purif.* (2008) 62:190-198. doi: 10.1016/j.pep.2008.09.002.) (Fig. R7). Our results demonstrated that all the variants, A700C, U778C, and U780C, showed reduced IRES activity compared to the wild-type.

Fig. R7 β -galactosidase assay

β -galactosidase assay using a human cell-derived *in vitro* coupled transcription/translation system ($n=7$ independent replicates). Total amounts of β -galactosidase translated from the EMCV IRES-mediated translation were quantified as luminescence emitted as a result of β -galactosidase reaction. Data are presented as mean \pm standard error of the mean (s.e.m.). Statistical significance was determined by two-tailed unpaired Student's t test.

The A700C and U780C variants were designed to suppress the dynamics in the J and StA_{SL} domains, respectively, observed in the NMR analyses. For the J domain, changing A700 to cytidine would allow the canonical Watson-Crick base pair formation with G723, which would suppress the plasticity of the J domain bulge G723-A724 to allow for the conformational change in the complex formation process (Fig. R8a). For the StA_{SL} domain, changing U780 to cytidine would stabilize the base pairing with G689, since the canonical Watson-Crick G-C base pair is more stable than the G-U wobble base pair, which would suppress the register shift toward the excited states required for the conformational change in the complex formation process (Fig. R8b). The NMR R_{ex} analyses of these isolated domain variants revealed that the dynamics in the μ s-ms timescale are indeed suppressed (Fig. R8), while the secondary structures of these isolated domains are not deteriorated by the mutation (Fig. R9). The ITC experiments revealed that A700C and U780C variants of J-K-St possess reduced affinity for eIF4G^{HEAT1} (Fig. R10). Importantly, both A700 and U780 are not directly interacting with eIF4G^{HEAT1} in the complex structure, making it possible to interpret that the effect of the mutation is not because the direct interaction with the eIF4G^{HEAT1} is perturbed, but is because it stabilized the ground conformation, which is not compatible for the interaction as is. Notably, the U778C mutation, which stabilizes the excited conformations, did not increase the affinity of J-K-St for eIF4G^{HEAT1}, because it weakens the key cation- π interaction of U778 and Lys826 in eIF4G^{HEAT}, as discussed in #2-5 above.

Fig. R8 ^{13}C Relaxation dispersion analyses of the variants ^{13}C R_{ex} analyses of the J domain A700C variant (a) and the StA_{SL} domain U780C variant (b). Compared to the wild-type, the R_{ex} values are decreased compared to the wild-type, indicating that the dynamics in the ms-ms timescale are suppressed.

Fig. R9 Comparison of the ^1H 8 - ^{13}C 8 aromatic TROSY

(a) Overlay of the ^1H 8 - ^{13}C 8 aromatic TROSY spectra of the isolated J domain A700C (top left), StA_{SL} domain U778C (top middle), and StA_{SL} domain U780C (top right) variants with the corresponding wild-type isolated domains. All spectra were acquired at 30°C and at the ^1H frequency of 800 MHz. Mutated variants and the wild-type are shown in red and black, respectively. The signals perturbed by the mutation are labeled in green, and mapped onto the secondary structure (bottom). Perturbed bases are located neighboring to the mutated bases, corroborating that overall secondary structures are not disrupted by the mutation.

Fig. R10 ITC experiments of the J-K-St variants

(a) Overlay of the ^1H - ^{13}C aromatic TROSY spectra of the J-K-St A700C (left), U778C (middle), and U780C (right) variants with the corresponding wild-type J-K-St. All spectra were acquired at 35°C and at the ^1H frequency of 900 MHz. Although the signals observed for J-K-St is not assigned, more than 80% of the signals observed for the variants are overlaid with those from the wild-type, indicating that the overall secondary structures of these variants are not disrupted by the mutation.

(b) Summary of the ITC experiments of J-K-St and its variants. While the wild-type J-K-St exhibited an affinity for eIF4G^{HEAT1} of 149 ± 12 nM, all the other variants, A700C, U778C, and U780C, exhibited lower (K_d value larger than 10 μM) affinity for eIF4G^{HEAT1}.

From the *in vitro* translation assay, it is demonstrated that all these three variants possess lower IRES functions compared to the wild-type (Fig. R7 in #2-6), indicating that the observed decrease in affinity between the isolated J-K-St and eIF4G^{HEAT1} domain in the ITC experiments, hence the dynamics observed in the NMR experiments, is related to the biological function of EMCV IRES. It is interesting to note that the observed decrease in the IRES function is not necessarily correlated with the amount of β -galactosidase translated, which suggest that the RNA binding affinity of the HEAT1 domain of eIF4G to the regions other than EMCV IRES J-K-St, or the other interactions among the translation initiation factors and the RNA, compensate the weakened interaction between the HEAT1 domain and J-K-St.

In summary, our study demonstrates that the dynamic properties of the EMCV IRES, as observed by NMR, are important for its biological function, i.e., initiating the IRES-dependent translation. The mutations introduced in this study, A700C and U780C, partly impair the IRES activity by perturbing the dynamic properties important for the interaction with eIF4G^{HEAT1}. The β -galactosidase assay provides a comprehensive understanding of the functional importance of these dynamic properties, making our study more than just a technical structure paper. Accordingly, we added Supplementary Figs. 12, 13, and 14, and modified the main text, as follows. Thanks to the constructive comment, we believe that these insights into the functional role of the EMCV IRES dynamics will give our study broader appeal to the readership of *Nature Communications*.

Results

P. 19, lines 326-359

Functional relevance of the dynamics in the J and StA_{SL} domains

Next, we conducted ITC experiments to investigate whether the conformational dynamics observed for the two eIF4G^{HEAT1} binding sites on the J and StA_{SL} domains are critical for the interaction with eIF4G^{HEAT1} (Supplementary Fig. 12). Given that the U778C mutation increases the populations of the excited conformations, which possess the structural characteristics of J-K-St as in the complex structure, it might be expected that the J-K-St U778C variant would have a similar, or even stronger,

affinity for eIF4G^{HEAT1} compared to the wild-type. However, the J-K-St U778C variant exhibited a decreased affinity for eIF4G^{HEAT1}, with a K_d value larger than 10 μ M, compared to 149 ± 12 nM for the wild-type (Supplementary Figs. 2a and 12b). This is likely because U778 is on the interface with eIF4G^{HEAT1} in the complex structure (Figs. 1g and 2a, and Supplementary Fig. 13), and mutating it might result in decreased free energy upon interaction with eIF4G^{HEAT1}, reducing the affinity for eIF4G^{HEAT1}. We then designed the A700C and U780C variants, to suppress the conformational dynamics in the J and StA_{SL} domains observed in the NMR analyses, respectively, without mutating the bases that directly interact with eIF4G^{HEAT1} (Supplementary Fig. 14). For the J domain, changing A700 to cytidine would allow for the canonical Watson-Crick base pair formation with G723, which would suppress the plasticity of the J domain bulge G723-A724 to allow for the conformational change in the complex formation process (Supplementary Fig. 14, a and b). For the StA_{SL} domain, changing U780 to a cytidine would stabilize the base pairing with G689, as the canonical Watson-Crick G-C base pair is more stable than the G-U wobble base pair, which would suppress the register shift toward the excited conformations required for the conformational change in the complex formation process (Supplementary Fig. 14, d and e). NMR R_{ex} analyses of these isolated domain variants revealed that the dynamics in the μ s-ms timescale are suppressed (Supplementary Fig. 14, c and f), while the secondary structures of these isolated domains are not altered by the mutation (Supplementary Fig. 14, a and d). The ITC experiments revealed that A700C and U780C variants of J-K-St possess reduced affinities for eIF4G^{HEAT1} (Supplementary Fig. 12, a and b), indicating that the suppression of the conformational dynamics to stabilize the ground conformation results in decreased affinities for eIF4G^{HEAT1}.

Finally, we sought to investigate the correlation between the dynamics and the biological function of IRES. To this end, we utilized the human cell-free translation system³⁰, where the EMCV IRES is used to initiate the translation of a model protein, β -galactosidase, by interacting with the translation initiation factors in the human cell (Supplementary Fig. 12c). As a result, it was

shown that introducing A700C, U780C, or U778C mutations in the full length EMCV IRES decreased the protein expression levels to 86.9 ± 3.2 , 89.5 ± 3.4 , and $72.3 \pm 3.4\%$, respectively, demonstrating that the conformational dynamics observed in the NMR analyses are related to the biological function of IRES.

Methods

P.38, lines 670-681

In vitro translation assay

A human cell-free expression system (TaKaRa) was used for transcription-coupled IRES-dependent translation of the reporter protein, β -galactosidase. The DNA construct containing the EMCV IRES and β -galactosidase (supplied as the positive control vector from TaKaRa) was used as a template to obtain A700C, U778C, and U780C variants by PCR-based mutagenesis. These wild-type and mutated DNA constructs were used as templates for *in vitro* T7 transcription-translation reactions according to the manufacturer's instructions, at the final concentration of 15 ng/ μ l, in a 10 μ l reaction scale. After incubation for 3 hrs at 32°C, EDTA was added to the final concentration of 25 mM. The amount of β -galactosidase expressed was quantified using the Luminescent β -galactosidase Detection Kit II (Clontech), according to the manufacturer's instructions. The luminescence was measured with an Envision 2104 multilabel plate reader (PerkinElmer). The experiments were repeated seven times and mean values and standard error are reported.

Supplementary Fig. 12 | Correlation between the NMR-observed dynamics and function

(a) Overlay of the ^1H - ^{13}C aromatic TROSY spectra of the J-K-St U778C (left), A700C (middle), and U780C (right) variants with the corresponding wild-type J-K-St. All spectra were acquired at 35°C and at the ^1H frequency of 900 MHz. Although the signals observed for J-K-St is not assigned, more than 80% of the signals observed for the variants are overlaid with those from the wild-type, indicating that the overall secondary structures of these variants are not disrupted by the mutation.

(b) ITC experiments of the J-K-St variants. U778C, A700C, and U780C variants exhibited a lower affinity (K_d value larger than $10 \mu\text{M}$) for eIF4G^{HEAT1}, compared to the wild-type ($K_d = 149 \pm 12 \text{ nM}$). N.D., not determined. The ITC experiments were conducted at least two times with similar results.

(c) *In vitro* translation assay. β -galactosidase assay using a human cell-derived *in vitro* coupled transcription/translation system (n=7 independent replicates). Total amounts of β -galactosidase translated from the EMCV IRES-mediated translation were quantified as luminescence emitted as a result of β -galactosidase reaction. Data are presented as mean \pm standard error of the mean (s.e.m.). Statistical significance was determined by two-tailed unpaired Student's t test.

Supplementary Fig. 13 | Interaction of U778 in J-K-St and Lys826 in eIF4G^{HEAT1}

(a) Close-up view of the U778-Lys826 interaction. The side chain of Lys826 of eIF4G^{HEAT1} is extruded to stack onto the pyrimidine ring of U778.

(b) The ITC experiment of the J-K-St wild-type and eIF4G^{HEAT1} K826A. N.D., not determined (K_d larger than $10 \mu\text{M}$). The ITC experiment was conducted at least two times with similar results.

Supplementary Fig. 14 | Suppression of the conformational dynamics by mutations

(a) Overlay of the ¹H8-¹³C8 aromatic TROSY spectra of the J domain A700C variant (red) with the wild-type (black). All spectra were acquired at 30°C and at the ¹H frequency of 800 MHz. The signals perturbed by the mutation are indicated with green labels.

(b) Mapping of the bases perturbed by the A700C mutation on the secondary structure. Perturbed bases are located neighboring to the mutated bases, corroborating that overall secondary structure is not disrupted by the mutation. A700C mutation is designed to form a base pair with G723, suppressing the conformational dynamics observed in the J domain bulge, without substituting the bases that directly interact with eIF4G^{HEAT1} (highlighted with purple lines).

(c) R_{ex} analyses of the A700C variant of the J domain. R_{ex} values observed for the J domain bases are largely suppressed by the mutation (see Fig. 3a for wild type), indicating that the conformational dynamics in the J domain are suppressed. Error bars indicate experimental errors derived from the

signal-to-noise ratio of each correlation, as written in Methods.

(d) Overlay of the ^1H - ^{13}C aromatic TROSY spectra of the StA_{SL} domain U780C variant (red) with the wild-type (black). All spectra were acquired at 30°C and at the ^1H frequency of 800 MHz. The signals perturbed by the mutation are indicated with green labels.

(e) Mapping of the bases perturbed by the U780C mutation on the secondary structure. Perturbed bases are located neighboring to the mutated bases, corroborating that overall secondary structure is not disrupted by the mutation. U780C mutation is designed to stabilize the base pair with G689, suppressing the register-shift observed in the upper stem of the StA_{SL} domain, without substituting the bases that directly interact with eIF4G^{HEAT1} (highlighted with purple lines).

(f) R_{ex} analyses of the U780C variant of the StA_{SL} domain. R_{ex} values observed for the StA_{SL} domain bases are largely suppressed by the mutation (see Fig. 3b for wild type), indicating that the conformational dynamics in the StA_{SL} domain are suppressed. Error bars indicate experimental errors derived from the signal-to-noise ratio of each correlation, as written in Methods.

Specific comments:

Abstract:

#2-7. Lines 6-7: "...which is not an intrinsic RNA binder" See comment above about previous data showing intrinsic RNA binding activity of eIF4G.

We have carefully considered the reviewer's comment regarding the intrinsic RNA binding activity of eIF4G, and have deleted this description from the Abstract section. Please refer to our response to comment **#2-2** above for details of the changes made.

#2-8. Line 8: "Here, we analyzed the three-dimensional structure of the complex formed by the EMCV IRES with eIF4G..." Consider editing to clarify that only a fragment of eIF4G was used in this structure, and that eIF4A was also present.

We would like to thank the reviewer for the suggestion to clarify the protein component used in the cryo-EM analyses in the Abstract. Accordingly, we edited the sentence in the Abstract, as follows. Thanks to the change, the components of the complex used in the cryo-EM analyses are now represented precisely.

Abstract

P.3, lines 44-46

Here, we analyzed the three-dimensional structure of the **ternary complex formed by the EMCV IRES with the HEAT1 domain fragment of eIF4G and eIF4A** by cryo-electron microscopy, and identified two distinct eIF4G-interacting sites on the J and St domains in the IRES.

#2-9. Line 10: "...to hijack a host protein that lacks an intrinsic RNA binding site." See comment above about previous data showing intrinsic RNA binding activity of eIF4G.

We have carefully considered the reviewer's comment regarding the intrinsic RNA binding activity of eIF4G, and have deleted this description from the Abstract section. Please refer to our response to comment **#2-2** above for details of the changes made.

Introduction:

#2-10. Paragraph 2, line 17: "...and is not an intrinsic RNA binder per se" See comment above about previous data showing intrinsic RNA binding activity of eIF4G.

We have carefully considered the reviewer's comment regarding the intrinsic RNA binding activity of eIF4G, and have deleted this description from the Introduction section. Please refer to our response to comment **#2-2** above for details of the changes made.

#2-11. Paragraph 2: the reasoning in the second half of this paragraph seems shaky to me. It's predicated on the assumption that eIF4G doesn't bind RNA, and visual inspection of the surface charge, a very qualitative measure, is offered as evidence to support that idea.

We appreciate the reviewer's comment on the reasoning of the difficulty for an RNA to target eIF4G^{HEAT1}. For reference, we have written as follows in the original version, which is deleted in the revised version.

As eIF4G^{HEAT1} is not an RNA-binding protein, it does not harbor an intrinsic RNA-binding site, where J-K-St would compete with the intrinsic RNA molecules. Typical RNA-binding proteins use interfaces where positively-charged residues such as arginine or lysine, and/or aromatic residues such as tyrosine or phenylalanine, are clustered and aligned for the specific recognition of the target RNA molecules (10–12). However, eIF4G^{HEAT1} lacks clusters of such residues serving as interfaces with RNA (13) (Supplementary Fig. 1a). Indeed, the prediction of RNA binding residues in eIF4G^{HEAT1} from the structure and/or sequence analyses failed to identify an RNA binding interface (14, 15), corroborating that eIF4G^{HEAT1} is not configured as an RNA-binding protein. Furthermore, in order to hijack and exploit the host translational system, the viral IRES RNA should bind to eIF4G^{HEAT1} without perturbing its innate function in translation; namely, recruiting eIF4A. To meet this condition, J-K-St should bind to eIF4G^{HEAT1} without competing with eIF4A, which further restricts the possible binding sites on eIF4G^{HEAT1}.

The reasoning here was not based on the assumption that eIF4G^{HEAT1} does not bind to RNA or our visual inspection of the surface charge, but came from the structural, or bioinformatic investigation of the HEAT1 domain. First, HEAT domains in general are known to interact with a variety of binding partners, both proteins and nucleic acids, which can lead to diverse binding surfaces and potentially obscure the specific RNA-binding surface (Friedrich *et al.*, *Nucl Acids Res* (2022) 50:5424-5442, doi: 10.1093/nar/gkac342). Second, since the HEAT domains exhibit low sequence conservation, it was challenging to identify conserved motifs or residues that may be involved in RNA binding based solely on sequence alignments (Marcotrigiano *et al.*, *Mol Cell*. 2001 7:193-203, doi: 10.1016/s1097-2765(01) 00167-8). Third, sequence and structure-based scoring of RNA-binding propensity did not predict any binding surface on the HEAT1 domain of eIF4G, supporting that there is

not a surface area that readily bind to RNA (Li *et al.*, *Nucleic Acids Res.* (2014) 42:10086–98 doi:10.1093/nar/gku681). Importantly, although cellular RNAs are reportedly bind to eIF4G, there is a critical distinction between the nonspecific interactions of eIF4G with mRNA and the stronger and specific interactions with various IRESs. Indeed, single-molecule fluorescence assays and RIP-seq experiments indicate that eIF4G orthologs bind nonspecifically and dynamically with mRNA, whereas the interactions of eIF4G with IRESs are structure- and/or sequence specific (Friedrich *et al.*, *Nucl Acids Res* (2022) 50:5424-5442, doi: 10.1093/nar/gkac342, and references therein). Therefore, the EMCV IRES should possess structural characteristics that could dictate the distinctive interactions with the HEAT1 domain of eIF4G among the other RNA molecules, which should be important for the hijacking of the translational machinery of the host cell by the viral RNA.

In response to the reviewer’s comment, we have modified this paragraph as follows.

Introduction

P.5, lines 81-95

Using biochemical assays, several interaction sites on the J-K-St or eIF4G^{HEAT1} have been reported^{10–12}. Although eIF4G^{HEAT1} reportedly binds to cellular RNAs^{13,14}, the interaction between J-K-St and eIF4G^{HEAT1} is stronger and more sequence- and/or structure specific¹⁵, suggesting that J-K-St has evolved to specifically capture eIF4G^{HEAT1}. However, the structural mechanism by which the J-K-St region specifically recognizes eIF4G^{HEAT1} has remained largely unknown. Typical RNA-binding proteins use interfaces where positively-charged residues such as arginine or lysine, and/or aromatic residues such as tyrosine or phenylalanine, are clustered and aligned for the specific recognition of the target RNA molecules^{16–18}. However, eIF4G^{HEAT1} lacks clusters of such residues¹² (Supplementary Fig. 1), which is corroborated by the prediction of RNA binding residues in eIF4G^{HEAT1} from the structure and/or sequence analyses identify no RNA binding interface^{19,20}. Furthermore, in order to hijack and exploit the host translational system, the viral IRES RNA should

bind to eIF4G^{HEAT1} without perturbing its innate function in translation; namely, recruiting eIF4A. To meet this condition, J-K-St should bind to eIF4G^{HEAT1} without competing with eIF4A, which further restricts the possible binding sites on eIF4G^{HEAT1}.

References

10. Clark, A. T., Robertson, M. E. M., Conn, G. L. & Belsham, G. J. Conserved nucleotides within the J domain of the encephalomyocarditis virus internal ribosome entry site are required for activity and for interaction with eIF4G. *J. Virol.* **77**, 12441–9 (2003).
11. Kolupaeva, V. G., Lomakin, I. B., Pestova, T. V & Hellen, C. U. T. Eukaryotic initiation factors 4G and 4A mediate conformational changes downstream of the initiation codon of the encephalomyocarditis virus internal ribosomal entry site. *Mol. Cell. Biol.* **23**, 687–98 (2003).
12. Marcotrigiano, J. *et al.* A conserved HEAT domain within eIF4G directs assembly of the translation initiation machinery. *Mol. Cell* **7**, 193–203 (2001).
13. Yanagiya, A. *et al.* Requirement of RNA Binding of Mammalian Eukaryotic Translation Initiation Factor 4GI (eIF4GI) for Efficient Interaction of eIF4E with the mRNA Cap. *Mol. Cell. Biol.* **29**, 1661–1669 (2009).
14. Berset, C., Zurbriggen, A., Djafarzadeh, S., Altmann, M. & Trachsel, H. RNA-binding activity of translation initiation factor eIF4G1 from *Saccharomyces cerevisiae*. *RNA* **9**, 871–80 (2003).
15. Friedrich, D., Marintchev, A. & Arthanari, H. The metaphorical swiss army knife: The multitude and diverse roles of HEAT domains in eukaryotic translation initiation. *Nucleic Acids Res.* **50**, 5424–5442 (2022).
16. Jones, S., Daley, D. T., Luscombe, N. M., Berman, H. M. & Thornton, J. M. Protein-RNA interactions: a structural analysis. *Nucleic Acids Res.* **29**, 943–54 (2001).
17. Corley, M., Burns, M. C. & Yeo, G. W. How RNA-Binding Proteins Interact with RNA: Molecules and Mechanisms. *Mol. Cell* **78**, 9–29 (2020).
18. Krüger, D. M., Neubacher, S. & Grossmann, T. N. Protein-RNA interactions: structural characteristics and hotspot amino acids. *RNA* **24**, 1457–1465 (2018).
19. Li, S., Yamashita, K., Amada, K. M. & Standley, D. M. Quantifying sequence and structural features of protein-RNA interactions. *Nucleic Acids Res.* **42**, 10086–98 (2014).
20. Yan, J. & Kurgan, L. DRNAPred, fast sequence-based method that accurately predicts and discriminates DNA- and RNA-binding residues. *Nucleic Acids Res.* **45**, e84 (2017).

Cryo-EM structure of the J-K-St/eIF4GHEAT1/eIF4A complex:

#2-12. Paragraph 2, lines 10-11: “All of these bases, except for A724, interact mainly with the positively-charged patch on eIF4GHEAT1...” Consider listing some of these residues, and making a reference to Supplementary Figure 3b and c so that the reader can find the side chain depiction of these sites.

We have now listed all the residues in the main text and made a reference to previous Supplementary Figure 3b (now Supplementary Fig. 4b), as follows. Thanks to the suggestion, it is now a lot easier for the reader to find the side chain depiction of these sites.

Results

P.8, lines 133-138

In the J domain, G702, U703, A704, C705, U722, G723, A724, and U725 interact with eIF4G^{HEAT1} from the minor groove side (Fig. 1f and Supplementary Fig. 4b). All of these bases, except for A724, interact mainly with the positively-charged patch on eIF4G^{HEAT1} **including side chain atoms of Arg 840, Lys841, Lys922, Lys925, and Arg954**, whereas the nucleobase of A724 is extruded from the stem to deeply interact with the negatively-charged cleft **including sidechain atoms of Asp918**.

Supplementary Fig. 4 | Analyses of the J-K-St/eIF4G^{HEAT1}/eIF4A structure

(a) Structural alignment of the eIF4G^{HEAT1}/eIF4A complex in the absence and presence of J-K-St. The human eIF4G^{HEAT1} (brown) / human eIF4A (cyan) dimer from the J-K-St/ eIF4G^{HEAT1}/eIF4A ternary complex (this study) is overlaid with the yeast eIF4G^{HEAT1} (yellow) / yeast eIF4A (violet) dimer (PDB ID: 2VSO). Even though the primary sequences are different, with sequence identities of 64.2% for eIF4A and 28.8% for eIF4G^{HEAT1}, the r.m.s.d. value of the two dimers is as low as 2.4 Å, indicating that the structural rearrangement upon the interaction with J-K-St is small.

(b) eIF4G^{HEAT1} residues interacting with the J domain in the J-K-St/eIF4G^{HEAT1}/eIF4A complex. Cryo-EM map of the J domain from J-K-St is shown as a purple surface, where the nucleobase extruding from the J domain, A724, is labeled, and the model the J-K-S RNA is shown in stick representation. A part of the eIF4G^{HEAT1} backbone structure is shown in cartoon representation (brown), and the side chains at the RNA-protein interface are shown in stick representation.

(c) eIF4G^{HEAT1} residues interacting with the St domain in the J-K-St/eIF4G^{HEAT1}/eIF4A complex. Cryo-EM map of the St domain from J-K-St is shown as a purple surface, where the nucleobase extruding from the St domain, G777, is labeled, and the model the J-K-S RNA is shown in stick representation. A part of the eIF4G^{HEAT1} backbone structure is shown in cartoon representation (brown), and the side chains at the RNA-protein interface are shown in stick representation. Lys826, whose side chain is inserted within the cavity formed by the flipping-out of G777, is highlighted by a box.

#2-13. Paragraph 2, lines 13-14: “All of these residues, except for G777, interact mainly with the positively-charged patch on the eIF4G^{HEAT1}...” As written, it is unclear whether this is the same or a different positively charged patch than described earlier.

Similarly to the previous comment **#2-12**, we have now listed the residues in the main text and made a reference to previous Supplementary Figure 3c (now Supplementary Figure 4c), as follows. Thanks to the suggestion, it is now clear this is a different positively charged patch than described earlier.

Results

P.8, lines 138-144

In the St domain, G690, A691, U692, G693, C776, G777, U778, and C779 also interact with eIF4G^{HEAT1} from the minor groove side (Fig. 1g and Supplementary Fig. 4c). All of these residues, except for G777, interact mainly with the positively-charged patch on the eIF4G^{HEAT1} **including sidechain atoms of Lys826 and Asn838**, whereas the nucleobase of G777 is extruded from the stem to interact with another negatively-charged cleft **including the mainchain carboxy atoms of Thr784, Leu786, and Ala824**, which differs from that contacted by the J domain.

Characterization of the excited states of the StASL domain:

#2-14. Paragraph 3: How did U778C affect binding of eIF4G^{HEAT1} to RNA? This would be nice to tie the NMR data to the ITC data reported for other constructs.

The U778C variant exhibited weakened affinity for eIF4G^{HEAT1} compared to the wild-type (K_d values of larger than 10 μ M and 149 nM for the U778C variant and the wild-type, respectively). The data and interpretation have been written above as the response to the comment **#2-5**. We have also included this point in the main text,

as written above as the response to the comment #2-5.

#2-15. Paragraph 5, lines 11-13: "...as suppressing the dynamics by introducing mutations..." Why were these particular mutations chosen? Please describe why.

These A700C and U780C mutations were chosen to stabilize the ground conformation of the J and StA_{SL} domains, respectively, without changing the bases that directly interact with eIF4G^{HEAT1}, as written above in response to the comment #2-6.

#2-16. Another explanation for the loss of binding in these mutants is that they disrupt the secondary structure of the RNA. Were any experiments done to show that the structure is not disrupted? Something like what is shown in Supplementary Figure 5b and c for the isolated J and StASl constructs could suffice.

Yes, in response to this constructive comment, we conducted the NMR experiments and confirmed that the structures of RNA were not disrupted (Fig. R11a). We have compared the ¹H8-¹³C8 aromatic TROSY spectra of A700C, U778C, and U780C mutants of the J and StA_{SL} domain, respectively, with the corresponding wild-type isolated domains. As a result, from the overlay of the signals that are not close to the mutated bases, it was demonstrated that the secondary structures of the isolated J and StA_{SL} domains are not disrupted by these mutations. Furthermore, we have also compared the spectra of the A700C, U778C, and U780C variants of J-K-St with the wild-type, in order to investigate if the secondary structures were disrupted or not (Fig. R11b). As for the isolated domains, it was demonstrated that these mutations do not disrupt the secondary structure of J-K-St, further confirming that the secondary structures were not disrupted by these mutations. Since these mutations were introduced to the residues that do not directly interact with eIF4G^{HEAT1}, it was concluded that the conformational dynamics, which allow for the conformational transition from the ground conformations to the excited conformations, are important for these domains to interact with the eIF4G^{HEAT1}.

Fig. R11 Comparison of the $^1\text{H}_8$ - $^{13}\text{C}_8$ aromatic TROSY

(a) Overlay of the $^1\text{H}_8$ - $^{13}\text{C}_8$ aromatic TROSY spectra of the isolated J domain A700C (top left), StA_{SL} domain U778C (top middle), and StA_{SL} domain U780C (top right) variants with the

corresponding wild-type isolated domains. All spectra were acquired at 30°C and at the ^1H frequency of 800 MHz. Mutated variants and the wild-type are shown in red and black, respectively. The signals perturbed by the mutation is labeled in green, and mapped onto the secondary structure (bottom). Perturbed bases are located neighboring to the mutated bases, corroborating that overall secondary structures are not disrupted by the mutation.

(b) Overlay of the ^1H - ^{13}C aromatic TROSY spectra of the J-K-St A700C (left), U778C (middle), and U780C (right) variants with the corresponding wild-type J-K-St. All spectra were acquired at 35°C and at the ^1H frequency of 900 MHz. Although the signals observed for J-K-St is not assigned, more than 80% of the signals observed for the variants are overlaid with those from the wild-type, indicating that the overall secondary structures of these variants are not disrupted by the mutation.

In response to the reviewer's comment, we have added Supplementary Figs. 12 and 14, and referred these from the main text, as follows.

Results

P.18 lines 327-329

Next, we conducted ITC experiments to investigate whether the conformational dynamics observed for the two eIF4G^{HEAT1} binding sites on the J and StA_{SL} domains are critical for the interaction with eIF4G^{HEAT1} (Supplementary Fig. 12).

P.20 lines 346-352

NMR R_{ex} analyses of these isolated domain variants revealed that the dynamics in the μs -ms timescale are suppressed (Supplementary Fig. 14, c and f), while the secondary structures of these isolated domains are not altered by the mutation (Supplementary Fig. 14, a and d). The ITC experiments revealed that A700C and U780C variants of J-K-St possess reduced affinities for eIF4G^{HEAT1} (Supplementary Fig. 12, a and b), indicating that the suppression of the conformational dynamics to stabilize the ground conformation results in decreased affinities for eIF4G^{HEAT1}.

Supplementary Fig. 12 | Correlation between the NMR-observed dynamics and function

(a) Overlay of the $^1\text{H}8$ - $^{13}\text{C}8$ aromatic TROSY spectra of the J-K-St U778C (left), A700C (middle), and U780C (right) variants with the corresponding wild-type J-K-St. All spectra were acquired at 35°C and at the ^1H frequency of 900 MHz. **Although the signals observed for J-K-St are not assigned, more than 80% of the signals observed for the variants are overlaid with those from the wild-type, indicating that the overall secondary structures of these variants are not disrupted by the mutation.**

(b) ITC experiments of the J-K-St variants. U778C, A700C, and U780C variants exhibited a lower affinity (K_d value larger than 10 μ M) for eIF4G^{HEAT1}, compared to the wild-type ($K_d = 149 \pm 12$ nM). N.D., not determined. The ITC experiments were conducted at least two times with similar results.

(c) *In vitro* translation assay. β -galactosidase assay using a human cell-derived in vitro coupled transcription/translation system (n=7 independent replicates). Total amounts of β -galactosidase translated from the EMCV IRES-mediated translation were quantified as luminescence emitted as a result of β -galactosidase reaction. Data are presented as mean \pm standard error of the mean (s.e.m.). Statistical significance was determined by two-tailed unpaired Student's t test.

Supplementary Fig. 14 | Suppression of the conformational dynamics by mutations

(a) Overlay of the ^1H - ^{13}C aromatic TROSY spectra of the J domain A700C variant (red) with the wild-type (black). All spectra were acquired at 30°C and at the ^1H frequency of 800 MHz. The signals perturbed by the mutation are indicated with green labels.

(b) Mapping of the bases perturbed by the A700C mutation on the secondary structure. **Perturbed bases are located neighboring to the mutated bases, corroborating that overall secondary structure is not disrupted by the mutation.** A700C mutation is designed to form a base pair with G723, suppressing the conformational dynamics observed in the J domain bulge, without substituting the bases that directly interact with eIF4G^{HEAT1} (highlighted with purple lines).

(c) R_{ex} analyses of the A700C variant of the J domain. R_{ex} values observed for the J domain bases are largely suppressed by the mutation (see Fig. 3a for wild type), indicating that the conformational dynamics in the J domain are suppressed. Error bars indicate experimental errors derived from the signal-to-noise ratio of each correlation, as written in Methods.

(d) Overlay of the ^1H - ^{13}C aromatic TROSY spectra of the StA_{SL} domain U780C variant (red) with the wild-type (black). All spectra were acquired at 30°C and at the ^1H frequency of 800 MHz. The signals perturbed by the mutation are indicated with green labels.

(e) Mapping of the bases perturbed by the U780C mutation on the secondary structure. **Perturbed bases are located neighboring to the mutated bases, corroborating that overall secondary structure is not disrupted by the mutation.** U780C mutation is designed to stabilize the base pair with G689, suppressing the register-shift observed in the upper stem of the StA_{SL} domain, without substituting the bases that directly interact with eIF4G^{HEAT1} (highlighted with purple lines).

(f) R_{ex} analyses of the U780C variant of the StA_{SL} domain. R_{ex} values observed for the StA_{SL} domain bases are largely suppressed by the mutation (see Fig. 3b for wild type), indicating that the conformational dynamics in the StA_{SL} domain are suppressed. Error bars indicate experimental errors derived from the signal-to-noise ratio of each correlation, as written in Methods.

Discussion:

#2-17. Paragraph 2, line 1: “EMCV IRES enables this uninvited interaction...” See comment above about previous data showing intrinsic RNA binding activity of eIF4G.

We have carefully considered the reviewer's comment regarding the intrinsic RNA binding activity of eIF4G, and have deleted this description from the Discussion section. Please refer to our response to comment #2-2 above for details of the changes made.

Methods:

#2-18. Is there a reason why the buffers for the cryo-EM and NMR samples were so different? The pH 7.5 vs. pH 6.4 and 150 mM salt vs. 10 mM salt alone could modulate RNA structure.

Yes, there is a reason for the differences in the cryo-EM and NMR samples. Since eIF4G^{HEAT1} is prone to aggregate under the low pH and low salt concentration buffer condition, where our original NMR structural analyses (ref 9) were conducted, we needed to carry out the cryo-EM and ITC experiments in high pH and high salt concentration buffer. As the reviewer pointed out, RNA structures can be modulated by the differences in the pH and/or salt concentrations. Indeed, when we compared the NMR spectra of the J-K-St in the buffer used

in the other NMR experiments (10 mM potassium phosphate (pH 6.4), 10 mM NaCl), and the high salt concentration and high pH buffer (20 mM Hepes-NaOH (pH 7.4), 150 mM NaCl), some spectral changes, possibly due to the interaction with the cations and the negatively charged phosphate groups in J-K-St, were observed (Fig. R12). However, the overall signal distributions are similar and more than 80% of the signals are overlapped, indicating that the structure of J-K-St is virtually identical between these buffer conditions. Importantly, the signals from the A_{SL} domain were not largely perturbed, indicating that the structure of the A_{SL} domain does not change due to the differences between the ion concentrations or the pH values.

Fig. R12 Effect of salt concentration and pH

(a) $^1\text{H}_8$ - $^{13}\text{C}_8$ aromatic TROSY spectra of [u - ^2H , $\{^1\text{H}_2, ^1\text{H}_8, ^{13}\text{C}_8\}$ -Ade, $\{^1\text{H}_8, ^{13}\text{C}_8\}$] J-K-St at 35°C and at the ^1H frequency of 900MHz. Black: in buffer containing 10 mM potassium phosphate (pH 6.4), 10 mM NaCl, red: in buffer containing 20 mM Hepes-NaOH (pH 7.5), 150 mM NaCl.

(b) (left) The expanded region of (a) containing signals from the A_{SL} domain, A771, A772, A773, and A774. (right) Mg^{2+} dependence of the signals. Spectral change under the Mg^{2+} concentrations of 0, 1.2, and 2.5 mM in buffer containing 10 mM potassium phosphate (pH 6.4), 10 mM NaCl is shown.

According to the reviewer's comment, we added Supplementary Fig. 5b, and referred to these data from the main text, as follows. Thanks to the insightful suggestion, the effect of the differences in salt concentrations and pH values is now clearly shown.

Results

P.10 lines 175-179

To gain insights into the mechanism underlying the capture of the host target protein by the viral RNA, we used solution NMR to characterize the dynamic properties of the eIF4G^{HEAT1}-binding sites of J-K-St. To this end, we employed [$u\text{-}^2\text{H}$, Ade- $\{^1\text{H}2, ^1\text{H}8, ^{13}\text{C}8\}$, Gua- $\{^1\text{H}8, ^{13}\text{C}8\}$] isotope labeling of RNA samples, and used it in combination with ^1H - ^{13}C aromatic transverse relaxation optimized spectroscopy (TROSY)²¹ (Supplementary Fig. 5), which enables observation of NMR signals that reflect on the structure and dynamics of RNA in high sensitivity.

Supplementary Fig. 5 | Site-selective isotope labeling and buffer effects in NMR analyses

(a) Adenosine (left) and guanosine (right) are site-specifically isotope-labeled at the C8 positions, whereas the riboses are perdeuterated, thus alleviating the strong inter- and/or intra-residual dipole-dipole interactions that broaden the NMR signals. The ribose and nucleobase moieties of uridines and cytosines used in this study are perdeuterated.

(b) ¹H₈-¹³C₈ aromatic TROSY spectra of [u-²H, {¹H₂, ¹H₈, ¹³C₈}-Ade, {¹H₈, ¹³C₈}-Gua] labeled J-K-St. The spectra acquired in buffer containing 10 mM potassium chloride (pH 6.4) and 10 mM NaCl, and in buffer containing 20 mM Hepes-NaOH (pH 7.5), 150 mM NaCl are shown in black and red, respectively. Although several signals exhibited chemical shift perturbation or broadening of the line widths between these two buffer conditions, more than 80% of the observed signals overlapped, indicating that overall secondary structures are virtually identical. On the right, the expanded region containing signals from the A_{SL} domain, marked as green box, are shown. A_{SL} domain signals are not largely perturbed between these two pH and salt concentration conditions.

Figures:

#2-19. Figure 1: consider adding a schematic showing the domains of full-length eIF4G, and highlighting which portions are included in the construct used here.

In response to the comment requesting to add a schematic showing the domains of full-length eIF4G included in the construct used here, in Fig. 1b, as follows. We appreciate the suggestion, as it is now visually clear to the readers which region is defined and used as eIF4G^{HEAT1} here.

Fig. 1 | Cryo-EM structure of the J-K-St/eIF4G^{HEAT1}/eIF4A complex

(a) Schematic representation of the translation initiation complex formed by EMCV IRES. The J-K-St region directly interacts with the scaffolding protein eIF4G of the host cell, leading to the recruitments of eIF4A, eIF3, eIF2, and the ribosomal 40S subunit to the initiation codon. (b) **Domain organization of the full-length eIF4G and the eIF4G^{HEAT1} construct.** (c) NMR structure of J-K-St in the free-state⁹ (PDB ID: 2NBX). (d) Overall structure of the J-K-St/eIF4G^{HEAT1}/eIF4A ternary complex, overlaid with the lowpass-filtered cryo-EM density map. (e) Focused cryo-EM

map (left) and the structure (right) of J-K-St/eIF4G^{HEAT1}. The J, K, and St domains of J-K-St are labeled along with the A_{SL} domain at the junction. Mg²⁺ ions are shown as black spheres. (f and g) Electrostatic potential representations of the surface of eIF4G^{HEAT1} at the interface between the J domain (f) and the St domain (g), contoured from -5 kT/e (red) to +5 kT/e (blue). The residues of the J-K-St in contact with the eIF4G^{HEAT1} are shown as sticks. The labels of the extruded residues, A724 on the J domain (f) and G777 on the St domain (g), are highlighted with boxes. (h) Comparison of the domain orientations. The structures of J-K-St in the free-state and in complex with eIF4G^{HEAT1}/eIF4A are shown as surface and ribbon-and-stick representations, respectively. The J, K, St, and A_{SL} domains are colored green, blue, and orange, and red, respectively. The structures are aligned with the K domain, and changes in the J and St domain orientations upon complex formations are shown as black arrows.

#2-20. Figure 1a: this diagram makes it seem like J-K-St is sufficient to promote translation initiation. Consider including a cartoon of the full IRES element with J-K-St highlighted.

In response to the comment requesting to include a cartoon of the full IRES element with J-K-St highlighted, we modified Fig. 1a, as shown above (**#2-19**). We appreciate the suggestion, as it is now illustrated that J-K-St is not sufficient to promote translation initiation.

#2-21. Figure 2a: what do the light grey U residues in the K domain mean? Are these not modeled in the structure? Please indicate in the figure legend.

Yes, the light grey U residues in the K domain are the residues not modeled in the structure. We indicated this point in the figure legend, as follows. We appreciate the reviewer's suggestion to clearly define the color scheme.

Fig. 2 | Comparison of the secondary structures of J-K-St

(a) The secondary structure of J-K-St in the complex with eIF4G^{HEAT1} and eIF4A, determined by cryo-EM in this study. Bases that are not modeled are shown in light gray. (b) The secondary structure of the J-K-St in the free-state, determined by NMR⁹. Lines between bases indicate the canonical Watson-Crick base pairs, dots indicate the wobble base pairs, and dashed lines are for non-canonical base pairs. The eIF4G^{HEAT1}-binding residues are highlighted with purple lines. The J, K, St, and A_{SL} domains are colored green, blue, orange, and red, respectively. The differences between these two structures are highlighted by gray backgrounds.

#2-22. Supplementary Figure 1a: the authors use surface charge as an argument supporting no intrinsic RNA binding ability of eIF4G. Are there examples of other ‘known’ RNA binding proteins that could be shown alongside 4GHEAT1 to show the difference in appearance claimed in the text?

Yes, there are examples. We included the structures of some of the “known” RNA binding proteins, namely, RNA recognition motif (RRM), Arginine-rich motif, and double-stranded RNA binding domain (dsRBD), alongside eIF4G^{HEAT1} as examples of the surface charge distribution of the typical RNA binding proteins (Supplementary Fig. 1). Although these RNA binding proteins possess a surface where acidic residues (Asp and Glu, shown in red in Supplementary Fig. 1) are absent, and is rich in basic residues (Lys and Arg, shown in blue in Supplementary Fig. 1) or Phe/Tyr residues (shown in green in Supplementary Fig. 1). In contrast, the surface of eIF4G^{HEAT1} is rich in basic residues, which might be involved in the relatively weak non-specific interaction

with RNAs, but there are also acidic residues that neutralize the charge and breaks the continuous surface for specific RNA binding. This visual inspection is supported by the fact that the sequence and structure-based scoring of RNA-binding propensity did not predict any binding surface on the HEAT1 domain of eIF4G (Li *et al.*, *Nucleic Acids Res.* (2014) 42:10086–98 doi:10.1093/nar/gku681). Accordingly, we modified the main text and Supplementary Fig. 1 as follows. Thanks to the suggestion, these figures now illustrate the difference in appearance described in the text.

Red: Glu, Asp, Blue: Lys, Arg, Green: Phe, Tyr

eIF4G^{HEAT1} in the free state

RNA recognition motif (RRM)

Arginine rich motif

dsRNA binding domain (dsRBD)

Supplementary Fig. 1 | Comparison of the surface residue distributions of eIF4G^{HEAT1} and typical RNA binding proteins

Surface representations of the eIF4G^{HEAT1} homolog (the middle domain of eIF4GII, PDB ID: 1HU3) and typical RNA binding proteins. The RNA recognition motif (RRM) from human antigen R (PDB ID: 6GC5), the arginine rich motif from HIV Rev protein (PDB ID: 4PMI), and the double stranded RNA binding domain (dsRBD) from Staufen1 (PDB ID: 6HTU) are shown as surface where the bound RNAs are shown as violet sticks. Glu and Asp residues are colored red, Lys and Arg residues are colored blue, and Phe and Tyr residues are colored green. For RRM, arginine binding motif, and dsRBD, RNA binding surfaces are marked by dashed orange lines. The acidic residues, Glu and Asp, are rarely found on RNA–protein interfaces, while the basic and aromatic residues, Lys, Arg, Tyr, and Phe, are frequently found on RNA–protein interfaces. There is no surface area on eIF4G^{HEAT1} that is convincing as the RNA-binding site¹², as corroborated by the structure- and sequence-based predictions¹⁹.

#2-23. Supplementary Figure 1: how many replicates of the ITC experiment were performed? How were the reported errors determined? Please indicate in the legend or methods.

For the J-K-St and eIF4G^{HEAT1} binding in the presence of 2 mM Mg²⁺ (Supplementary Fig. 2a), three replicates of the ITC experiment were performed with similar results. For the other ITC experiments, at least two replicates were performed with similar results. The reported error values represent the standard error of the ITC fitting. We have now indicated these in the Methods and the legend of Supplementary Figs 2, 12 and 13, as follows.

Methods

P.36, lines 651-662

Isothermal Titration Calorimetry

Before all isothermal titration calorimetry (ITC) experiments, the eIF4G^{HEAT1} and RNAs were dissolved in ITC buffer (20 mM HEPES-NaOH, pH 6.5, 150 mM NaCl, 2 mM MgCl₂, and 1 mM Tris(2-carboxyethyl)phosphine hydrochloride), unless otherwise indicated. For each ITC experiment, reaction heats (in $\mu\text{cal s}^{-1}$) were measured for 2- μl titrations of 150–250 μM of eIF4G^{HEAT1} into 10–20 μM of RNA at 25°C, with a Microcal PEAQ-ITC (Malvern). The titration data were analyzed using the one-site binding model from MicroCal PEAQ-ITC Analysis Software (Malvern). The K_d values are reported when the c value, which is the ratio of the RNA concentration to the calculated K_d value, is smaller than 5. Error values represent the standard error of the ITC fit. The experiments for the interaction between the J-K-St wild-type and the eIF4G^{HEAT1} wild-type in the presence of 2 mM Mg²⁺ were repeated three times with similar results. All the other experiments were repeated at least twice with similar results.

Supplementary Fig. 2 | ITC experiments

(a, b) ITC data of the interaction between J-K-St and eIF4G^{HEAT1}, in the presence of 2 mM Mg²⁺ (a) and in the absence of Mg²⁺ (b).

(c) ITC data of the interaction between St-extended J-K-St and eIF4G^{HEAT1}, in the presence of 2 mM Mg²⁺. Extended regions at the terminus of the St domain are schematically shown in green in the inset. See also (f) for the reduced stoichiometry.

(d) ITC data of the interaction between the J domain and eIF4G^{HEAT1}, in the presence of 2 mM Mg²⁺. N.D., not determined. See Supplementary Fig. 6a for the design of the J domain.

(e) ITC data of the interaction between the StA_{SL} domain and eIF4G^{HEAT1} in the presence of 2 mM Mg²⁺. N.D., not determined. See Supplementary Fig. 6a for the design of the StA_{SL} domain.

(f) Native polyacrylamide gel electrophoresis (PAGE) analysis of J-K-St and St-extended J-K-St. Bands corresponding to the dimer and trimer were observed only for St-extended J-K-St. These multimers, formed due to misfolding, decrease the effective concentration of the variant with respect to that calculated from the absorbance at 260 nm, resulting in the reduced stoichiometry (N = 0.50 ± 0.01) observed in (c). It should be noted here that discrepancy of the effective concentration of the sample in the ITC cell (i.e., the RNA) does not affect the K_d values obtained in the one-site fitting model employed here. The misfolded molecules were not included in the cryo-EM analyses, as the samples were purified before the cryo-EM analyses, ensuring that only the RNA molecules

that formed a complex with eIF4G^{HEAT1}/eIF4A were used.

N.D., not determined. The ITC experiment (a) was repeated three times with similar results. The ITC experiments (b-e) were conducted at least two times with similar results. The \pm values indicate the standard error of the fitting.

Supplementary Fig. 12 | Correlation between the NMR-observed dynamics and function

(a) Overlay of the ^1H - ^{13}C aromatic TROSY spectra of the J-K-St U778C (left), A700C (middle), and U780C (right) variants with the corresponding wild-type J-K-St. All spectra were acquired at 35°C and at the ^1H frequency of 900 MHz. Although the signals observed for J-K-St is not assigned, more than 80% of the signals observed for the variants are overlaid with those from the wild-type, indicating that the overall secondary structures of these variants are not disrupted by the mutation.

(b) ITC experiments of the J-K-St variants. U778C, A700C, and U780C variants exhibited a lower affinity (K_d value larger than $10\ \mu\text{M}$) for eIF4G^{HEAT1}, compared to the wild-type ($K_d = 149 \pm 12\ \text{nM}$). N.D., not determined. **The ITC experiments were conducted at least two times with similar results.**

(c) *In vitro* translation assay. β -galactosidase assay using a human cell-derived *in vitro* coupled transcription/translation system ($n=7$ independent replicates). Total amounts of β -galactosidase translated from the EMCV IRES-mediated translation were quantified as luminescence emitted as a result of β -galactosidase reaction. Data are presented as mean \pm standard error of the mean (s.e.m.). Statistical significance was determined by two-tailed unpaired Student's t test.

Supplementary Fig. 13 | Interaction of U778 in J-K-St and Lys826 in eIF4G^{HEAT1}

(a) Close-up view of the U778-Lys826 interaction. The side chain of Lys826 of eIF4G^{HEAT1} is extruded to stack onto the pyrimidine ring of U778.

(b) The ITC experiment of the J-K-St wild-type and eIF4G^{HEAT1} K826A. N.D., not determined (K_d larger than $10\ \mu\text{M}$). **The ITC experiment was conducted at least two times with similar results.**

Reviewer #3 (Remarks to the Author):

#3-1. Imai et al describe in this manuscript a structural characterization of a fragment of the EMCV IRES in complex with the eukaryotic initiation factor eIF4. IRESs of this sub-type are highly understudied at a structural level what makes this study interesting as it contributes insights in the early recruitment of the EMCV IRES to the ribosome to hijack the protein synthesis capacity of the host cell. The study combines two structural methodologies, namely cryoEM and NMR and a solid biochemical characterization. Overall, I find the study well designed and executed with a clear and accurate presentation, it is also technically sound. I only have minor concerns which have to do mainly with the visualization of the cryoEM map obtained and the refinement of the atomic model in the cryoEM maps.

We greatly appreciate your positive evaluation of our study. We also appreciate your constructive comments to improve the quality and visualization of the cryo-EM structure in our manuscript. According to your suggestions, we have refined the cryo-EM map and model and revised the figure presentations. Please find our point-to-point responses to your comments below.

#3-2. The quality of the cryoEM map is hard to ascertain as only a far view of the map is shown in figure 1c. The map seems to agree with the reported resolution, but a much-detailed graphical explanation is needed. A multipaneled and multiscale subfigure should be provided, focusing in the most interesting regions of the complex so the reader can confidently assess the quality of the map in those regions with key interactions. Similarly, in supplementary figure 2, much-detailed views of the local resolution figure are needed so the reader can unambiguously judge the cryoEM quality map in all areas.

We agree with the comments that the graphical explanation should be improved for assessment of the reliability of the cryo-EM map and the model. As we described in responses **#2-4** and **#3-3**, we further processed the cryo-

EM data and obtained the J-K-St/eIF4G^{HEAT1}-focused cryo-EM map, using the original full-map to show the overall structure of the ternary complex and the new focused map to show the detailed structural information of the protein-RNA interactions. Accordingly, we have revised the previous Fig. 1c to now Fig. 1 d and e. We have also revised the figure panels of the model fit to the cryo-EM map as in Supplementary Fig. 3o to clearly indicate the representative regions. In addition, we have revised Supplementary Fig. 4 b and c by using surface representation of the cryo-EM map at the particularly important regions of the RNA-protein interface.

In response to the comment on the local resolution, we have revised the figure as in Supplementary Fig. 3 m and n with the local resolution maps (for the full- and focused-maps) in views from different orientations.

Fig. 1 | Cryo-EM structure of the J-K-St/eIF4G^{HEAT1}/eIF4A complex

(a) Schematic representation of the translation initiation complex formed by EMCV IRES. The J-K-St region directly interacts with the scaffolding protein eIF4G of the host cell, leading to the recruitments of eIF4A, eIF3, eIF2, and the ribosomal 40S subunit to the initiation codon. (b) Domain organization of the full-length eIF4G and the eIF4G^{HEAT1} construct. (c) NMR structure of J-K-St in the free-state⁹ (PDB ID: 2NBX). (d) Overall structure of the J-K-St/eIF4G^{HEAT1}/eIF4A ternary complex, overlaid with the lowpass-filtered cryo-EM density map. (e) Focused cryo-EM

map (left) and the structure (right) of J-K-St/eIF4G^{HEAT1}. The J, K, and St domains of J-K-St are labeled along with the A_{SL} domain at the junction. Mg²⁺ ions are shown as black spheres. (f and g) Electrostatic potential representations of the surface of eIF4G^{HEAT1} at the interface between the J domain (f) and the St domain (g), contoured from -5 kT/e (red) to +5 kT/e (blue). The residues of the J-K-St in contact with the eIF4G^{HEAT1} are shown as sticks. The labels of the extruded residues, A724 on the J domain (f) and G777 on the St domain (g), are highlighted with boxes. (h) Comparison of the domain orientations. The structures of J-K-St in the free-state and in complex with eIF4G^{HEAT1}/eIF4A are shown as surface and ribbon-and-stick representations, respectively. The J, K, St, and A_{SL} domains are colored green, blue, and orange, and red, respectively. The structures are aligned with the K domain, and changes in the J and St domain orientations upon complex formations are shown as black arrows.

Supplementary Fig. 3 | Cryo-electron microscopy analyses

(a) Size exclusion chromatography profile of the J-K-St/eIF4G^{HEAT1}/eIF4A ternary complex. The positions corresponding to the elutions of the complex and free components are labeled and indicated by arrows.

(b) SDS-PAGE analysis of the peak fraction corresponding to the ternary complex. Proteins in the gel were stained with Coomassie Brilliant Blue G-250.

(c) Representative cryo-EM micrograph of the vitrified J-K-St/eIF4G^{HEAT1}/eIF4A complex. Green circles indicate examples of individual particles. Scale bar is 50 nm.

(d) Selected 2D-class averages of the ternary complex.

(e) Low-pass-filtered Cryo-EM maps obtained from the pilot data of the ternary complexes containing WT J-K-St (left, grey) or St-extended J-K-St (right, green), collected on JEM-Z320FHC. Red arrows indicate the region of an extra density observed only in the St-extended J-K-St/eIF4G^{HEAT1}/eIF4A ternary complex.

(f) Cryo-EM data-processing workflow for the St-extended J-K-St/eIF4G^{HEAT1}/eIF4A ternary complex collected on JEM-Z320FHC.

(g and h) The refined cryo-EM map of the whole complex (g) and the refined map with local alignment and a mask (inset) focused on the region of J-K-St/eIF4G^{HEAT1} (h).

(i and j) Angular distribution plot of all particles that contributed to the refined map of the whole complex (i) and the J-K-St/eIF4G^{HEAT1}-focused map (j).

(k and l) Gold-standard Fourier shell correlation (FSC) curves for the whole complex map (k) and the J-K-St/eIF4G^{HEAT1}-focused map (l) after correction for masking effects (black line). The phase randomized FSC curves are colored red. The map resolutions were estimated based on the FSC = 0.143 criterion.

(m and n) Local resolutions for the whole complex map (m) and the focused map (n) were calculated using ResMap⁴⁸.

(o) Cryo-EM densities and refined models for selected regions at the St/A_{SL} domain (lower left) and the J domain (lower right) of J-K-St RNA, eIF4G^{HEAT1} (upper right), and eIF4A (upper left) in the ternary complex.

(p and q) Cross-validation FSC curves of the refined models versus the final maps. The correlation is above 0.5 up to resolutions of 3.9 Å and 3.8 Å for the ternary complex (p) and the focused region of J-K-St/eIF4G^{HEAT1} (q), respectively.

Supplementary Fig. 4 | Analyses of the J-K-St/eIF4G^{HEAT1}/eIF4A structure

(a) Structural alignment of the eIF4G^{HEAT1}/eIF4A complex in the absence and presence of J-K-St. The human eIF4G^{HEAT1} (brown) / human eIF4A (cyan) dimer from the J-K-St/ eIF4G^{HEAT1}/eIF4A ternary complex (this study) is overlaid with the yeast eIF4G^{HEAT1} (yellow) / yeast eIF4A (violet) dimer (PDB ID: 2VSO). Even though the primary sequences are different, with sequence identities of 64.2% for eIF4A and 28.8% for eIF4G^{HEAT1}, the r.m.s.d. value of the two dimers is as low as 2.4 Å, indicating that the structural rearrangement upon the interaction with J-K-St is small.

(b) eIF4G^{HEAT1} residues interacting with the J domain in the J-K-St/eIF4G^{HEAT1}/eIF4A complex. Cryo-EM map of the J domain from J-K-St is shown as a purple surface, where the nucleobase extruding from the J domain, A724, is labeled, and the model the J-K-S RNA is shown in stick representation. A part of the eIF4G^{HEAT1} backbone structure is shown in cartoon representation (brown), and the side chains at the RNA-protein interface are shown in stick representation.

(c) eIF4G^{HEAT1} residues interacting with the St domain in the J-K-St/eIF4G^{HEAT1}/eIF4A complex. Cryo-EM map of the St domain from J-K-St is shown as a purple surface, where the nucleobase extruding from the St domain, G777, is labeled, and the model the J-K-S RNA is shown in stick representation. A part of the eIF4G^{HEAT1} backbone structure is shown in cartoon representation (brown), and the side chains at the RNA-protein interface are shown in stick representation. Lys826, whose side chain is inserted within the cavity formed by the flipping-out of G777, is highlighted by a box.

#3-3. The final statistics for the atomic model refined into the cryoEM map are not ideal. A cryoEM map at 3.7Å resolution, granted a “modest” resolution, should be enough to obtain a refined model with a molprobity all-atom clashscore better than the reported 24.89. The authors should inspect and manually fix the model in problematic areas and re-refine the fixed model with the aim to obtain molprobity score ideally in the single digit range.

During the revision process, we realized a small error in the pixel size of micrographs collected on the microscope and re-calibrated the magnification more precisely. The calibrated pixel size is now updated from 0.75 to 0.765 Å on the specimen level. An iterative cycle of re-refinement of the model to the re-postprocessed map with the re-calibrated pixel size and careful inspection in COOT resulted into a better all-atom clashscore (13.6) for the J-K-St/eIF4G^{HEAT1}/eIF4A complex. By omitting eIF4A from the model, as we described in the response **#2-4**, the new model refined into the cryo-EM map focused on the J-K-St/eIF4G^{HEAT1} region showed an all-atom clashscore of 9.2. Accordingly, we revised Supplementary Table 1. Thanks to the suggestion, we now believe the models are more reliable in terms of fitting as well as geometry.

Supplementary Table 1 | Cryo-EM data collection and refinement statistics.

	J-K-St/eIF4G ^{HEAT1} /eIF4A	J-K-St/eIF4G ^{HEAT1} (Focused)
EMDB ID	EMD-35041	EMD-36046
PDB ID	8HUI	8J7R
Data collection		
Microscope	JEM-Z320FHC	
Detector	K2 Summit	
Pixel size (Å)	0.765	
Defocus range (µm)	-1.0 to -2.5	
Voltage (kV)	300	
Total electron dose (e ⁻ Å ⁻²)	68.3	
Reconstruction		
Final particle number	255,256	
Pixel size (Å)	1.1475	
Box size (pixels)	200	
Map resolution (Å) (FSC = 0.143)	3.76	3.70
Map sharpening B-factor (Å ⁻²)	-110	-132
Refinement		
Model composition		
Non-hydrogen atoms	7,143	4,034
Protein residues	621	238
Nucleotide residues	99	97
Water	1	1
Ligands	Mg: 3	Mg: 3
Map CC (mask)	0.78	0.76
Map CC (volume)	0.78	0.76
R.m.s. deviations		
Bond length (Å)	0.005	0.010
Bond angles (°)	1.012	1.340
Validation		
MolProbity score	2.2	2.2
All-atom clashscore	13.6	9.2
Ramachandran plot		
Outliers (%)	0.0	0.0
Allowed (%)	3.2	2.5
Favored (%)	96.8	97.5
Rotamer Outliers (%)	2.5	6.85
C-beta deviations (%)	0.0	0.0

#3-4. A phase randomized FSC curve should be provided in supplementary figure 2 to discard artefacts generated by a tight mask during postprocessing as described in: <https://pubmed.ncbi.nlm.nih.gov/23872039/>

According to the reviewer's suggestion, phase randomized FSC curves have been added to Supplementary Fig.

3, k and l in the revised manuscript, as follows.

Supplementary Fig. 3 | Cryo-electron microscopy analyses

- (a) Size exclusion chromatography profile of the J-K-St/eIF4G^{HEAT1}/eIF4A ternary complex. The positions corresponding to the elutions of the complex and free components are labeled and indicated by arrows.
- (b) SDS-PAGE analysis of the peak fraction corresponding to the ternary complex. Proteins in the gel were stained with Coomassie Brilliant Blue G-250.
- (c) Representative cryo-EM micrograph of the vitrified J-K-St/eIF4G^{HEAT1}/eIF4A complex. Green circles indicate examples of individual particles. Scale bar is 50 nm.
- (d) Selected 2D-class averages of the ternary complex.
- (e) Low-pass-filtered Cryo-EM maps obtained from the pilot data of the ternary complexes containing WT J-K-St (left, grey) or St-extended J-K-St (right, green), collected on JEM-Z320FHC. Red arrows indicate the region of an extra density observed only in the St-extended J-K-St/eIF4G^{HEAT1}/eIF4A ternary complex.
- (f) Cryo-EM data-processing workflow for the St-extended J-K-St/eIF4G^{HEAT1}/eIF4A ternary complex collected on JEM-Z320FHC.
- (g and h) The refined cryo-EM map of the whole complex (g) and the refined map with local alignment and a mask (inset) focused on the region of J-K-St/eIF4G^{HEAT1} (h).
- (i and j) Angular distribution plot of all particles that contributed to the refined map of the whole complex (i) and the J-K-St/eIF4G^{HEAT1}-focused map (j).
- (k and l) Gold-standard Fourier shell correlation (FSC) curves for the whole complex map (k) and the J-K-St/eIF4G^{HEAT1}-focused map (l) after correction for masking effects (black line). The phase randomized FSC curves are colored red. The map resolutions were estimated based on the FSC = 0.143 criterion.
- (m and n) Local resolutions for the whole complex map (m) and the focused map (n) were calculated using ResMap⁴⁸.
- (o) Cryo-EM densities and refined models for selected regions at the St/A_{SL} domain (lower left) and the J domain (lower right) of J-K-St RNA, eIF4G^{HEAT1} (upper right), and eIF4A (upper left) in the ternary complex.
- (p and q) Cross-validation FSC curves of the refined models versus the final maps. The correlation is above 0.5 up to resolutions of 3.9 Å and 3.8 Å for the ternary complex (p) and the focused region of J-K-St/eIF4G^{HEAT1} (q), respectively.

REVIEWERS' COMMENTS

Reviewer #1 (Remarks to the Author):

The revised manuscript by Imai et al. has addressed my concerns of the original manuscript. Hence, I highly recommend publication of this excellent work, which should be of significant interests to the readers of Nature Communications.

Point-to-point response to the reviewer's comments

Reviewer #1 :

The revised manuscript by Imai et al. has addressed my concerns of the original manuscript. Hence, I highly recommend publication of this excellent work, which should be of significant interests to the readers of Nature Communications.

Response to Reviewer #1:

We would like to appreciate the reviewer for providing us valuable comments and suggestions, which significantly improved the manuscript.